# Copy number losses of oncogenes and gains of tumor suppressor genes generate common driver mutations

Elizaveta Besedina [1] & Fran Supek [1,2,3] ✉

Cancer driver genes can undergo positive selection for various types of genetic alterations, including gain-of-function or loss-of-function mutations and copy number alterations (CNA). We investigated the landscape of different types of alterations affecting driver genes in 17,644 cancer exomes and genomes. We find that oncogenes may simultaneously exhibit signatures of positive selection and also negative selection in different gene segments, suggesting a method to identify additional tumor types where an oncogene is a driver or a vulnerability. Next, we characterize the landscape of CNA-dependent selection effects, revealing a general trend of increased positive selection on oncogene mutations not only upon CNA gains but also upon CNA deletions. Similarly, we observe a positive interaction between mutations and CNA gains in tumor suppressor genes. Thus, two-hit events involving point mutations and CNA are universally observed regardless of the type of CNA and may signal new therapeutic opportunities. An analysis with focus on the somatic CNA two-hit events can help identify additional driver genes relevant to a tumor type. By a global inference of point mutation and CNA selection signatures and interactions thereof across genes and tissues, we identify 9 evolutionary archetypes of driver genes, representing different mechanisms of (in)activation by genetic alterations.

The abundance of cancer genome sequences has accelerated the discovery of positively selected cancer driver genes (and revealed an apparent rarity of negatively selected genes) by analyzing the local mutation rate of somatic single-nucleotide variants (SNVs) and indels and/or occurrence of hotspots thereof[1–5]. However, accounting for the baseline mutation rate expected under neutrality is a difficult task, bearing in mind the confounding by local mutation risk heterogeneity; this occurs at various overlapping genomic scales[6–8]. Studies have addressed this challenge by modeling the mutation risk from covariates, such as DNA replication time, gene expression level, chromatin modifications, and the mutation type and the oligonucleotide (usually trinucleotide) context. Any significant difference from the neutral

somatic mutation rate modeled thusly is considered as a genomic signature of selection[1,3,5,8,9].

In addition to point mutations and small indels, also the somatic copy number alterations (CNAs) are very commonly observed in cancer genomes and drive tumorigenesis[10–12] by gene dosage changes and/or gene regulation changes. However, while some CNA are selected due to an effect on a specific gene(s), as with the SNVs, the majority of the genes affected by a CNA are non-selected passengers. Methods to identify the positively selected driver CNA from genomic data can be based on recurrence of CNA events, sometimes incorporating external data from gene networks, multi-omic analyses or genetic screens[13–17]. A particular difficulty with ascertaining selection on CNA is pinpointing

[1]Institute for Research in Biomedicine (IRB Barcelona), 08028 Barcelona, Spain. [2]Biotech Research and Innovation Centre (BRIC), University of Copenhagen, 2200 Copenhagen, Denmark. [3]Catalan Institution for Research and Advanced Studies (ICREA), 08010 Barcelona, Spain. ✉e-mail: fran.supek@bric.ku.dk

which is the actual causal gene in a broader CNA segment, which usually affects many neighboring genes. However, a simultaneous occurrence of point mutations at the CNA-affected locus may indicate the selective effect of the CNA on a particular gene. This is because mutations and CNA can be epistatic in the same driver gene, depending on the molecular mechanism by which genetic alterations activate or inactivate a particular driver gene.

An archetypal example of this are the SNVs or indels interacting with CNA deletions at the same locus, affecting two different alleles of a tumor suppressor gene (TSG). This two-hit mechanism is common for deleterious germline variants in cancer predisposition genes, first discovered for *RB1*[18], and later shown for various other cancer risk genes including *ATM* and *BRCA1*[19]. In addition to germline variants, two-hit inactivation mechanisms via CNA are broadly relevant also for somatic SNV mutations in TSG (of note, the two-hits can also occur by a copy number-neutral loss-of-heterozygosity, which is not technically CNA)[20–23]. In addition to TSG, also oncogenes (OG) were appreciated to be affected with selected genetic interactions between CNA gains and point mutations in the same allele: the gene dosage of the mutant allele tends to be increased[20,24]. Such allelic imbalances in oncogenes were well studied experimentally for the *RAS* genes, prominently *KRAS*[25,26], and genomic studies implicate several other oncogenes such as *KIT* and *EGFR*[20,23].

In addition to the positive selection on TSG and OG, a related point of interest is negative somatic selection. Genomic signatures thereof in tumor genomes are very subtle[1,2,4], and so for most genes in most cancer types below the threshold of detection. However, the (impactful but relatively infrequent) nonsense mutations do appear underrepresented in essential genes and in oncogenes considered collectively[27–29], and there is evidence that so are frameshifting indels[30]. Considering selection on SNVs jointly with CNAs may clarify the signatures of negative selection, which was reported to be increased in hemizygous regions[1,28,31]. This helped identify several individual examples of negatively selected genes, for instance, the *POLR2A* essential gene (subunit of RNA polymerase II), genes encoding essential protein complex members as well as protein translation genes, and spliceosome genes[4,20,31]. However, for most genes, including the known essential genes, signatures of negative selection on somatic mutations remain elusive.

Prior studies reported genetic interaction between somatic point mutations and CNA by measuring the statistical co-occurrence of mutations and CNAs across tumor genomes, or by studying the allelic imbalance in driver gene mutations due to CNA. However, the fitness effect of the altered dosage of the mutant allele remains understudied, as well as the modeling of local rate of neutral mutations upon CNA. This is relevant for studies of drivers affected by CNA, because it is anticipated that change in DNA dosage due to CNA− even if selectively neutral − may confound the local mutation risk estimation. The state-of-the-art methods for measuring selection (thus identifying driver genes) that model mutation risk using covariates[1,3] do not consider CNA states and are thus naive to such confounding. Methods using empirical mutation risk baselines such as dN/dS tests or InVEx[32] do inherently control for CNA effects (since the mutation rate baseline is located within the gene), however they have more limited application only to very large cohorts (dN/dS), or only to WGS data (InVEx)[1,4]. Moreover the existing methodologies and tools generally lack the facility to test for conditional somatic selection i.e. the changes in the selective regime between conditions; new methods to address this are being developed[33]. A method that controls for local neutral mutation rate heterogeneity and for changes thereof caused by CNAs, as well as being able to measure significant conditional selection, would be helpful for rigorously studying evolutionary impact of interactions between selected mutations and CNA.

In this study, we apply a custom statistical method for cancer genome analysis (MutMatch) that estimates local mutation rates by drawing on observed mutations in putatively neutral exonic regions, matched by mutation risk and CNA state to the gene-of-interest, and thus controls for various confounding effects on mutation risk. Applying MutMatch for testing conditional selection, we rigorously characterize selected genetic interactions between driver SNVs and CNA in the same gene. This revealed that copy number losses were frequently associated with positive selection not only in TSG but also in OG. Copy number gains were associated with increased selection not only in oncogenes but also in TSG, increasing the dosage of the mutant allele in both cases. Next, we suggest that oncogenes are under negative selection that purges SNVs, particularly in some tumor types where they do not commonly bear driver mutations. Drawing on these patterns − factoring out negative selection, as well as jointly considering CNA and mutations − can increase sensitivity to identify long-tail driver events in oncogenes and TSG, better characterizing their cancer type spectrum. Finally, we perform global analysis to infer signatures of selection on SNVs in different copy-number states of oncogenes and of TSG across ~18,000 tumor genomes. This generated a comprehensive, data-driven atlas of cancer genes categorized into 9 evolutionary archetypes by the types of genetic alterations affecting them, including both SNV and CNA drivers and their interactions across various somatic tissues.

## Results

### A statistical method to test interaction between somatic selection on mutations and CNA in genes

We sought to rigorously measure somatic selection associated with a specific copy number state (neutral, gain or loss), which motivated us to develop a custom statistical genomic methodology. The MutMatch method compares the mutation rate in the coding exons of a gene-of-interest with the baseline mutation rate, estimated directly from the observed mutations in matched loci that are presumably under neutral selection (Fig. 1a). By default, these matched loci are the coding regions of neighboring genes within 0.5 Mb both upstream and downstream; this adjusts for the known domain-scale variability in somatic mutation rates, wherein such domains typically span multiple genes[3,34,35]. Additionally, by using this approach, MutMatch aims to control for confounding effects of larger segmental or arm-level CNA on neutral mutation rates, since the control genes are expected to be similarly affected by the CNA event as the gene-of-interest (central gene) (schematics in Fig. 1a, Supplementary Fig. S1). The MutMatch implementation contains several refinements of the above-mentioned approach using neutral loci from neighboring gene exons. First, controls on neighborhoods are performed by filtering genes in which neutral mutation rates were empirically found to deviate from the expected mutation rate for the neighborhood; see Methods. Second, we address the cases of CNA that are smaller than the neighborhood size by excluding the examples of tumor genomes in which neighboring genes are in a different CNA state than the central gene (described in Methods). Third, importantly, trinucleotide composition differences between the query gene and the neighborhood genes are controlled for by a stratification into 96 mutational contexts, thus preventing confounding by differential activity of mutational signatures (Methods). Furthermore, to ensure that the method operates correctly on sparse data, we tested for biases during estimation on low mutation counts (Supplementary Fig. S2a) and implemented a randomization procedure that adjusts the observed effect sizes and the *p*-values (Supplementary Fig. S2a, d), drawing on a scrambled mutation counts baseline (Methods). MutMatch is comparable to both MutSigCV and dNdScv in the task of identifying genes under positive selection listed in Cancer Gene Census (CGC)[36] (Supplementary Fig. S2c). MutMatch estimates of fitness effects correlate with the covariate-based dNdScv tool, and are fairly robust to reduction in dataset size (see Supplementary Fig. S2e).

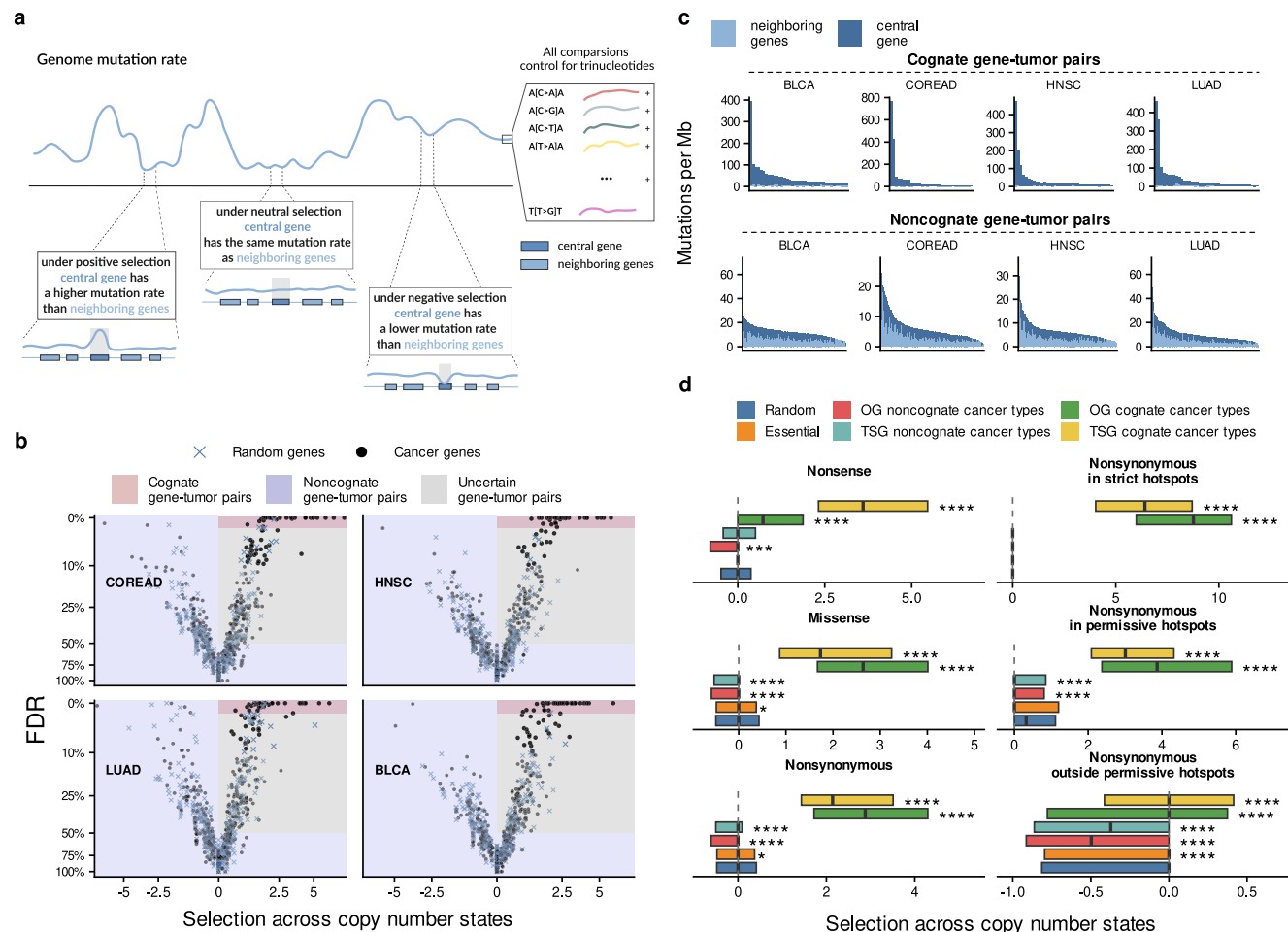

**Fig. 1 | Estimates of somatic selection in various driver gene groups, within and outside the known positively selected hotspot loci. a** Schematic depiction of the MutMatch method to estimate somatic selection by using exonic regions of neighboring genes as a mutation rate baseline, adjusting for trinucleotide composition. A higher relative mutation rate in the gene-of-interest (central gene) compared to its neighbors indicates positive selection, while a lower rate indicates negative selection. A mutation rate in the central gene similar to that of the neighboring genes signifies neutral selection. **b** Annotation of cognate and non-cognate driver gene-tissue pairs is based on the selection estimates ($\log_2$ fold-difference between the mutation rate in the central gene and the baseline rate estimated from neighboring genes) and its significance level (Methods). Random genes are shown for comparison. **c** Relative mutation rate in cancer genes and their corresponding neighboring genes for cognate and noncognate genes in four example cancer types (COREAD, HNSC, BLCA and LUAD). **d** Strength of selection, as $\log_2$ fold-difference in mutation rate in the central gene over neighboring genes. *P*-values are by Mann–Whitney U-test, two-tailed. The nonsynonymous mutations are the missense and the nonsense mutations considered together. The "Random" genes is a control group that excludes known driver genes and essential genes. The center line of box plots denotes medians of data points and the box hinges correspond to the 25th and 75th percentiles. Sample sizes for each group are listed in the Source Data file. Asterisks indicate significance between each gene group and the random genes, for each set of mutations: * for FDR ≤ 5%, ** for FDR ≤ 1%, *** for FDR ≤ 0.1%, **** for FDR ≤ 0.01%. Source data are provided as a Source Data file.

We applied MutMatch to aggregated mutational counts from various WES and WGS data sets, comprising over 17,000 tumor samples in total (see Methods for sources). We analyzed separately each cancer type and every gene to obtain the selection estimates (fitness effects) on mutations across copy number neutral, gain, and loss states.

## Overview of analyses and cancer genomes and gene sets considered

Our first aim was to separately estimate selection on known gain-of-function (GoF) versus loss-of-function (LoF) mutations in coding exons of known oncogenes (OG), tumor suppressor genes (TSG), cell-essential genes (CEG2 set)[37], as well as a negative control set of random genes from which we excluded the OG, TSG and CEG (see Methods for curation of gene sets and data sources). To this end, we considered separately the nonsynonymous mutations in recurrent hotspots (the vast majority are missense), the truncating mutations (nonsense), and finally missense mutations outside of known hotspots.

These three sets serve as representatives of GoF mutations, LoF mutations, and a mixed group of mutations that can be under varying degrees and types of selection, as investigated below.

In the initial analysis, we quantified signatures of selection across different gene sets – random genes, OG, TSG, and CEG2 – and cancer types, integrating across all copy number states. The measure of effect size here is the regression coefficient, which is the log fold-enrichment (expressed as log base 2) in relative mutation rate of a gene, compared to that in the neutral control (neighboring) genes, after adjusting for gene length and trinucleotide composition.

For each cancer gene, we divided the cancer types into two categories (Fig. 1b, c): those where a gene was significantly positively selected ("cognate" cancer types for the gene) and those where it was not strongly selected ("noncognate" cancer types). This classification was performed using an operational definition based on positive selection estimates for mutations – detected either on missense, or on nonsense, or on any nonsynonymous – measured across all copy number states (see Methods). Cognate genes in a cancer type were

defined as having positive selection estimates at an FDR ≤ 2% in that cancer type, while noncognate genes had FDR ≥ 50% or negative selection estimates in that cancer type; genes with positive selection estimates at an FDR 2−50% were considered uncertain and did not count towards either group (Fig. 1b). Our cognate/noncognate definitions broadly matched the cancer type-specificity spectrum from a recent comprehensive catalog derived via the MutPanning method[9] (Supplementary Fig. S2b).

As expected, the nonsense mutations in TSG in cognate cancer types were under a very strong positive selection (median log₂ fold-enrichment in mutation rates across all cognate gene-cancer type pairs = 3.6, Fig. 1d), while the TSG were under weaker positive selection in the non-cognate cancer types (median ≈ 0, however upper quartile $Q_3 = 0.51$ Fig. 1d). Thus there is (modest) selection for nonsense mutations in some of the TSGs in some cancer types that we defined as noncognate, implying that our following analyses that contrast cognate and noncognate genes may be conservatively biased. Moreover, our classification of cognate and noncognate cancer types is independent from gene copy-numbers (the regression model did not include CNA information). As a side note, when considering OGs, there was a modest enrichment of nonsense mutations in some cognate cancer types for the OG (median was not notable at 0.73, however upper quartile $Q_3 = 1.88$, Fig. 1d). This may be explained by either of the two (relatively rarely occurring) scenarios: instances of GoF truncating mutations in OGs[38–40] and/or by genes that may act as either OG or TSG depending on the cancer type[41,42] and that we classed as OGs.

### Signatures of selection in known hotspots identify additional cancer types where an oncogene is causal

Mutations in recurrent hotspots within coding regions of driver genes are highly likely to be causal, while the rest of mutations in cancer genes represent a mix of causal and other mutations in varied proportions. Next, we considered these two groups of mutations separately, estimating selection across all CNA states (Fig. 1d; note that these analyses nonetheless control for possible confounding of CNA on baseline mutation rates) or in diploid state only (Supplementary Fig. S3b). We considered two definitions of hotspot loci: (i) strict hotspots with likely functional effects as defined previously by Trevino[43] and available for $n = 255$ cancer genes under consideration here; (ii) permissive hotspots detected in this study, based on recurrence of mutations and so containing some non-selected hotspots, but are available for a larger set of $n = 961$ genes and are derived from a newer, more comprehensive genomic dataset.

Expectedly, a much stronger positive selection was observed in strict hotspot sites than for all nonsynonymous mutations: for cognate OGs, median 8.77 *versus* 2.88 for selection across CNA states for all nonsynonymous mutation effects (Fig. 1d), and similarly so when considering selection in diploid state only (Supplementary Fig. S3b). Also as expected, there was a more prominent signal in the strict hotspot set than in the permissive hotspot set; the latter did nonetheless exhibit stronger selection than the baseline, supporting its utility (Fig. 1d/Supplementary Fig. S3b).

However, interestingly, the selection in permissive hotspots of OGs in the uncertain cancer types was significantly higher than in the control set of random genes in diploid state (Supplementary Fig. S3b, Mann–Whitney $p = 1.33 \times 10^{-77}$; median of OG hotspots log₂ fold enrichment in uncertain cancer type 1.01, upper quartile $Q_3 = 1.97$, median of random genes ≈ 0, $Q_3 = 0.95$). This suggests that some gene-tissue pairs from this uncertain i.e. tentatively non-selected group of OG-tissue pairs are in fact bona fide drivers in that cancer type. Presumably the selection signal for the OGs, evident within hotspots, was diluted by considering the entire gene locus when classing cognate and noncognate cancer types. Thus, we suggest a simple method to identify driver genes relevant to particular cancer types – including those where they might be in the

"long tail" of rare drivers – by focussing the test for selection on the known hotspot-containing regions within a gene. Examples of thusly identified driver genes are shown in Fig. 2c, and here we highlight the *EGFR* gene in the luminal subtypes of breast cancer (log₂ fold-enrichment = 5.72 across all three copy number states; $p = 2.5 \times 10^{-4}$, FDR = 0.03, Fig. 2f), and *ERBB2* in several cancer types (FDR ≤ 5%, at Fig. 2f). Further examples where tissue-specificity signal was clarified in hotspots include major OGs such as *KRAS* and *NRAS*, which were implicated in melanoma, multiple myeloma, kidney for *KRAS*, or uterus, lung, stomach and other cancers, for *NRAS* (Fig. 2c, Supplementary Fig. S4a); see additional discussion and context for these examples in Supplementary Note 1. Full complement of genes shown in Supplementary Fig. S4b. Overall, 110 OG-cancer type pairs and 118 TSG-cancer type pairs were significant in the hotspot test using strict Trevino or permissive set of hotspots at an FDR ≤ 25% but were not identified in the general analysis (FDR ≥ 50% i.e. "noncognate"). For comparison, the original definition of cognate gene-tumor type pairs yielded 316 OG-cancer type pairs and 479 TSG-cancer type pairs. Thus, focussing on hotspot regions enriches for positive selection signal, allowing to measure the cancer type spectrum of driver genes with better signal-to-noise.

### Prevalent signatures of negative selection on nonsynonymous mutations in oncogenes

A suggestive signal of negative selection was found on nonsense mutations in some OGs: the lower quartile of the nonsense mutation fitness effects was trending towards more negative values in OGs, compared to random genes (log₂ mutation rate fold-enrichment = −0.79 in OG in noncognate cancer types and −0.48 in random genes, $p = 0.15$ by permutation test, Fig. 1d; we note the median was at 0 thus the majority of OG do not show a signal in this test, and neither did essential genes). This is in line with recent reports, where purifying selection for nonsense somatic mutations was shown on OGs in a group analysis[27,29]. Some top-ranked candidates were *PIM1* in bladder cancer (FDR ≈ 1%), *ZFHX4* in colorectal cancer (FDR ≈ 1%), *ERBB2* in prostate, stomach, breast luminal subtype, and bladder cancers (FDR ranging from 4% to 14%) and others (Supplementary Fig. S3a). Similarly, among the known cell-essential genes in the CEG2 set, we do not observe a notable footprint of negative selection on nonsense mutations (neither in the diploid state, nor when considering all three copy number states jointly), consistent with prior reports[1,28]. However, selection on all nonsynonymous mutations were significantly lower for the upper quartile ($Q_3$ of essential genes = 0.38, $Q_3$ of random genes = 0.42, $p ≈ 0$ by permutation test), suggesting a subset of essential genes has detectable negative selection in cancer. We do recover the previously-reported *POLR2A* gene[31] having considerable (albeit nonsignificant) depletion of nonsynonymous mutation rates: log2 fold enrichment of −1.32 in UCEC-POLE cancer type, −0.80 in BRCA-Lum, −0.78 in BRCA and PRAD cancer types and −0.77 in COREAD-POLE cancer type.

Motivated by the trend of depletion in nonsense mutations in OGs, we next asked if the negative selection in OGs is seen in the far more numerous missense mutations. We hypothesized that the pattern of mutations on OGs might be explained not only by positive selection acting on GoF mutations, but also by a negative selection that acts to purge deleterious missense mutations in regions located outside of hotspots, where most GoF mutations are contained. To examine this, we estimated selection in hotspot-free gene regions, using a permissive set of hotspot loci for exclusion (see Methods) thus minimizing the effects of positive selection in the remaining regions (Figs. 1d, 2b).

Indeed, the out-of-hotspot nonsynonymous mutations in OGs in noncognate cancer types were selected negatively (Fig. 1d, lower quartile of distribution for OGs $Q_1 = −0.91$ and median = −0.497) in comparison to the set of random genes (in $Q_1 = −0.81$ and median ≈ 0);

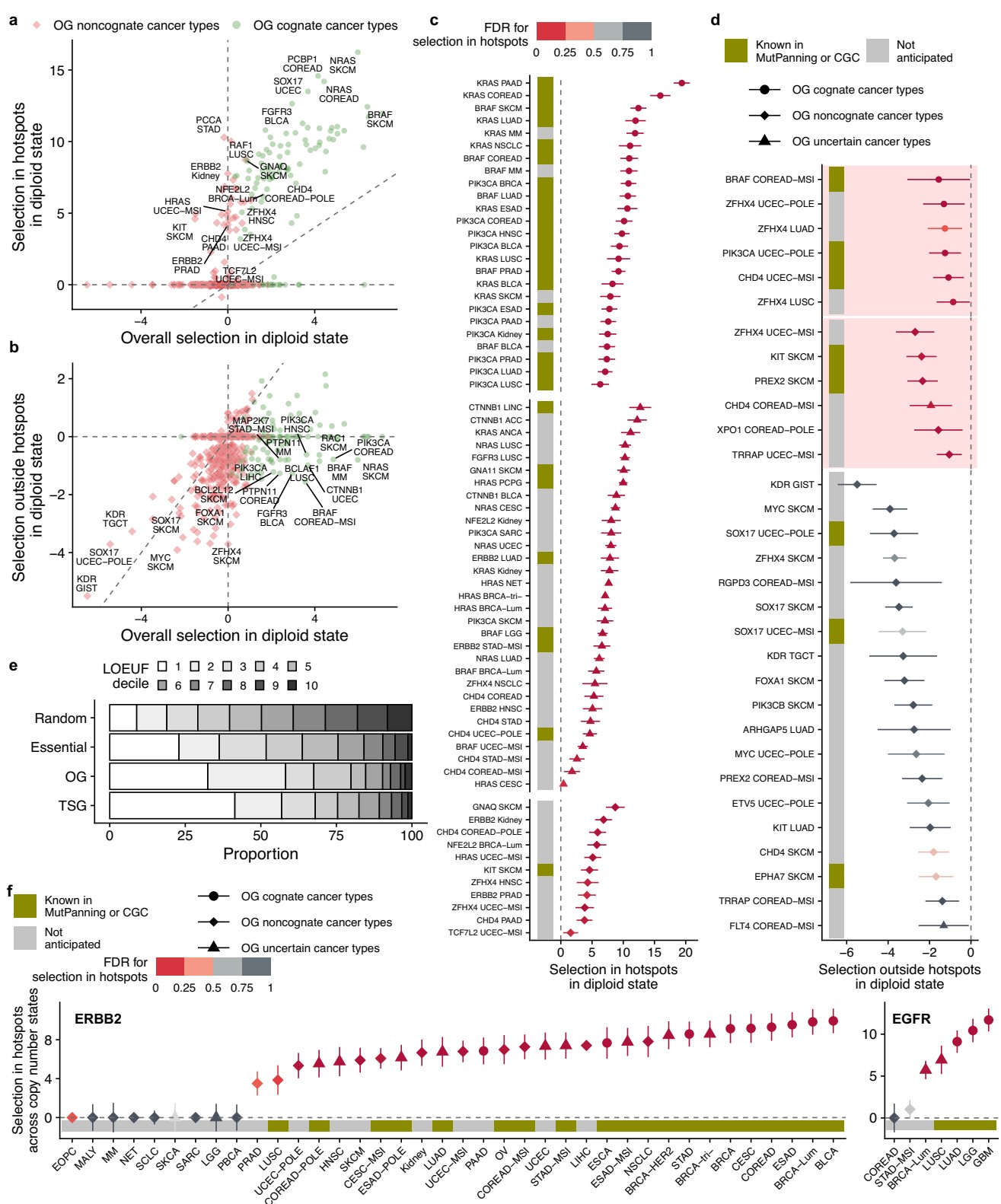

this difference in medians is significant ($p \approx 0$ by Mann–Whitney test). For the OGs in cognate cancer types this signal is not clear, given that median selection effect on non-hotspot mutations is $\approx 0$. However, we do not rule out negative selection could also act on this group, because when using a more stringent threshold to declare a cognate cancer type (FDR = 1%; see Supplementary Fig. S13) indeed the median for cognate OGs is also negative (−0.082 versus median $\approx 0$ for random genes). In other words, we estimate that OGs have -29% fewer

mutations than expected (based on the median of the distribution of noncognate cancer types, and normalizing by the random genes group). This estimate is likely conservative, as some of the subtly positively selected mutations may not have been removed together with the hotspots.

Our analyses indicate that there is negative selection on mutations in oncogenes after excluding the known GoF mutations in the positively selected hotspots (Fig. 2b). Therefore in the remainder of OG

**Fig. 2 | Positive and negative selection on oncogenes across cancer types.**
**a** Positive selection on oncogenes is observed within known hotspot regions (y axis) in some cancer types that are annotated as noncognate (weak overall selection [x axis]). One point corresponds to one gene-tumor type combination. **b** Negative selection in some oncogenes can be observed after removing permissive hotspot loci (y axis). This is seen in the oncogenes that are under net positive selection in a given cancer type (cognate cancer types) and more so in those oncogenes that are not (noncognate). **c** Oncogenes with the strongest positive selection measured within known (strict) hotspots, in which the lower bound of CI > 0. Top, examples of cognate genes as positive controls (showing the top-10 frequently mutated MutPanning genes); middle and bottom, significant (FDR ≤ 25%) oncogenes classified as noncognate (bottom) or uncertain (middle) (shown from set of top-100 frequently mutated genes in MutPanning). The color represents the FDR for selection in hotspots, and the left bar denotes whether a gene-tumor type pair is a

previously known driver in databases (MutPanning or CGC). Error bars show 95% CI. **d** Oncogenes with the strongest negative selection effects (FDR ≤ 50% and the upper CI bound < 0) outside permissive hotspot regions. Error bars show 95% CI. Visualization as in panel (**c**), meaning point color denotes positive selection in hotspots: red points (in panels with pink background) indicate genes with evidence of positive selection within hotspots. Gray points correspond to the genes lacking positive selection in strict hotspots but similarly under negative selection ("noncanonical oncogene addiction"). **e** LOEUF scores distribution measuring population genomic constraint from Loss-of-Function (LoF) variants in the human germline. Cancer genes and cell-essential genes tend to have lower LOEUF scores. **f** Positive selection of point mutations within hotspots in *ERBB2* and *EGFR* genes in cancer types where they are either known drivers (according to MutPanning or CGC; colors in bottom bar) or not currently known. Error bars show 95% CI. Source data are provided as a Source Data file.

coding regions, some sites are essential for the activity of the gene and the oncogene itself is essential for the tumor.

## Noncanonical oncogene addiction, where oncogenes exhibit negative but not positive selection signatures

Next, we asked if the signatures of negative selection seen on nonsynonymous mutations in oncogenes are contained to oncogenes that are positively selected in a given cancer type. We compared signals of selection on OGs inside strict hotspots (from Trevino[43]), presumably under clear positive selection, against hotspot-free areas, outside the permissive set of hotspots, to minimize the signal of positive selection on this remainder of gene coding sequence (Fig. 2a, b; this analysis can consider the $n = 94$ OGs that had available the definition of strict hotspots[43]). The oncogenes formed two clusters as expected: first, where hotspot mutations were not strongly selected, which contained predominantly the genes from noncognate cancer types, and the second group that had a strong selection in hotspots and consisted mainly of cognate cancer types meaning that OGs were overall under positive selection.

Notably, some negatively selected oncogenes were observed in both cognate and in noncognate cancer types, with the signal being clearer for mutations outside of permissive hotspots OGs in noncognate cancer types (median = −0.497 versus median = 0 for control, random genes; Fig. 1d) but is not evident for the median OGs in cognate types (Fig. 1d). This contrasts two hypothetical mechanisms underlying selective pressures that remove deleterious mutations from OGs. The first mechanism acts on driver oncogenes in cognate cancer types and prevents that a LoF mutation would nullify the fitness gain of the GoF mutation if it occurs on the same allele, and/or may incur an additional fitness penalty because of "oncogene addiction", where the cell state after the GoF mutation is such that a withdrawal of its activity is deleterious. The genomic signatures of this mechanism appear less prevalent in our data, although we do not exclude individual examples thereof. The second mechanism of negative selection on OGs highlighted here, however, demonstrates that tumors depend on the oncogene function even in some cases where that specific oncogene is not a driver of tumorigenesis in that particular cancer type. Herein we term this phenomenon "non-canonical oncogene addiction", where a cancer cell is dependent on the function of an oncogene even without its activation by a GoF in that cancer type. For instance *TRRAP, PREX2, SOX17, ZFHX4, MYC, KIT, KDR* and other genes are suggestive examples of genes from this group showing negative selection signals in apparently noncognate cancer types (Fig. 2d, lower panel) with the effect sizes (log$_2$ relative mutation risk depletion compared to baseline) ranging from −5.49 to −2.74 for top-10 hits without positively selected hotspots, with individually significant FDR ≤ 25% for some of them (*KDR* in GIST and TGCT, *ZFHX4* in SKCM, *SOX17* in UCEC-POLE and SKCM).

There is another intriguing implication of these signals of negative selection on OGs in the noncognate cancer types i.e. those in which

overall signal selection was not strongly positive in our analyses. Considered together with the subtle signal of positive selection in hotspots of these putatively noncognate OGs (see above) suggested that a fraction of these cancer types might have been wrongly annotated as noncognate in our (and likely other previous) analyses, because of confounding between positive and negative selection. In other words, negative selection may offset some part of the signal of positive selection on the same oncogene, reducing the power to detect the positive selection, causing a false-negative i.e. failure to identify this as a driver cancer gene in a given tumor type.

In addition to oncogenes, we further considered a set of known cell essential genes (CEG2 set, derived from cell line genetic screening data[37]) in a pan-cancer analysis across 13 cancer types with the largest numbers of mutations in our data. These essential genes are indeed negatively selected in the pan-cancer analysis both in diploid state and across copy number states (for the latter Mann–Whitney $p = 3.85 \times 10^{-7}$, median of random genes = 0.045 and median of CEG2 genes = −0.062). The known essentiality metrics from CRISPR screens in cell lines and from population genomics correlated modestly with our negative selection estimates across the cancer genomes; see specific analyses and discussion in the Supplementary Note 1. The positively selected genes, including OGs and interestingly also TSGs, had higher essentiality metrics (Fig. 2e); discussed in Supplementary Note 1.

## Selection change upon CNAs identifies additional cancer types where a gene acts as a driver

To evaluate how somatic CNAs affect the selection of somatic point mutations in genes, we explicitly considered the copy number in our analyses: CNA gain, CNA loss or neutral (henceforth we call this neutral state "diploid" for simplicity, notwithstanding whole-genome duplication events). This was performed by including the CNA category as a covariate in the regression model, which additionally included an interaction term with this CNA variable; the interaction term would measure conditional selection on the mutation upon CNA (Supplementary Fig. S1).

We performed two analyses based on this setup. First, we estimated conditional selection on point mutations associated with a gene loss (i.e. a change in selection strength that is observed in tumor genomes where a gene copy is deleted, compared to copy-number neutral, Fig. 3a). Second, we estimated selection change on point mutation associated with a gene copy-gain (Fig. 3a). (Of note, these analyses do not consider the temporal order of the mutational versus the CNA events at the gene. Our descriptions of the observations below imply that the selected mutation preceded the CNA, without prejudice to actual ordering of these genetic alterations).

First, we checked for genomic evidence supporting the known two-hit model of TSG inactivation[18,21]. Indeed, selected nonsense mutations and deletions were co-occurring more than expected by chance in TSGs. (Fig. 3a, median effect size for TSG nonsense

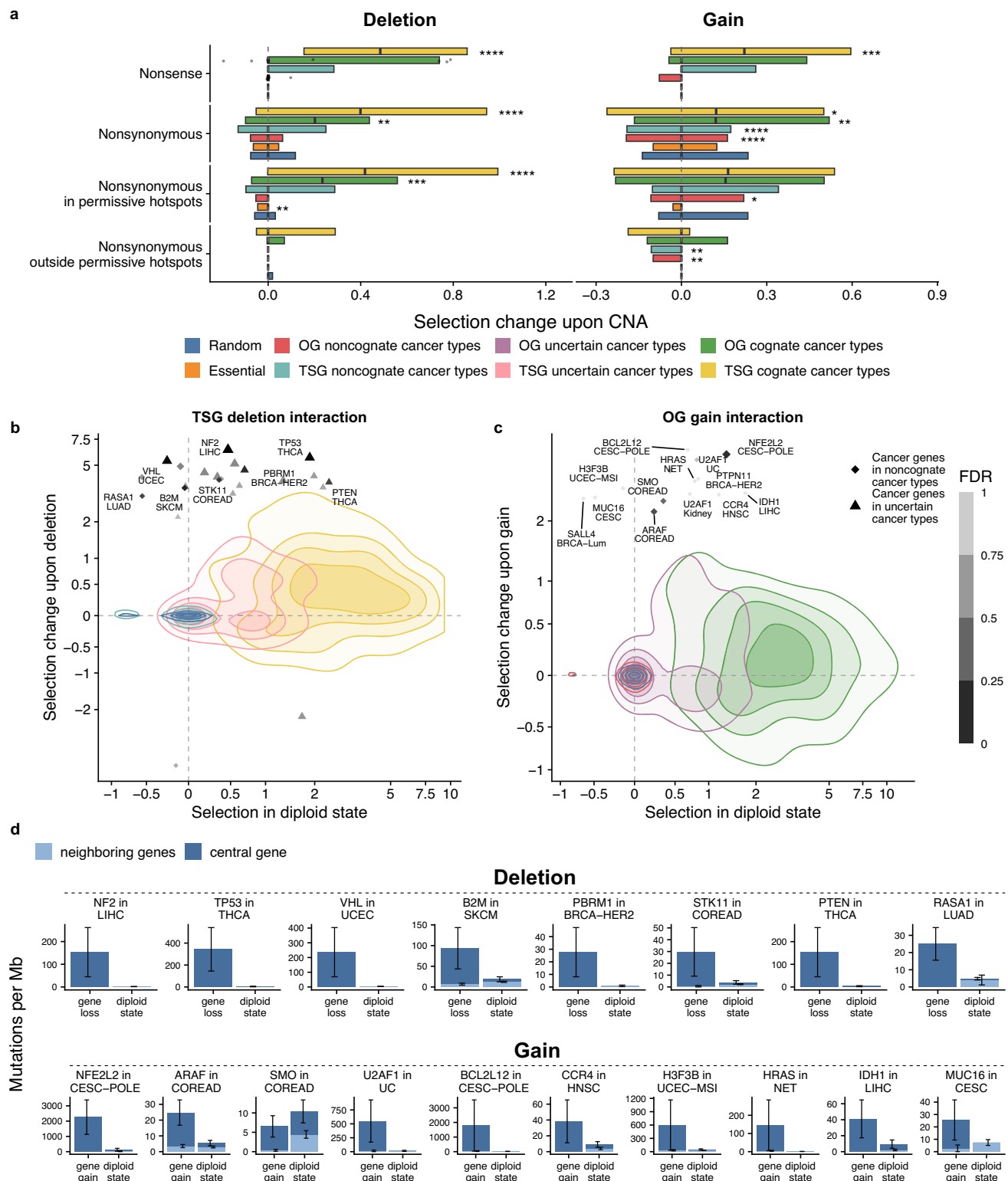

mutations in diploid state = 4.03, while in loci with CNA deletions = 4.52 log₂ fold-enrichment in mutation risk). Interestingly, although the effect was strong for cognate cancer types, a distribution tail of positive interaction between selected nonsynonymous mutations and deletions was also shown for TSGs in apparently non-cognate cancer types (we identified cognate versus non-cognate cancer types using selection estimates across all CNA states). In particular, the TSGs *NF2* in LIHC, *TP53* in THCA, *VHL* in UCEC, *B2M* in SKCM, *PTEN* in THCA, *STK11* in SKCM and COREAD (FDR ≤ 25%) and other noncognate or uncertain

gene-tumor pairs were not showing significant selection in diploid state, but did show selection in tumors with deletion in the same gene (Fig. 3b, d, Supplementary Fig. S6a, b). This suggests that considering somatic mutations specifically in the genome segments harboring CNA deletions — thus focussing on two-hit events — has the potential to identify driver genes for particular cancer types by enriching for causal mutations. The approach would be analogous with a germline variation analysis that monitored co-occurrence of pathogenic germline variants with somatic LOH to identify heritable cancer predisposition

**Fig. 3 | Positive interaction between mutation rates and CNA identifies driver genes for additional cancer types. a** The change of selection strength between tumors where genes are in the copy number neutral ("diploid") state and where a gene copy was lost or gained, estimated using neighboring genes as a mutation rate baseline. The negative control "Random" genes group excludes known driver genes and essential genes. The center line of box plots denotes medians of data points and the box hinges correspond to the 25th and 75th percentiles. Sample sizes for each group are listed in the Source Data file. For boxplots with ≤25 data points, each individual data point is displayed. Asterisks indicate significant difference of the selection strength distribution for a gene group versus the Random genes, for each set of mutations separately: * for FDR ≤ 5%, ** for FDR ≤ 1%, *** for FDR ≤ 0.1%, **** for FDR ≤ 0.01%. **b** Effect sizes of selection change upon CNA deletion for all non-synonymous mutations compared to the selection strength in diploid state. Colors denote gene groups as in panel (**a**). Gene-tumor pairs with FDR ≤ 25% for selection

change upon deletion for top-100 most frequently mutated genes are labeled as individual points, while the rest of the gene-tumor type pairs are shown as a distribution (density plot). The remaining genes are in Supplementary Fig. S6b. Axes are pseudo log-transformed[80]. **c** Effect sizes of selection change upon gain for all nonsynonymous mutations compared to the selection strength in diploid state. Visualization as in panel (**b**). Gene-tumor type pairs with the highest positive selection differential upon CNA gain (for top-200 most frequently mutated genes or genes from CGC) are shown. The remaining genes are in Supplementary Fig. S6c. **d** Examples of identified driver gene-tumor type pairs with altered mutation rate upon CNA loss or gain, compared to the neutral mutation rate in neighboring genes (ordered by FDR). Mutation frequencies (number of mutations per Mb per sample) shown in this plot were not adjusted for trinucleotide composition. Error bars are standard errors of the proportion. Source data are provided as a Source Data file.

genes from genomic data[19]. We estimated the number of gene-cancer type pairs that appeared to be under a positive selection change upon copy number events (here defined as: lower boundary of 95% CI of $\log_2$ mutation rate enrichment interaction term $>Q_3$ of random genes). We found that 80 noncognate and 90 uncertain TSG-tumor type combinations were positively selected in tumor samples with deletion, suggesting they are bona fide TSGs in that tissue. For comparison, 479 TSG-tumor type combinations were selected (cognate) according to our definition across copy number states.

Second, by analogy to the above, we considered the case of OGs and how CNA status might be used to boost the power to identify driver oncogenes pertinent to certain cancer types. The two-hit pattern involving CNA gains was reported to be common in some oncogenes, not only in the well-known cases of highly amplified OGs such as *EGFR*, but also in the case of low-level gains affecting various genes[20,23]. Indeed, also in our data we find that in the CNA gain state, overall positive selection (on all nonsynonymous mutations) in OGs is modestly increased compared to the diploid state (median effect size for cognate OGs = 2.54 upon gain versus 2.42 in diploid; Fig. 3a shows interaction terms from the regression, which are the difference of the two effect sizes). Motivated by this, we hypothesized that by considering the CNA gain state, we may identify some driver genes selected in certain tissues that would otherwise be classified as non-cognate or uncertain i.e. not significantly selected by our standard classification. Indeed at least some of the uncertain OGs do show an increased selection in the permissive hotspots upon CNA gains ($Q_3$ of the interaction term = 0.66, compared to $Q_3$ of random genes set = 0.23, $p = 0$ by permutation test; Fig. 3a; we note the medians are the same, meaning this principle would not apply to the majority of OGs). Indeed in some cases of apparently non-cognate OG-tumor type combinations, there was selection in the copy-number gained state (*NFE2L2* in CESC-POLE, *U2AF1* in UC and other gene-tumor pairs with FDR ≤ 25%) (Fig. 3c, d, Supplementary Fig. S6c). We found that 180 noncognate and 139 uncertain OG-tumor pairs were selected positively upon CNA gain either for all nonsynonymous mutations or for mutations in permissive hotspots (with lower boundary of 95% CI of $\log_2$ mutation rate enrichment $>Q_3$ of random genes), compared to 316 OG-tumor type combinations found selected according to our definition across copy number states.

Finally, we further considered the case of negative selection on mutations in OGs, as mentioned above, here additionally hypothesizing an interaction thereof with CNA state. This is motivated by an analogy with previous reports that essential genes in cancer may show signatures of negative selection specifically in hemizygous segments[1,31]. We found subtle negative conditional selection upon copy gain of nonsense mutations for OGs in noncognate cancer types ($Q_1$ of the interaction term from the model = −0.078 for OGs in non-cognate cancer types compared to -0 for random genes, $p = 0.0049$ by permutation test, Fig. 3a), which might be explained by the dominant-negative character of some instances of truncated oncogenes. We did

not observe a significant depletion in the rate of nonsense mutations in OGs upon deletion (negative conditional selection), compared to the random genes baseline (Fig. 3a), potentially due to the small sample size i.e. rarity of nonsense mutations in oncogenes.

## Positive selection on point mutations is increased by deletions in OGs and gains in TSG

In addition to the known case of two-hit interactions between mutations and CNA gains in OGs, and mutations in CNA losses in TSGs, we hypothesized that the opposite case may also be selected. Recently it was reported that allele imbalance of mutations at OGs can result from deletions[20] and we thus turned to examine differential selection on point mutations in OGs upon deletions. For symmetry, we additionally checked the case of TSGs and amplifications.

Both in TSGs as in OGs, selection on all nonsynonymous mutations was stronger in samples with a CNA loss of the same cancer driver gene, compared to a control set of genes (Fig. 3a) (effect size in the following text refers to an interaction term from the model, i.e. difference of $\log_2$ mutation rate fold-enrichment, comparing the diploid state *versus* the CNA loss state). In particular, the differential selection effect size for random genes has the distribution upper quartile $Q_3 = 0.12$ and median = 0, while for cognate OGs $Q_3 = 0.44$ and median = 0.20 and for cognate TSGs $Q_3 = 0.94$ and median = 0.40 (Fig. 3a). Therefore the two-hit deletion effect in potentiating positive selection in some OGs ($Q_3$: top 25% OG-cancer type combinations) is higher than the two-hit deletion effect on the median of cognate TSGs (Figs. 3a and 4b). Next, as a validation we hypothesized that this two-hit OG deletion effect should become more pronounced in the case of mutations in hotspots, which are in OGs very strongly enriched with positively selected mutations. Indeed, supporting our observations, by considering the permissive hotspots $Q_3 = 0.56$ for OGs and $Q_3 = 0.99$ for TSGs, while for random genes the $Q_3$ is expectedly much lower ($Q_3 = 0.031$). Therefore, deletions in OGs − similarly as was known for TSGs − can in fact commonly increase tumor fitness via two-hit events.

Next, we turn to consider how the selection of mutations in OG and TSG changes in tumor samples with a gene gain (Fig. 3a; effect sizes shown are differences of $\log_2$ mutation rate fold-enrichment, comparing diploid state and CNA gain state). We observed that driver missense mutations in OGs had, expectedly, an increased selection in samples with CNA gain in OGs[20,23], compared to non-selected gene-tissue pairs (meaning: either the control set of random genes, or cancer genes in noncognate cancer types). However we observed that TSGs are also under increased selection upon gains, considering nonsense and missense mutations as potential drivers. In particular, the differential selection effect sizes ($\log_2$ fold increase in mutation rates upon gain, obtained from the interaction term in regression) for all nonsynonymous mutations was median = 0.12 and $Q_3 = 0.50$ in cognate TSGs. For comparison, in noncognate TSG median = 0 and $Q_3 = 0.17$ and in control random genes was median = 0 and $Q_3 = 0.23$ (Fig. 3a). Remarkably, this difference in selection on TSGs upon CNA

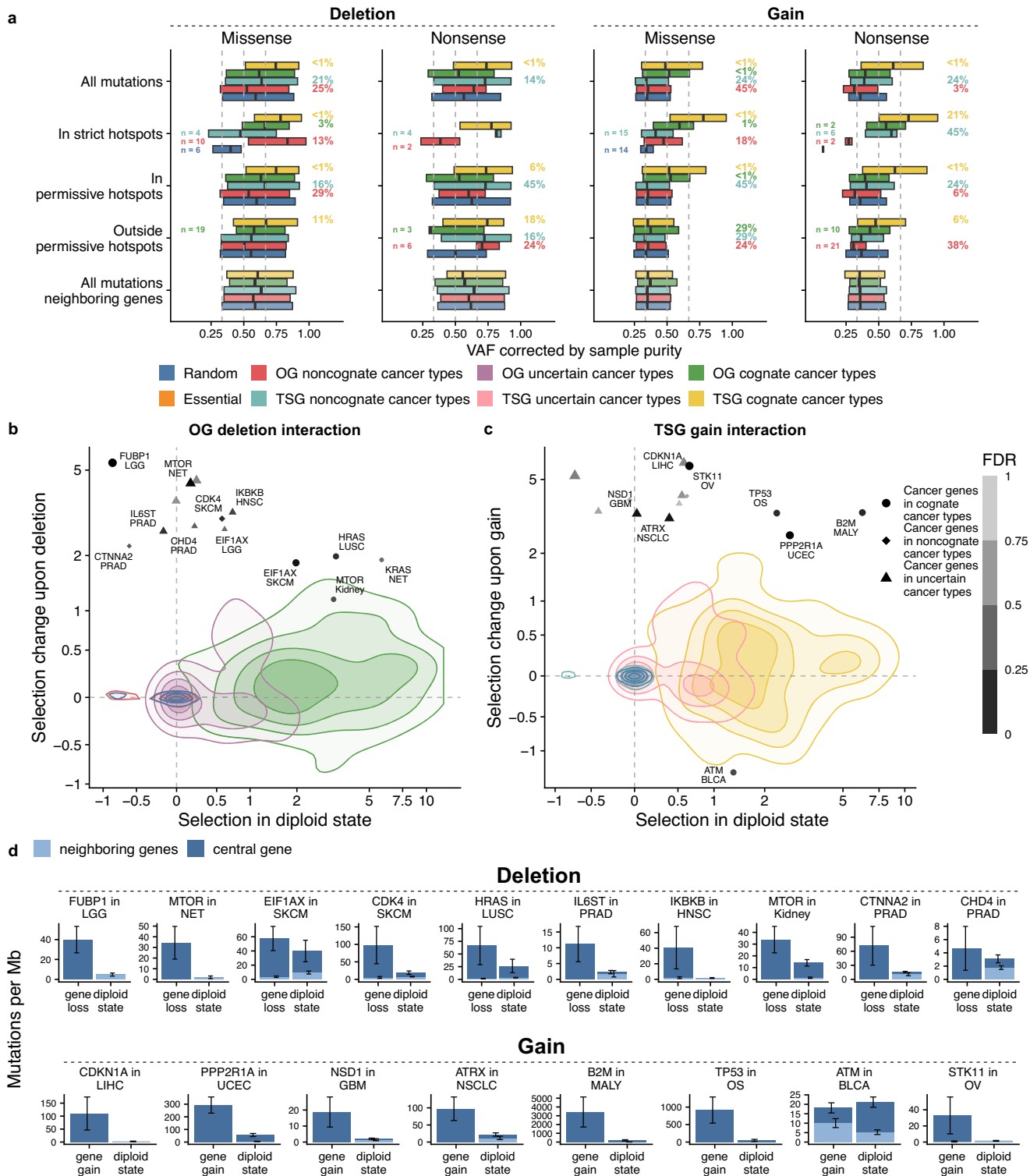

gains was similar to the known two-hit effect on selection on OGs upon CNA gain (median = 0.12, Q₃ = 0.52).

Removal of subclonal mutations, of low-purity tumor samples, and separation by cohort did not qualitatively change this TSG-CNA gain association, nor did it abolish the OG-CNA loss association we noted above (Supplementary Fig. S14a, b). In specific, the median effect size for OG-loss interaction was 0.20 in the original analysis (FDR = 0.069%), 0.10 (FDR = 14.9%) after removing low-purity tumor samples and 0.10 (FDR = 0.96%) after removing subclonal mutations. For TSG-gain interactions the median effect size was 0.12 in the original

analysis (FDR = 2%), 0.12 (FDR = 0.0657%) after removing low-purity samples and 0.12 (FDR = 0.031%) after removing subclonal mutations. Some of the tumor suppressor genes with an increase of selection upon CNA gain included *CDKN1A* in LIHC, *PPP2R1A* in UCEC, *ATRX* in NSCLC, *NSD1* in GBM, *B2M* in MALY, *TP53* in OS, and others (FDR ≤ 25%, listed in Fig. 4c, Supplementary Fig. S7b).

Overall, there is increased selection of hotspot, likely gain-of-function mutations in OGs bearing CNA deletions, suggesting that for many OGs the presence of the *wild-type* allele is detrimental for the tumor fitness. Additionally, we find additional positive selection

**Fig. 4 | CNA gains associate with positive selection in TSGs, and CNA deletions associate with positive selection in OGs. a** A supporting analysis using variant allele frequencies (VAFs) for somatic mutations in tumor samples with a somatic CNA loss or gain. One data point corresponds to a purity-adjusted VAF of one missense or nonsense mutation. The center line of box plots denotes medians of data points and the box hinges correspond to the 25th and 75th percentiles. Sample sizes for each group are listed in the Source Data file. Groups with a low number of mutations ($n \leq 25$) are indicated. Adjusted $p$-values (two-sided Mann–Whitney U-test for the difference between gene groups and a set of random genes) are shown for FDR ≤ 25%. The "Random" genes group serves as a negative control and excludes known driver genes and essential genes. **b** Effect sizes of selection change for genes upon CNA deletion, compared to the selection strength in copy number neutral ("diploid") state. Colors denote gene groups as in (**a**). Gene-tumor type pairs with the highest positive selection change upon CNA gain (for top-200 most

frequently mutated genes or genes from CGC) are highlighted. The remaining genes are in Supplementary Fig. S7a. Axes are pseudo log-transformed[80]. **c** Effect sizes of selection change for genes upon CNA gain compared to the selection strength in diploid state. Visualization as in panel (**b**). Gene-tumor pairs with the lowest FDR of selection change upon deletion (FDR ≤ 25%) or highest effect size are labeled as individual points (for top-100 most frequently mutated genes), while the rest of the gene-tumor type pairs are shown as a distribution (density plot). The remaining genes are in Supplementary Fig. S7b. **d** Examples of identified gene-tumor type pairs with altered mutation rate upon CNA loss or gain, compared to the neutral mutation rate in neighboring genes (ordered by FDR). Mutation frequencies (number of mutations per Mb per sample) shown in this plot were not adjusted for trinucleotide composition. Error bars are standard errors of the proportion. Source data are provided as a Source Data file.

---

pressure on TSG mutations in tumors with CNA gains in the same gene, indicating that dominant-negative mechanisms may be common among TSG point mutations.

## Variant allele frequency distributions shifted by copy-number losses reflect selection in TSGs and OGs

To substantiate that CNA deletions in OGs and also gains on TSGs can increase selection of concomitant mutations, we further considered which allele is affected by the CNA. We hypothesized that mutant allele variant allele frequencies (VAFs) of the driver mutations (here, the assumption is that most mutations in driver gene groups are drivers) will change unidirectionally in samples with either CNA losses or CNA gains, i.e. allelic imbalance will arise[19–21]. As a baseline, this is here compared to the random genes set, in which mutations are not selected (i.e. passengers) and so each CNA will presumably affect the mutant allele and the *wild-type* allele with similar frequencies. We corrected VAF estimates for purity of tumor samples (Methods), while to account for ploidy effects on VAF, the comparison to a control group of random genes provides a baseline.

In this test, higher VAFs of mutant alleles in sets of cancer genes would be consistent with a stronger positive selection. As a positive control, we considered the known two-hit TSG mechanism: in tumors with a CNA loss in a TSG, the allele frequencies of nonsynonymous mutations in the TSG in cognate cancer types were significantly higher than those in random genes (Mann–Whitney test $p = 3 \times 10^{-21}$, with the median VAF for TSG = 0.75 and random genes = 0.59) (Fig. 4a). In other words, the copy number loss preferentially affected the wild-type allele, resulting in loss-of-heterozygosity (LOH) of the TSG. For the individual TSGs *TP53*, *SPEN*, *CDKN2A*, *ATRX*, *RBM10*, and *AMER1* it was possible to show a significant increase in VAFs in CNA loss states, compared to the set of random genes as a baseline, at FDR ≤ 5% (12 genes significant in total across all cognate cancer types at FDR ≤ 25%, see full list in Supplementary Fig. S8a).

Next, to support our finding of selective effects of the CNA deletions not only in TSG, but also in OGs, we considered VAFs of nonsynonymous mutations therein (Supplementary Fig. S8b). Here, the VAFs for missense mutations upon deletion were higher than those of random genes for the *AR* gene (at FDR < 1%) and with non-significant trends in various others including e.g. notable OGs *BRAF*, *NRAS*, and *KRAS* (Supplementary Fig. S8b) Interestingly however, some genes showed decreased VAFs of missense mutations, suggesting that CNA deletions are not associated with positive selection in all oncogenes, but the effect may be gene or tissue-specific.

To reduce signals of negative selection in OGs confounding this analysis, we analyzed VAFs of missense mutations separately for hotspots and non-hotspot sites. Supporting an increased selection in OGs upon deletion we observed an increased VAFs of oncogenes in cognate cancer types compared to those of random genes in strict hotspots (FDR = 3%, $p = 0.006$). This was also evident in VAFs of mutations of OGs in noncognate cancer types (FDR = 13%, $p = 0.04$). In samples with

gene CNA loss, median VAF for OGs in cognate cancer types was 0.61 and 0.66, for non-hotspot and strict hotspot mutations, respectively. This OG effect of CNA gene loss exceeds that of the known two-hit mechanism linking OGs and CNA gene gain, where the median of the VAF distribution for OGs in cognate cancer types was 0.51 and 0.60 for non-hotspot and strict hotspot mutations, respectively. In contrast, VAFs of mutations outside of strict hotspots of OGs in noncognate cancer types trended towards a decrease compared to the random genes baseline (FDR = 29% for mutations in permissive hotspots and FDR = 68% for mutations outside of permissive hotspots), consistent with negative selection on OGs outside of hotspots. Considered jointly, this supports that when an OG bears a mutation and a deletion occurs, the wild-type allele tends to be the one deleted rather than the mutant OG allele, signaling selection in deleted OGs. Increase in VAF of the mutant allele supports that the selected CNA event occurs after the mutation, although we do not rule out individual cases of the converse ordering.

## Copy-number associated conditional selection on TSGs and OGs confirmed in VAF analysis

Next, we turn to examine driver mutation VAFs in tumor samples with a CNA gain in OGs and TSGs. In accordance with reports of two-hit events involving gains on OGs[20,23], the frequencies of missense mutations in OGs showed higher VAFs compared to random genes (Mann–Whitney test test $p = 2.57 \times 10^{-66}$, with the median VAF for the OGs = 0.51 and random genes = 0.35), suggesting that the mutant allele is the one that gets duplicated more often (Fig. 4a). For example, oncogenes *NRAS*, *KRAS*, *BRAF*, *PIK3CA*, *CACNA1D*, *STAT3* and other genes individually had significantly higher VAF in all nonsynonymous mutations, when compared to a random genes baseline at FDR ≤ 5%; total number of significant oncogenes with an increased VAFs in samples with CNA gain (across all cognate cancer types) reached 24 at FDR ≤ 25% (see Supplementary Fig. S9b).

On the contrary, nonsense mutations in OGs in noncognate cancer types (with the lower quartile for VAF for the group of random genes $Q_1 = 0.27$ and OGs = 0.23, $p = 0.02$) had lower allele frequencies upon CNA gain, compared to random genes. This indicates there may be a negative selection against the increasing dosage of truncated OG proteins, supporting that some truncated OGs may be deleterious.

Next, we consider TSG, for which our analyses of conditional selection (see above) suggested that there is, interestingly, overall increased positive selection on point mutations upon gains, thus extending the two-hit TSG paradigm. This is indeed supported also in the VAF analysis: potentially inactivating mutations (both missense and nonsense) had higher VAFs in tumor suppressor genes upon copy number gain, compared to a random genes baseline (Fig. 4a). This implies that, when CNA gains occur in TSG, they preferentially amplify the mutant TSG alleles, bolstering the findings above about increased selection upon point mutations in TSGs bearing CNA gains. The increased mutation VAFs imply that at least in some cases of selected

events the CNA occurred after the mutation, although the converse ordering of events is not ruled out.

Alternatively or additionally, there may be a negative selection against amplification of wild-type alleles in TSGs bearing mutations. Both of these scenarios are compatible with dominant-negative activity of many mutations in TSGs. Further, our data suggests a model where the dominant-negative mutations in these TSG undergoing CNA gains are only partially dominant, and thus increasing their proportion versus the *wild-type* allele benefits fitness of cancer cells. Some examples of genes associated with CNA gains in this test include *TP53, PTEN, CDKN2A, KEAP1, NSD1, ARHGAP35, RBM10, MGA, NF1* and *FBXW7* genes, which had significantly higher values of mutant VAFs at FDR ≤ 5% (Supplementary Figs. S9a and 30 genes in total at FDR ≤ 25%), suggesting an (incompletely) dominant-negative GoF character of some mutations in those tumor suppressor genes.

## Replicating associations between selection on point mutations and CNA events in an independent data set

To further verify the positive selection change upon deletion of a gene or copy number gain in TSGs and OGs, we estimated CNA-conditional selection in an independent dataset with 89,243 tumor samples from the Genomics Evidence Neoplasia Information Exchange (GENIE) consortium[44,45], consisting of panel sequencing data, where we examined 170 genes therein. Here, to estimate a baseline mutation rate neighboring genes were not available, however our MutMatch method used the low-impact nonsynonymous mutations within cancer genes (defined as with CADD[46] score <20; Supplementary Fig. S10). Additionally, since the sequenced genes are all driver genes, a "random genes" control set is not available, so we employed a one-sample Wilcoxon test comparing selection estimates against zero. Further, the cancer genes in non-cognate cancer types were used for comparisons.

In this replication data set, the selection change upon CNA gain was positive for the TSGs as with the OGs, substantiating the overall findings in the discovery cohort. The effect sizes of TSG conditional selection in the gained state were comparable to those of OGs (median for TSGs = 0.126 and OGs = 0.121). Regarding the selection change upon CNA deletion for OGs in cognate cancer types, although this was not shifted from zero in this panel-sequencing data set, we did observe a difference between the cognate OGs and the noncognate OGs used as a control (Supplementary Fig. S10), in support of the notion that deletions in OGs act to potentiate selection on oncogenic mutations therein.

## Main trends in genomic signatures of CNA-associated selection on somatic mutations across driver genes and tissues

Our analysis of >17,000 cancer genomes/exomes estimated selection on different types of point mutations − nonsense, missense and all nonsynonymous, either in hotspots or outside of hotspots − in different copy number states, for every cancer gene and every cancer type. This represents a rich dataset for deriving an unbiased, systematic classification of cancer genes, by their mechanism of activation or inactivation across tissues. Previous classifications tended to focus on distinguishing TSG from OG, and two-hit from one-hit driver genes[11,23,47], while here we consider a more comprehensive, multi-dimensional classification for each gene simultaneously considering GoF versus LoF mutations, CNA states, tissue specificity, and interactions of these factors.

To outline general trends in this data, we performed a principal component (PC) analysis on all the estimates of selection for each driver gene (as above, the effect sizes are $\log_2$ fold-enrichments of mutations in a cancer gene, normalized by the neutral rates estimates). The aim was to measure covariation between selection on different genetic alterations across cancer gene-cancer type combinations, thus inferring how common are various two-hit or one-hit mechanisms in groups of genes (Fig. 5c). A visualization of the major PCs

(Supplementary Fig. S11) suggests that cancer genes do not form distinct clusters by the mechanism of (in)activation via mutations and CNA (Fig. 5a, b). Instead, we observe a continuous spectrum of cancer-driving potential realized via multiple mechanisms − as summarized by each PC − which are observed in different relative frequencies on different genes and tissues (Fig. 5a, b).

The most prominent trend reflects simply the overall intensity of positive selection across all copy number states and mutation types (PC1, 14.4% variance explained): some genes are stronger drivers and others are weaker drivers, which is quantified by PC1 (Supplementary Fig. S11a, b). The following two PCs describe interaction of selection on mutations with the CNA state − PC2 for conditional selection associated with gene loss, and PC3 with gene gain − and jointly can explain 25.1% of systematic variance in the selection effects across cancer genes (Supplementary Fig. S11a, b). It is notable that these two PCs describing CNA-specific selection effects jointly explain more variance than the general selection PC1, suggesting that cancer driver genes vary more in their CNA-specific selection than in the overall selection. These PC2 and PC3 suggests that changes in driver potential across CNA states are substantial for many cancer genes. Finally, the PC4 accounted for the differential selection of nonsense versus missense mutation types, thus separating TSG from OG. This captures a property commonly used to classify cancer genes − presence of obvious LoF mutations − however explains 7.6% variance, considerably less than the CNA-interaction PC2 and PC3. Interestingly, PC4 also correlated with changes of selection upon CNA gains and, to a smaller extent, with gene loss, in agreement with the two-hit selection effects of both deletions and gains in TSGs. Overall, this PC analysis suggests that genetic alterations causing mutant allele imbalance in driver genes (as per PC2 and PC3) can be very informative features for categorizing cancer genes. Therefore we proceed to delineate gene categories based on this data set.

## A data-driven, comprehensive mechanistic classification of cancer driver genes based on CNA and mutations

To derive a categorization of driver genes, we considered regularized estimates of selection effects (Methods) to derive PCs; this means the more noisy estimates of selection are brought towards 0, while the more confident ones remain at higher absolute values. Next, we applied k-medoids clustering to the cancer genes described by the resulting 9 factors (PCs to which rotation was applied for sparseness, see Methods). We focused on the 25 cancer types with the largest number of mutations available, and additionally only focused on those gene-tissue pairs where selection effect sizes were confidently non-zero (after regularization, as above).

The driver gene-tissue pairs were classified into 9 clusters by mechanisms of genetic alterations (Fig. 6b, Supplementary Fig. S12a), the clusters can be further organized into several overarching groups.

The first group consists of genes that predominantly act via the one-hit mechanisms (as per Factor 1 and Factor 2, Fig. 6a). These are the Cluster 1 (32.5% gene-tissue pairs) representing gene-tumor type pairs with prevalent selection on missense hotspot mutations and enriched with cognate OGs (and the "uncertain" TSGs that are likely selected). Additionally, there is the Cluster 2 (10.3% gene-tissue pairs) representing genes with positive selection on nonsense mutations, with a strong enrichment of cognate TSGs. Importantly it is not excluded that some of these genes may additionally act via two-hit mechanisms (with elevated scores in Factors 5-9, see Fig. 6b, Supplementary Fig. S12b) however they are classified in this group because of the one-hit effect being widespread and/or strong.

The second group consists of clusters of two-hit genes, with emphasis on CNA losses (quantified by the Factors 4, 5 and 6). This would encompass Cluster 4 (6.4% gene-tissue pairs), Cluster 5 (2.0% gene-tissue pairs) and Cluster 6 (1.9% gene-tissue pairs). Cluster 4 and

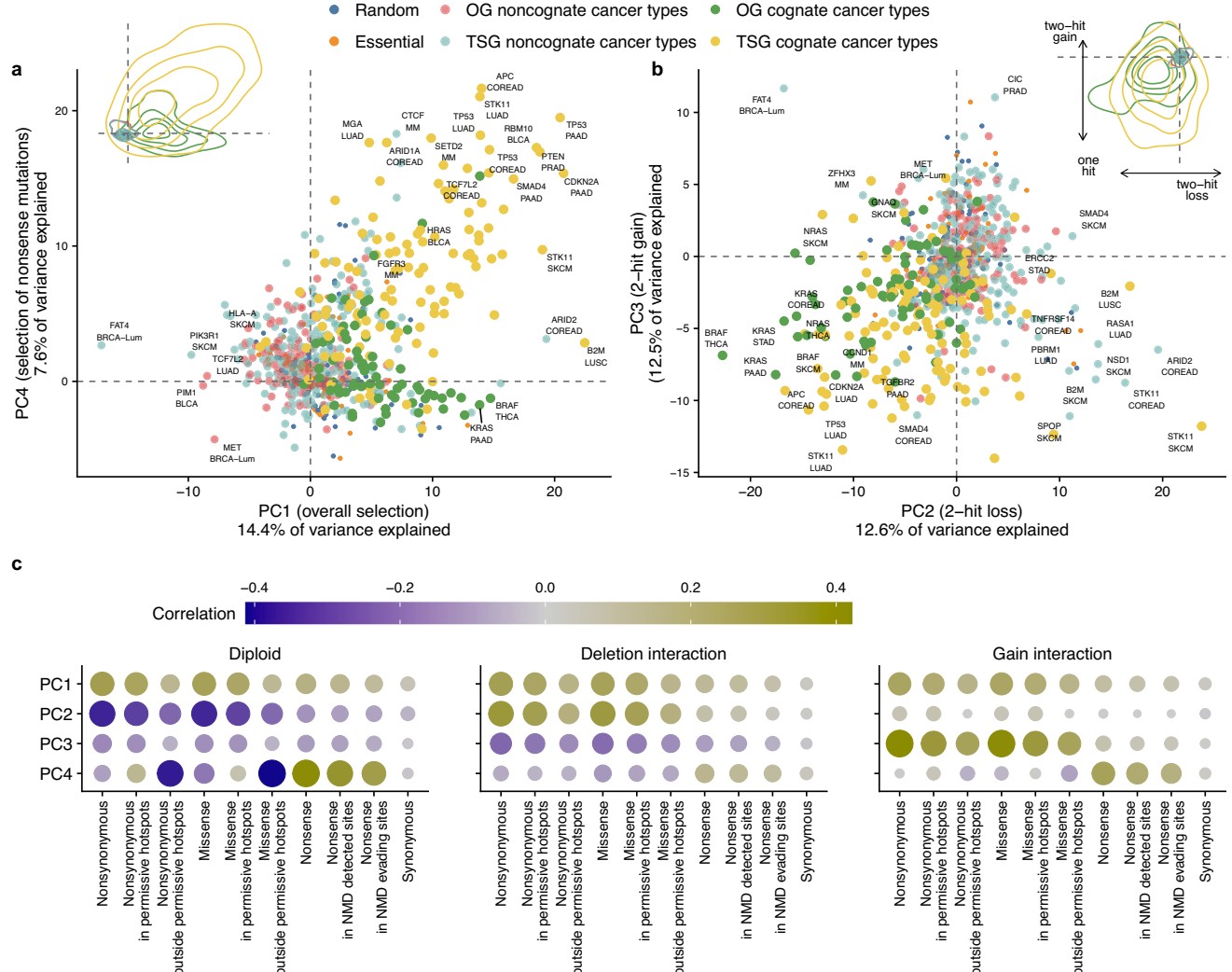

**Fig. 5 | Main trends in variation in somatic selection effects across cancer genes, cancer types and mutation and CNA states.** Principal components (PCs) 1 to 4 are shown for cognate and noncognate cancer genes and for a subset of random and essential genes in cancer types COREAD, LUAD, BRCA-Lum, PRAD, LUSC, BLCA, MM, LUSC, SKCM, PAAD, STAD, THYM, and THCA. **a**, **b** Gene groups in the space defined by the four first PCs. Genes with the highest absolute scores are labeled.

Scatter plots show the 150 most frequently mutated genes in cancer, and a subset of random and essential genes. Density plots (inlaid) are created using the complete set of cancer genes. **c** Correlations of the PCs 1-4 with the selection effects for different mutation types, used to assign an interpretation to the PCs (see axis labels in panels **a**, **b**). Source data are provided as a Source Data file.

Cluster 6 are characterized by positive selection of hotspot and non-hotspot, respectively, missense mutations in the deleted state (Fig. 6b). In this group, the Cluster 5 is the canonical two-hit TSG mechanism, enriched with cognate TSGs with a strong selection of nonsense mutations in the deleted state and subtle selection of mutations in the diploid state.

The third group also consists of two-hit genes, but here with emphasis on CNA gains (Cluster 7, Cluster 8 and Cluster 9). The Cluster 7 represents two-hit gain genes with prevalent selection on hotspot missense mutations, and weak negative selection of non-hotspot missense mutations (13.7% gene-tissue pairs). This Cluster 9 (5.7% gene-tissue pairs) underscores our finding of selection of CNA gains in tumor suppressor genes and contains two-hit gain genes with positively selected nonsense mutations. Similarly to the Cluster 7, Cluster 8 (8.2% gene-tissue pairs) exhibits two-hit gain selection on missense non-hotspot mutations (Fig. 6b). Finally, the fourth group is made up exclusively of Cluster 3 (19.3% gene-tissue pairs), which is a mixed category consisting of noncognate gene-tissue pairs, both TSGs and OGs. We observed decreased scores of Factor 3 in this cluster (Fig. 6b, Supplementary Fig. S12b), which is correlated with negative selection

of non-hotspot missense mutations, indicating that this cluster contains genes that may be essential for tumors via a hypothetical "non-canonical addiction" mechanism.

Interestingly, the membership of driver genes in the OG-like clusters (such as Cluster 1 and Cluster 7) versus TSG-like clusters (such as Clusters 2, Cluster 5 and Cluster 9) depends on the cancer type (Fig. 6c), suggesting that it is not uncommon that a gene can have either an OG-like or a TSG-like mechanism of selection in different tissues. It also suggests that OGs tend to be less variable in their selection mechanisms across tissues, compared to TSGs.

Another way of classifying these 9 driver gene clusters is by one-hit genes (Clusters 1 and 2) versus two-hit genes (Clusters 5 to 9). Considering the genes within these clusters supports that the majority of the genes may switch between one- and two-hit mechanisms between different cancer types[23], while the minority of the cancer genes (6.1%) being predominantly one-hit genes, or similarly 8.1% being predominantly two-hit genes (Fig. 6d). In summary, our analysis delineated 9 cancer gene archetypes with different selection pressures on different mutation types and copy number states, and different spectrum of affected cancer types.

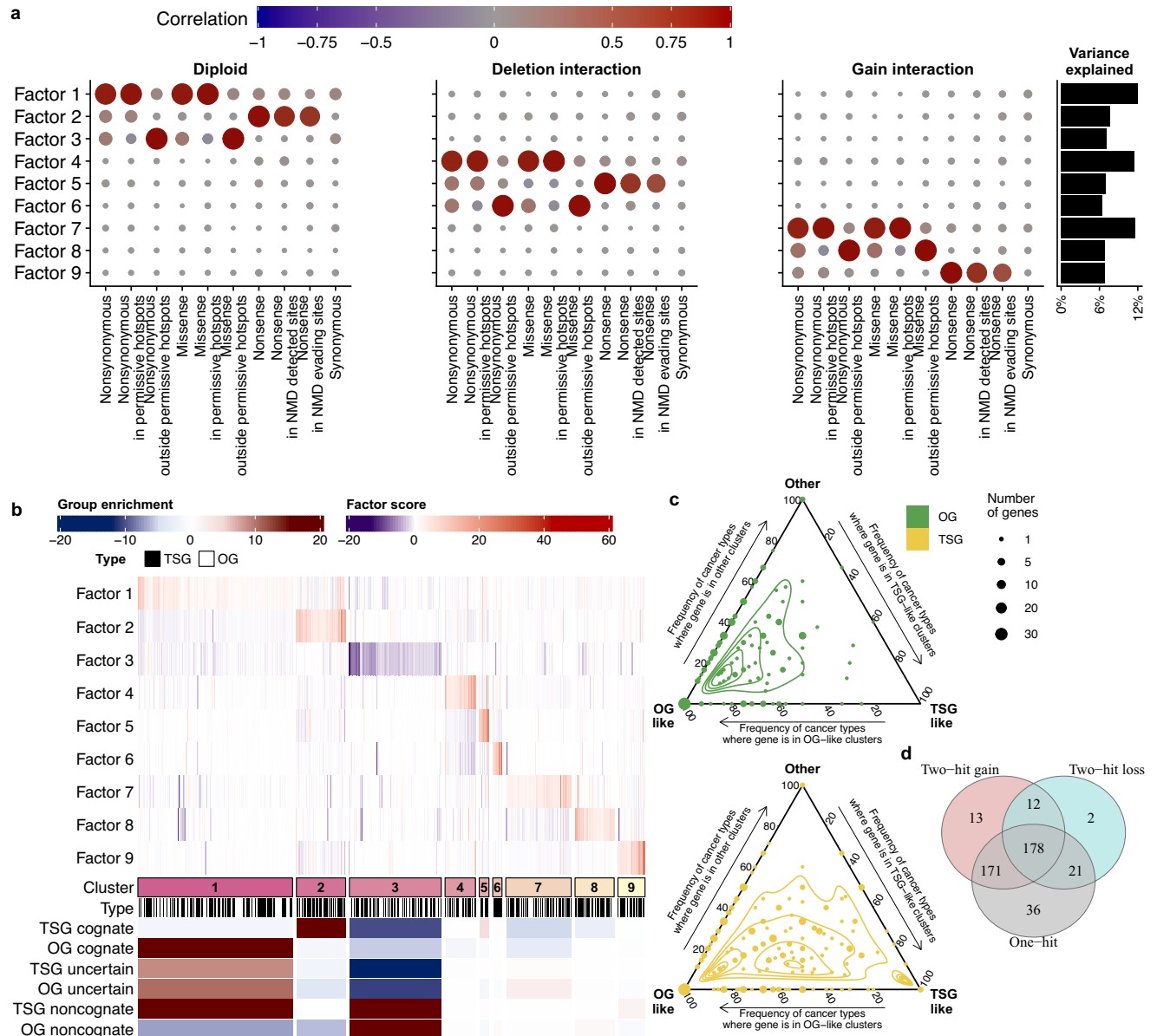

**Fig. 6 | Deriving cancer gene archetypes from a comprehensive genomic analysis of driver mutations and CNA. a** Interpretation of the factors (rotated PCs derived from selection effect estimates) by correlation to the source data with selection effects upon driver genes, across mutation types and interactions with copy number states. **b** Clustering of driver genes across 25 most numerous cancer types (excluding MSI and ultramutated tumors). Enrichments of clusters with gene group types (lower part of plot) are shown with signed logarithms of *p*-values (two-sided) derived from chi-squared residuals; deep blue corresponds to depletion, and deep red to enrichment. Within each cluster, the gene-tumor type pairs (columns in heatmap) are arranged by hierarchical clustering. **c** Frequency of cancer types where a gene (OGs, above; TSGs, below) is selected via an OG-like pattern of mutations/CNA (increases to the left, Clusters 1 and 7), via a TSG-like pattern (increases to the bottom, Clusters 2, 5 and 9) or via another pattern (rest of clusters, excluding the weakly negatively selected Cluster 3). Point size shows the number of genes at a certain spot in the ternary plot. **d** Cancer genes can switch between one-hit and two-hit patterns depending on the cancer type; numbers show count of gene-cancer type pairs which are classed into one-hit pattern (Clusters 1 and 2), two-hit loss pattern (Clusters 4, 5, 6) or two-hit gain pattern (Clusters 7, 8, 9). Source data are provided as a Source Data file.

## Discussion

The prevalence and the strength of the selection acting to purge deleterious somatic mutations have been unclear, however an interaction thereof with the copy-number state of genes was suggested[1,28,31]. Thus we considered both somatic mutations and CNA jointly to interrogate signatures of negative selection acting on essential genes and also on cancer genes. We suggest that point mutations in cancer genomes can be both positively and negatively selected in oncogenes, in which we estimate ~30% of point mutations may be removed. Previous studies suggested negative selection against specifically nonsense mutations in OGs[27,29] (possibly, a genomic signature of oncogene

addiction[48]), consistent with the function of OG being vital for the survival and proliferation of cancer cells. Quantifying this negative selection may have implications to identifying cancer vulnerabilities. Such analyses require nuance, in light of our data suggesting that OGs may be negatively selected (and so constitute therapeutic targets) particularly in some cancer types where they are not commonly bearing driver mutations. Based on genomic signatures, we have tentatively named this prediction "non-canonical oncogene addiction", however experimental validation of this awaits future work. A limitation of our statistical genomic analysis is that it is not powered to detect negative selection in individual OGs, thereby identifying

particular candidate vulnerabilities; larger datasets would be expected to remedy this.

With respect to essential genes[37], our methodology suggests that mutations in them were depleted compared to a mutation rate baseline (here derived from select neighboring genes and by further matching by trinucleotide composition). Consistent with previous work[1,31], the effect size of this negative selection is relatively small — we estimate it removes approximately ~10% mutations from a set of known essential genes — therefore it can be detected only in very large datasets, such as our pan-cancer analysis with ~18,000 somatic genomes. Moreover, our method drawing on a mutation rate baseline that controlled for effects of concurrent CNA on neutral mutation rates may have facilitated detecting the subtle signal of negative selection. Identifying individual genes under negative selection, likely cancer dependencies, will require substantially larger sample sizes.

Our analyses suggest that the net distribution of mutations in OGs results from simultaneous activity of positive selection that increases the frequency of cancer-driving mutations, and the more subtle purifying selection that removes the deleterious mutations. The canceling out of the two signals can lead to underestimation of the positive selection strength. We suggest a trick to circumvent this issue and increase power to identify the less frequently-mutated "long-tail" driver OGs in certain cancer types, by focussing the selection test on the gene segments that are a priori assumed to concentrate the positive selection signal. This is related to existing approaches that test for mutation clustering in cancer[49,50] but is limited to known driver genes, in which it can identify additional cancer types driven by a gene thus better describing the tissue spectrum for drivers.

We comprehensively analyzed the relationships between selection on point mutations and the CNA events at the same locus. One interesting observation concerns two-hit mechanisms, which are well established for TSGs and deletions, but were also reported in OGs in gains/amplifications[20,23,51]. We suggest that drawing on these interactions may be used to identify additional selected driver genes relevant to a certain cancer type, by testing for selection in tumors bearing CNA in the gene, potentially boosting signal-to-noise in the analysis. We report various plausible examples of genes that would be missed in the general selection test but would be identified by focussing on CNA-altered states.

Further, our observations clearly implicate not only CNA gains, but also losses as associated with increased positive selection on mutations in OGs. This was anticipated by an analysis of allelic imbalances of oncogene mutations, which were observed not only with CNA gains but also losses[24]. Based on this and our analysis of CNA-associated selection, we suggest that many mutations in OGs may not be fully dominant, but necessitate a two-hit mechanism for full fitness benefits to tumor cells. Classically, this would be via a CNA gain of the mutant allele; however, we propose a CNA deletion of the wild-type allele may be another common path towards the same fitness effect. (Our selection of data sources and design of our analyses (see Methods) suggest these are likely not CNA-neutral loss-of-heterozygosity events, but instead bona fide deletions). Thus, our study provides genomic evidence that the wild-type allele of various OGs can have a negative effect on tumor fitness, as previously shown for the particular case of *RAS* genes[24,52–54]. Positive selection can act on relative dosages of oncogenic alleles, rather than absolute ones.

Tumor suppressive effects of a mutated TSGs can be lowered by deletion of the wild-type allele, as per classical two-hit mechanism[18,19]. However, we find genomic evidence that a fitness benefit is also commonly obtained in TSGs by gaining additional mutant gene copies. This suggests that many of the TSG mutations act in a dominant-negative manner, and also that they are incompletely dominant. Thus two-hit mechanisms, appreciated to be prevalent in TSG, are often effectuated via copy number gains of a mutant allele.

However, our analysis also suggests there is some nuance to TSG inactivation mechanisms: while two-hit is common (including with CNA gains), only a few TSGs in cognate cancer types have an exclusively two-hit mechanism of inactivation. In other words, TSGs are often both one-hit (haploinsufficient or dominant-negative) and also two-hit genes and that these two mechanisms are not mutually exclusive. One explanation underlying this would be that the tumor would gain some fitness by the first hit on the TSG and additional fitness would be gained by the second hit. Also, the one-hit and two-hit mechanisms may have different relative importance for different genes: our global analysis of selection effect sizes in different CNA states suggest that cancer genes do not form well-separated clusters. Rather, cancer genes span the spectrum between one-hit and two-hit mechanisms of (in)activation, meaning that both mechanisms can be relevant to the same gene with varying frequencies across tumor types.

## Methods

### Modeling mutation rates with the MutMatch approach to obtain selection effect estimates

To describe the variability in mutation counts in a genomic locus, we model the expected mutation counts in vector E[**Y**] using the following count model, implemented as generalized linear model with a log link function, further regularizing the regression coefficients by using a weakly informative prior[55]:

$$\log E[\mathbf{Y}] = \omega t + \sum_i \mu_i m_i + \sum_j \beta_j z_j + \alpha + \log r \quad (1)$$

where $t$ is a target variable used to distinguish mutations accumulated in a genomic locus that is currently being tested for a selection signal (central gene) and the control loci (here, neighboring genes). The coefficient $\omega$ (or selection effect) reflects the log-fold change of mutation rates in the tested genomic locus (where $t = 1$), compared to the region used to model a baseline mutation rate (where $t = 0$). Positive $\omega$ estimate indicates enrichment of mutations and thus positive selection, while negative $\omega$ estimate corresponds to the mutation depletion in the locus of interest and thus negative selection. The use of the log link function ensures that the mutation counts modeled are always non-negative, regardless of the sign of the regression coefficients.

To control for the different activities of mutational processes on different oligonucleotides[1,4,56], the model stratifies mutation counts (non-negative integers) according to the 96-component spectrum in trinucleotide context using the categorical mutation type variables $m_i$ and their corresponding real-valued effects $\mu_i$, where $i$ iterates over the possible values that the mutation type variable can assume (such as A[C > A]A, A[C > G]A, etc.). Other types of optional variables can be included to control for e.g. inter-cancer type differences in selection (in a pan-cancer analysis), confounders or batch effects, such as the study where the data was sourced. We denote these variables as $z_j$ (can be any type of variable) and their corresponding real-valued effects as $\beta_j$. The base mutation rate $\alpha$ is included as the intercept of the model. Finally, to adjust the mutation counts to the maximal number of "opportunities" for mutations to occur, we include the number of nucleotides-at-risk $r$ as an offset in the count model. This allows us to implicitly normalize mutation counts to mutation rate (per nucleotide per sample), accounting for variations in DNA lengths across genomic loci and different composition of trinucleotide contexts in the central gene versus neighboring genes.

**Y** has a length that corresponds to the product of distinct categories for each variable on the right side of the equation. For example, considering $i = 96$ mutation types for the categorical variable $m$, 2 possible mutation loci (inside the central gene or inside the neighboring [control] genes) for the categorical variable $t$, and no additional dimensions (e.g. batch effects or tissues) meaning the variable $z$ has

only 1 possible value, the length of **Y** would be the product $2 \times 96 \times 1 = 192$. The corresponding effects $\mu_i$, $\beta$ are the fitted coefficients (they quantify the effect of a variable they correspond to on the mutation rate: $m_i$ and $z$, respectively) and can take any real-numbered value. When testing for conditional selection signals in genes (here, differentiating CNA states of the gene), we use an extended version of the regression formula that includes a condition variable $c$ encoding the state of the genomic region (can be a categorical or a continuous variable, depending on the condition of interest) with respect to the condition and the interaction term of the target variable $t$ with the condition variable $c$.

$$\log E[\mathbf{Y}] = \omega t + \sum_i \mu_i m_i + \sum_j \beta_j z_j + \gamma c + \delta tc + \alpha + \log r \qquad (2)$$

In the present study, the condition variable $c$ is a categorical variable that encodes the copy number state of the gene. Of note, the same MutMatch framework is generally applicable to comparing various types of conditions i.e. groupings of tumor samples by an arbitrary variable of interest. Here, we encode the CNA state as a categorical variable, with 3 possible levels: CNA loss, CNA neutral ("diploid") or CNA gain. The regression is performed to compare 2 levels of this variable at a time: loss versus neutral, excluding the gain state (Supplementary Data 3); or gain versus neutral, excluding the loss state (Supplementary Data 4). For each analysis, the length of the **Y** vector is 384, calculated as 2 (central vs. neighboring genes) × 96 (mutation spectrum) × 2 (diploid state vs. copy number altered state). The method makes no assumptions on the linearity of association between allele dosage and mutation rates.

The neighboring genes, largely sharing the CNA state of the central gene, provide an empirical baseline for the change in neutral mutation rates due to the allele dosage change resulting from the CNA. In cases where neighboring genes that do not share the CNA state of the central gene in a particular tumor sample, that sample was excluded from analysis for that gene. For cancer genes across various cancer types, the mean fraction of excluded samples was 0.56% for the diploid state ($Q_1 = 0$, $Q_3 = 0.82\%$), 1.06% for the gained state ($Q_1 = 0$, $Q_3 = 1.91\%$), and 0.85% for the deleted state ($Q_1 = 0$, $Q_3 = 0$). To reduce the effect of mutation rate heterogeneity between genes within the neighborhood, which could confound the analysis, we exclude genes in the neighborhood that have a different mutation profile than the central gene. This is quantified by the outlier score $S_{outlier}$, defined as follows:

$$S_{outlier} = \ln \frac{M_x / L_x}{M_y / L_y} \qquad (3)$$

where $M$ is the number of mutations observed in a gene, and $L$ is the gene length. The indices $x$ and $y$ refer to two different genes being compared: one is the central gene ($x$), and the other is a single gene from its neighborhood ($y$). Mutation counts of individual genes for this outlier analysis were obtained from intergenic and intronic regions within the ±20 kb of the center of each gene, using WGS data processed in Salvadores et al.[57], where the control for trinucleotide content was applied with minor adaptations (matching tolerance threshold ≤0.035 with 10,000 iterations). 11 genes without mutations (after a matching procedure) were excluded to avoid infinite values of $S_{outlier}$, and an $S_{outlier}$ value was calculated for 18,214 transcripts. For each gene, neighborhood genes were excluded if their mutation rates differ by more than approximately 22% from that of the central gene, using an outlier score $|S_{outlier}|$ of 0.2. The median number of neighboring genes for cancer genes after the removal of outlier genes was reduced from 9 to 5 genes (with $Q_1 = 2$ and $Q_3 = 11$), with some variation across chromosomes ($Q_1$ of medians = 3, $Q_3$ of medians = 6.5, with the overall median 4).

For cancer genes such as *KRAS* (chromosome 12), *TP53* (chromosome 17), *BRAF* (chromosome 7) and *PIK3CA* (chromosome 3) the final count of neighboring genes after all filters were applied was 2 (for *KRAS* and *TP53*), 4 (for *BRAF*) and 9 (for *PIK3CA*). The filtering process, which involved excluding samples where neighboring genes had a differing copy number state from the central gene, led to the removal of the following percentages of samples across the 13 major cancer types: 3.5% in the diploid state, 10.5% in the gained state, and 11.2% in the deleted state for *KRAS*; 3.2% in the diploid state, 8.8% in the gained state, and 11.5% in the deleted state for *TP53*; 1.1% in the diploid state, 1.7% in the gained state, and 4.3% in the deleted state for *BRAF* and 0.8% in the diploid state, 1.4% in the gained state, and 1.7% in the deleted state for *PIK3CA*.

For each gene, only nucleotides of coding exonic sequences of the most expressed transcript[58] (with adjacent 5 nucleotides upstream and downstream of each exon to account for splice sites) that are mappable according to the CRG75 Alignability track[59] were considered. Additionally, we removed positions that were unstable when converting between GRCh37 and GRCh38 (conversion-unstable positions)[60].

To remove biases in selection estimates caused by the low mutation counts in a region, we generated a null distribution of selection estimates for $\omega$ and $\delta$ coefficients using a randomization procedure. The mutations observed in each cohort are randomly shuffled between the tested and the control genomic loci, with a chance of acquiring mutations proportional to the number of sites in the loci. Then, the same selection model was applied to the randomized data (using 500 repetitions for all models except for whole-genome pancancer analysis across 13 major cancer types or for analyses in the Supplementary Fig. S14 − in this case 50 repetitions were used) to estimate the null distribution of selection effects. We subtract the median of this null distribution for a coefficient from the coefficient estimates obtained from the actual data for the later analysis. Original $p$-values obtained from regression models (by a Z-test on the coefficient and its standard error) were observed to be conservative, indicated by a genomic inflation factor $\lambda$ less than 1 (see Supplementary Fig. S2d). To address this, we adjusted the observed $p$-values using a distribution of $p$-values expected from randomized data, by determining an empirical $p$-value using quantiles in the simulated $p$-value distribution. This adjustment resulted in corrected $p$-values with a genomic inflation factor $\lambda$ approximately equal to 1. These corrected $p$-values were then utilized in all subsequent analyses, including the calculation of the FDR. The null distribution of selection effects was also utilized to estimate the standard error for confidence interval calculations.

We illustrate our model with a single fit example using the *KRAS* gene in models with CNA gain interactions in ovarian cancer. The coefficients should be interpreted as follows: The intercept of −17.20 represents the log of the baseline mutation rate when all variables are set to their baseline levels, specifically the neighboring genes and the diploid states. Exponentiating this intercept, $e^{-17.2}$, gives the baseline mutation rate per nucleotide, per tumor sample. The *isTarget1* coefficient of 4.56 indicates that the mutation rate in the *KRAS* gene, when all other variables are at their base levels, is $e^{4.56}$ times the rate of the neighboring (baseline) genes, suggesting a significantly higher mutation rate (positive selection) in the diploid state. The coefficient of 0.28 for the *CNA* variable signifies that the mutation rate in the CNA gained state is $e^{0.28}$ times the rate in the diploid state for neighboring genes. The interaction coefficient *isTarget1:CNA1* of 0.38 reflects that the mutation rate in the *KRAS* gene in samples with a *KRAS* gain is $e^{0.38}$ times (1.46-fold) higher than what would be expected from the multiplicative effects of the *KRAS* gene being selected without the CNA gain effect, and the CNA gain effect on the baseline mutation rates. For the *Mutation* variable, the coefficients ranging from −0.03 to 5.90 show how the mutation rates (in neighboring genes in the diploid

state) vary across the trinucleotide spectrum, expressed relative to the baseline mutation type (arbitrarily set at A[C > A]A), with $e^{-0.03}$ to $e^{5.9}$ times the rate of A[C > A]A. Importantly, as all effect sizes in this study are presented on a $\log_2$ scale, exponentiation should use base 2 to derive the corresponding mutation rates.

To obtain selection estimates on point mutations across all copy number states (for Figs. 1 and 2f analyses), we considered all samples together without stratifying into different copy number states as in (1), and not controlling for any additional factors $z$. Next, to obtain selection estimates on point mutations for the conditional selection study, we performed two separate analyses as in (2) with the categorical condition variable $c$ varying between analyses. One regression included diploid samples versus samples with CNA loss, another separate regression with diploid samples against samples with CNA gain. The conditional selection analyses do not consider the temporal ordering of the mutation versus the CNA event. Estimation of conditional selection in the GENIE validation dataset was performed similarly, with an additional variable denoting the cohort (DFCI or MSK) to control for possible batch effects between cohorts. In case of pan-cancer selection estimation, the cancer type was additionally included as a variable in the model.

Finally, we only focused on genes that had at least 2 (3) mutations across all samples where a gene was in a diploid state and at least 2 (3) mutations in samples where a gene copy was deleted or gained, separately for each cancer type for the discovery (validation) datasets, respectively. For the pan-cancer analysis we required at least 10 mutations to occur in the gene of interest across all cancer types used, and for selection estimates across copy number states we required at least 8 mutations to be present in each regression model.

The statistical test applied on the regression coefficients (log enrichments) was the Z-test, two-tailed, as implemented in the R function summary(). Multiple testing correction was performed using the FDR method separately for each coefficient type (for the diploid state, interaction with copy number loss or interaction with copy number gain), set of mutations, cancer type and group of genes (CEG2, TSG in cognate cancer types, OG in cognate cancer types, TSG in noncognate cancer types, OG noncognate cancer types and random genes). FDR correction was applied, accounting for a doubled number of tests, thereby representing the FDR for one-sided tests. For analyses in Figs. 3 and 4, multiple testing FDR correction was performed across all cancer types and therefore is more conservative.

For the selection in the diploid state $\omega$, we used one of the coefficients obtained from two different regression models: one comparing diploid to CNA loss and another comparing diploid to CNA gain; both should in principle yield similar estimates for the baseline diploid state (Supplementary Data 2). The choice of the coefficient for the diploid state was determined by whichever had the more significant $p$-value and the narrower confidence interval, or, in cases where the previous criteria were equivalent, the one derived using a dataset with a larger number of mutations.

## Mutation and copy number data collection and processing

We collected mutation and copy number data for two aggregated datasets in this study: a discovery data set and a validation data set. The discovery data comprised WES and WGS datasets from WES somatic single nucleotide variants (SNVs) from the The MultiCenter Mutation Calling in Multiple Cancers (MC3) Project[61], WGS somatic SNVs from the TCGA consortium[62], WGS somatic SNVs from Hartwig Medical Foundation (HMF) project[63], WGS somatic SNVs from Pan-Cancer Analysis Of Whole Genomes (PCAWG) dataset[64], WGS somatic SNVs from Personal Oncogenomics project (POG570) program[65], WES somatic SNVs from The Clinical Proteomic Tumor Analysis Consortium (CPTAC)-3 program[66,67], WES somatic SNVs from MMRF-COMPASS study[68]. The mutations that were called against the human version assembly GRCh38 were converted to the hg19 reference genome using

LiftOver[69]. Variant (nonsense, missense, synonymous) were annotated using ANNOVAR software[70]. Ultramutated samples, and samples with a high fraction of indels were separated from the rest of the samples for UCEC, CESC, COREAD, ESAD, ESCA, STAD, UCS cancers into separate subtypes denoted as "POLE" or "MSI". Cancer types between different datasets were matched to increase the sample size for each of them.

Altogether, we collected the genomes or exomes of ~23,000 tumor samples with mutations from 117 cancer type datasets; the number of tumor samples per each tumor type was ranging from 1 to over 1000 for PRAD (prostate adenocarcinoma), BRCA-Lum (a luminal subtype of breast cancer), COREAD (colorectal adenocarcinoma), MM (multiple myeloma), kidney, PAAD (pancreatic adenocarcinoma), and SKCM (skin cutaneous melanoma). The median number of tumor samples per cancer type was 57. The average number of SNV mutations in each cancer type was 28,805, and the median number of mutations was 4136. The cancer types with the biggest number of mutations are listed in Supplementary Data 6.

For the validation dataset, we downloaded mutation calls for 90,713 tumor samples across 75 cancer types from MSK-IMPACT and the Dana Farber Cancer Institute (DFCI) Oncopanel of the American Association for Cancer Research Project Genomics Evidence Neoplasia Information Exchange (GENIE) (Release 11.1; syn32309524)[44,45]. In these studies, only a limited number of cancer genes were sequenced. We determined the list of cancer genes that were targeted in both of these cohorts.

We collected CNA data estimated with GISTIC2, ASCAT or PURPLE programs for the majority of the samples listed above, covering 17,644 samples of the discovery cohort and 89,243 samples of the validation cohort. The estimates of the gene-level copy number status for the GISTIC2 copy number levels[12] were retrieved as binarized data, according to the recommended sample-specific thresholds. Copy number neutral loss-of-heterozygosity was not considered specifically in our analyses, and these events should not figure prominently in our CNA deletion tally. For the discovery cohort, only low-level gains or hemizygous deletions were considered for estimation of conditional selection upon CNA event, while high-level amplifications and homozygous deletions were not considered. For the validation cohort, we have aggregated the estimates for the copy number status of the gene such that any sample with the number of gene copies greater than 2 was considered to be in a gain state, and any sample with the number of gene copies less than 2 was considered to be in a deleted state.

## Annotation of cognate cancer types

To determine which cancer types were cognate (i.e. where cancer genes are positively selected), we used different sources of data: annotation available in the known databases such as Cancer Gene Census (CGC) or the MutPanning list[9,36], and additionally as the main criterion we used the selection estimates that we derived using MutMatch with the neighboring genes baseline in the discovery dataset of WES/WGS tumor. The latter approach (our annotations based on MutMatch) prevents missing cognate cancer types due to the different labels between annotation sources and our data, or uncertainty in the case when annotations are not detailed to specify which cancer subtype is cognate. Moreover, this helps to make sure that we do not focus on gene-tumor pairs that are listed as cognate, but may have too few mutations in datasets analyzed here due to the low number of samples in these cancer types.

To infer which gene-cancer type pairs are cognate, we have estimated the selection effects in the discovery data set for all the cancer genes in the analysis across all copy number states i.e. without including the CNA variable in the regression (Supplementary Data 1). This was for (i) consistency with methods to identify drivers in previous work, which do not account for CNA when identifying drivers, and (ii) to avoid circularities in subsequent analyses in this study that use these cognate and noncognate sets to dissect CNA-specific

selection. (Here we note that despite not including this variable, the analysis to identify cognate/noncognate genes does account for possible confounding effects of CNA on neutral mutation rate, by comparing against CNA-matched neighboring genes, as the other analyses we performed). Cancer types where a gene was positively selected ($\omega > 0$ at a FDR $\leq 2\%$) were considered to be cognate cancer types for this gene. To infer noncognate cancer types (where cancer genes are not strongly selected), we required FDR $\geq 50\%$ or negative selection ($\omega < 0$). The rest of the gene-cancer type pairs that did not fall into either of these categories were separated into the "uncertain" group.

## CADD scores
We downloaded a bigWig file that contained pre-computed PHRED-like $-10 \times \log_{10}$ rank total scaled Combined Annotation-Dependent Depletion (CADD) scores for each genomic position (v.1.4)[46]. The highest CADD score of any 3 possible substitutions at a site is provided in this file, with higher values indicating a higher level of deleteriousness of the variant. Scaled CADD-scores assign value 10 to the top-10% of all the CADD scores in the reference genome, value 30 to the top-0.1% and so on. To separate regions where mutations are likely to have a functional impact versus regions that are evolutionary unconstrained, we used a cut-off of 20. In this way, positions in a gene with top-1% most deleterious mutations were tested for selection using a background mutation rate model from unconstrained gene regions where CADD score was less than 20.

## Essentiality scores
We used mean cell-essentiality scores across different cancer cell lines (CERES score by CRISPR-Cas9 screening approach[71]) and essentiality score derived from a population data (selection of Loss-of-Function (LoF) germline point mutations summarized in Loss-of-function Observed over Expected Upper bound Fraction (LOEUF) score[72]) to independently define cell line-essential and population genomics-essential genes.

## Hotspots
We defined "permissive hotspots" based on the codon-specific frequency of mutations in a discovery cohort with the cut-off of 2 mutations per codon, which corresponds to the 91.6% of recovery of Trevino hotspots and 59% of recovery of 3D hotspots[73] (0.235% and 0.007% of all permissive hotspot sites, respectively). Some 3D clustered hotspots that were not recovered with this cut-off are located in proximity to a hotspot in the discovery cohort. To avoid missing such sites that are closely located to a detected hotspot, but have fewer mutations (due to the limited number of mutations in the discovery dataset), we lengthened our hotspots to include them. If another mutation was found within 3 codons upstream or downstream from the hotspot, the hotspot was extended to include this mutation. The median length of the hotspots defined this way was 12 nucleotides, or 4 amino acids. This recovered 75.2% of 3D clustered hotspots, and 95.4% of Trevino hotspots[43].

## NMD-detected and NMD-evading regions
Genomic regions were split into those where premature termination codons (PTCs) lead to the degradation of the mRNA in a process of nonsense-mediated mRNA decay (NMD) or those where nonsense mutations lead to a translation of a truncated protein sequence. The efficacy of NMD for PTCs in a human model was predicted using the NMDetective algorithm[74]. Regions with the NMDetective-A score >0.52 were classified as NMD-detected regions, and the rest were classified as NMD-evading regions.

## VAF analysis
VAF analysis was performed using the mutations from TCGA, PCAWG, Hartwig, MMRF and POG570 datasets. We calculated allele frequencies

for missense and nonsense mutations located in the same genomic regions that were used for the estimation of selection (including removal of unmappable and conversion-unstable positions). VAF estimates were then adjusted to control for a sample purity (Consensus measurement of Purity Estimation from TCGAbiolinks[75] for TCGA samples, purity estimates from the Purple tool for PCAWG, Hartwig and MMRF samples and sample annotation data for POG570 samples),

$$\frac{n_a}{(n_a + n_r)} \times \frac{1}{P} \qquad (4)$$

where $n_a$ and $n_r$ is the number of alternative and reference reads, and $P$ is the sample purity. In this analysis, we excluded samples with a tumor purity exceeding 1, and those where more than 10% of the mutations had an adjusted VAF exceeding 1.

## Statistical tests
To assess whether a specific collection of mutations (such as nonsense and nonsynonymous mutations located in permissive hotspots) within a targeted gene group is subject to selection pressures, we compared the adjusted estimates for each mutation type and coefficient type with those of a random gene group, derived using the same mutations, through a two-tailed Mann–Whitney U-test. In a similar fashion, to evaluate the significance of differences in the first or third quartiles ($Q_1$ or $Q_3$) between our gene group of interest and the random genes, a permutation test with 10,000 iterations was employed.

## PCA on selection estimates
The de-biased estimates for (i) selection effect in the diploid state $\omega$, (ii) selection effect change upon deletion $\delta_{\text{deletion}}$, and (iii) selection effect change upon gain $\delta_{\text{gain}}$, for each gene in each cancer type were used to perform a global analysis of copy-number dependent selection in the human soma. To reduce the likelihood of selection in the control group of random genes, we excluded from them genes with low LOEUF score (ranked in the most essential 30% of the genes) derived from population variants[72,76], CEG2 essential genes[37] and cancer genes using MutPanning and CGC gene lists[9,36,77]. We also excluded genes that were either very short or very long, allowing up to 30% of difference between the median number of the nucleotides in a gene that was used in the regression model. Finally, we removed genes with very high or very low gene expression using the normalized transcript expression levels summarized per gene in 54 tissues based on transcriptomics data from Human Protein Atlas (HPA) and Genotype-Tissue Expression (GTEx)[78].

In this analysis, we focused on cancer types and the genes for which selection effects were estimated in at least 85% of gene-tumor type pairs i.e. common driver events. Because of the high number of missing values in the case of selection estimates for strict (Trevino) hotspots, we excluded estimates inside or outside of the strict hotspots from the PCA analysis (the permissive hotspot estimates were still included). For the gene-tumor type combination passing these filters, missing values were imputed using *imputePCA* function from the "missMDA" package in R.

PCA was performed on the centered and scaled data. To avoid overplotting, we only show 15 genes from the control group of random genes and 15 genes from the control group of essential genes in a subset of cancer types (COREAD, LUAD, BRCA-Lum, PRAD, LUSC, BLCA, MM, LUSC, SKCM, PAAD, STAD, THYM, THCA).

## Clustering of cancer genes in different cancer types
We performed centered and scaled PCA on the restrained and corrected coefficients to reduce the impact of outliers. To do so, we first forced the highest absolute estimates to be within 98% of all estimates (extreme estimates of top and bottom 1% were capped at the level of

1st percentile or 99th percentile). Next, we subtracted one standard error from each positive estimate (or added it to the negative ones) to obtain more conservative estimates of the selection for each gene-tumor pair.

Missing values were imputed using *imputePCA* function from the "missMDA" package in R. We chose the first 9 PCs based on the scree plot for further classification and performed a rotation of PCs using the varimax method to ease the interpretability of obtained factors. We clustered gene-tumor pairs for 25 largest cancer types excluding hypermutated and MSI samples using partitioning around medoids (*pam* function from package "cluster" in R) based on Euclidean distances between samples. We performed two clusterings: first yielded 10 clusters (silhouette width criterion), with the largest cluster corresponding to the gene-tumor type pairs showing no selection (factor scores ~0). To refine this analysis, we excluded pairs from this cluster and performed another clustering, finally discovering 9 clusters of gene-tumor type pairs (Supplementary Data 5). Enrichments of particular gene classes for each cluster were estimated using two-sided *p*-values derived from chi-squared residuals (*chisqtest* function from "zebu" package in R).

### Reporting summary
Further information on research design is available in the Nature Portfolio Reporting Summary linked to this article.

### Acknowledgements
We are grateful to Marina Salvadores for assistance with determining mutation burdens of genes. Work in the F.S. lab is supported by an ERC StG "HYPER-INSIGHT" (757700) and ERC CoG "STRUCTOMATIC" (101088342), Horizon2020 project "DECIDER" (965193), Horizon Europe project "LUCIA" (101096473), Spanish government project "REPAIRSCAPE", CaixaResearch project "POTENT-IMMUNO" (HR22-00402), an ICREA professorship to F.S., the SGR funding of the Catalan government, the Novo Nordisk Fonden starting package, and the Severo Ochoa centers of excellence award of the Spanish government to the hosting institution. E.B. was funded by an AGAUR-FI fellowship of the Catalan government. This publication and the underlying research are partly facilitated by Hartwig Medical Foundation and the Center for Personalized Cancer Treatment (CPCT) which have generated, analyzed and made available data for this research (request number DR-260). The results published here are in whole or part based upon data generated by the TCGA Research Network: https://www.cancer.gov/tcga.

### Author contributions
E.B. collected and curated the data, devised methods and implemented the code, performed all analyses, visualized the data, and interpreted the results. F.S. conceptualized the analysis, devised the overall methodology and supervised the study. F.S. and E.B. drafted the manuscript jointly.

### Competing interests
The authors declare no competing interests.

### Data availability
The relevant data generated is included in the Supplementary material and can be found at this link: https://figshare.com/articles/dataset/Supplementary_Data_26029309. Interactive visualizations for selected figures from this publication are available at https://ebesedina.shinyapps.io/mutmatch_web/. Published data and resources used in this work: vcf files and CNA data for TCGA WGS [https://portal.gdc.cancer.gov/] (raw data available under restricted access via dbGaP accession phs000178 [https://www.ncbi.nlm.nih.gov/projects/gap/cgi-bin/study.cgi?study_id=phs000178]), HMF [https://www.hartwigmedicalfoundation.nl/en/] (available under restricted access via https://www.hartwigmedical foundation.nl/en/data/data-access-request/), TCGA WES mutation calls from MC3, TCGA WES CNA calls [https://portal.gdc.cancer.gov/], PCAWG, POG570, CPTAC-3 [https://portal.gdc.cancer.gov/] (available under restricted access via dbGaP accession phs001287 [https://www.ncbi.nlm.nih.gov/projects/gap/cgi-bin/study.cgi?study_id=phs001287]), MMRF-COMMPASS [https://portal.gdc.cancer.gov], GENIE [https://www.synapse.org/], Project Score CRISPR genetic screening data [https://depmap.org/portal/download/], gnomAD [https://gnomad.broadinstitute.org/downloads], CADD scores [https://krishna.gs.washington.edu/download/CADD/bigWig/], NMDetective [https://figshare.com/articles/dataset/NMDetective/7803398], CRG75 Alignability track [https://hgdownload.soe.ucsc.edu/goldenPath/hg19/database/], CUP bed files [https://github.com/cathaloruaidh/genomeBuildConversion/#2-novel-cup-bed-files], Top-rank expressed transcripts of protein-coding genes [https://tregt.ibms.sinica.edu.tw/index.php#tab6]. Source data are provided with this paper.

### Code availability
The implementation of the MutMatch algorithm is available at https://github.com/ebesedina/mutmatch[79]. For citation of the code used in this study, please reference https://zenodo.org/doi/10.5281/zenodo.11619414.

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
