## [Peer Review File · Nature Communications]

REVIEWER COMMENTS

Reviewer #1, expertise in cancer genomics and evolution, selection and statistics (Remarks to the Author):

The authors developed MutMatch, a statistical method to measure the selective advantage of mutations combined with copy number alterations, occurring in tumour suppressor genes and oncogenes in specific tumour types. The question of how different types of mutations interact to confer tumours more or less fitness is key to the development of targeted therapies in the context of precision oncology. The model is based on a linear regression that associates the number of mutation counts on a gene from WGS and WES samples, to a coefficient of selection (positive or negative) contributed by an interaction term of the mutation with the copy number state of the affected gene, plus other correction terms. In particular, the selective advantage is measured with respect to a baseline estimated from neighbouring genomic regions. The size of the aggregated cohort seems reasonably large (~18.000 samples) to provide enough statistical power to the proposed results and, in general, the paper is well written and the content exhaustive. Although the model is well thought and has the potential to provide valuable information for perspective translational studies, I have some major concerns about confounders possibly arising in its application:

The model assumes a linear relationship between the observed counts on a genomic locus and the number of allele copies thereof. While this might be a good approximation for RNA sequencing data (the more the number of copies, the higher the expression level), I am not sure about its applicability to somatic mutations, for which the mutation rate is proportional to the multiplicity of the mutation, i.e. the number of mutant alleles, rather than to the total number of copies of the gene. For example, consider a gene that is commonly subject to trisomy events. While the number of allele copies is always $n = 3$, the expected number of counts from that gene will depend on whether the mutation is more often previous or subsequent to the CNA. This bias may systematically affect the analyses proposed in the paper, and it clearly shows up in the analysis of VAF distribution. In Figure 4a for instance, the median adjusted VAF of missense mutations in amplified oncogenes is shown to be higher than that of random genes, but the distribution is clearly bimodal, with two seemingly equal-size populations of mutations occurring after (not amplified, adjusted VAF ~ 0.33) and before (amplified, adjusted VAF ~ 0.66) the CNA. Another conclusion that the authors draw from the model, and that might be confounded by mutation multiplicity, regards LOH as a novel mechanism of double-hit activation for oncogenes. If in the deleted oncogenes the LOH is copy-neutral, mutations might still be present in two copies, yielding the same effect of an amplification. Again, this is evident from the presence of multiple VAF peaks.

Another major concern regarding the model is that it does not take into account the clonality of mutations. Especially in WGS and WES samples, most of the mutational burden is usually contributed by passenger mutations. While the authors have focused on non-synonymous variants, still I would expect the majority of them to be subclonal, and a confirmation of that is found in the VAF spectra the authors present in the paper, where adjusted VAF values < 0.5 are strongly supported. The current common

interpretation of a variant under selection is that it has the power to drive cancer growth, and it is thus expected to be found at high frequency. I think the authors' estimates of gene selection coefficients might be dramatically confounded by the presence of large numbers of subclonal passenger mutations, especially because they use the number of mutations as a proxy, rather than their frequencies.

In order to reach adequate statistical power, the authors assembled a cohort of ~18.000 samples from multiple studies. Whereas in the regression model they take into account the possible effect of different data sources using separate terms, I think it might be useful to check a posteriori whether their results are validated separately across the various cohorts. Additionally, the authors do not mention any a priori check on the quality of their samples: I expect the assembled cohort to be widely heterogeneous in terms of sequencing coverage and sample purities, contributing very differently in terms of statistical power. For example, a sample with low coverage and purity would result in the exclusion of lots of low-frequency mutations under neutral or even negative selection. I also noticed that the presented VAF spectra have, in the majority of cases, support from values greater than one. Since it is impossible that a variant is present in more than 100% of the tumour cells, there must be errors in a number of purity estimates. I think it might be useful to quality-check purity estimates and, if needed, exclude unreliable cases from the cohort.

From the authors' analysis of negative selection, it is not clear to me how excluding hotspots from genes should help in revealing negative selection acting on nearby spots. Assuming infinite size and a uniform rate of neutral mutations on the genome, I would expect positively selected regions to decrease the mutation rate of nearby regions, with the effect of masking neutrality as negative selection.

In addition, I have some minor comments:

I think the Methods section is not detailed enough to be clear and easily followed. For instance, the authors should define the domain of every variable and coefficient in the model. The specific statistical tests used for different variables (e.g. selection coefficient, median adjusted VAF) should be reported and motivated.

The authors exclude outlier genes based on a threshold of a score defined as the log ratio of the number of mutations per base. Is the choice of the threshold arbitrary? Are their results robust upon changes in its value? How many outliers are they excluding from the analysis?

The variable Q3 is introduced in the main text without any previous definition.

For the determination of cognate gene-cancer type pairs, the authors have computed selection coefficients across all copy number states. What is the reason for excluding this parameter, given that it represents a major contribution in their model of selective advantage? For example, a certain cancer type could be related to a greater number of mutations on a gene due to a greater exposure to copy number alterations in certain chromosomes, rather than to the cancer type-specific positive selection of that gene.

The dN/dS method is mentioned by the authors as a method for estimating selective advantage of mutations that, differently from the presented MutMatch model, does not take CNA states into account. I think a comparison between dN/dS MutMatch estimates of selection coefficients might be interesting. In particular, the authors could check whether there are differences in dN/dS values between genes harbouring mutations and CNA with respect to diploid cases (in analogy with Figure 3).

Reviewer #2 (Remarks to the Author):

Reviewer #3, expertise in cancer genomics and evolution and selection (Remarks to the Author):

Besedina & Supek conduct a rigorous investigation of patterns of selection of oncogenes and tumour suppressor genes across cancer types and varying copy number states. A statistical method called MutMatch, which estimates selection by comparing exonic mutation rate with that of surrounding mutations assumed to be under neutral selection, was applied to over 18,000 tumour samples. Indications of negative selection were often observed for somatic mutations outside of hotspot regions, suggesting certain levels of gene essentiality for oncogenes. Additionally, selection patterns in line with somatic copy number aberrations were analysed to discern additional driver events which may arise from either one- or two-hit events. Intriguingly, the authors found that two-hit events including point mutations appeared to be associated with both copy number gains and losses, suggesting alternative mechanisms of gene loss which may allude to previously overlooked therapeutic vulnerabilities.

Overall, this paper presents a very interesting rethink of established truths about genetic selection, with wide-ranging implications. The figures are broadly very clear, although it would be worth ensuring that they are properly referred to within the text and that the captions are more explicit. If the manuscript

suffers from one issue, it is that a fair amount of the results appear slightly under-powered, which manifests as insignificant results presented as substantial (e.g. lines 562-564), or results with high significance which appeared marginal. However, the authors are open about these limitations in the text, and it could be argued that with some additional explanation these issues do not detract from the broader messages of the paper.

Although the insights provided by the manuscript are valuable to the cancer evolution community, some concerns need to be addressed:

- Overall, the Results part was rather difficult to read due to the sheer number and complexity of the analyses performed. It would be helpful if the authors tried to make the text easier to read for a lesser specialised audience, e.g. by introducing the question that they aim to answer at the beginning of each subsection and concluding each subsection with a sentence that summarises the main messages taken from the respective analysis. Shortening the main text could also help.

- I would also recommend reviewing the orders of figures and labelling within the text. At points the figures are not referred to in order, making the text somewhat difficult to follow, whilst other figures (e.g. Fig. 5c) are not referenced at all (likely a mislabelling). Can you also please make sure all figure captions are sufficiently explicit to be understood without referring to the main text or the methods (e.g. what does "bin" mean in Fig 2E?).

- Equations (1) and (2): For the MS96 variable, $m_i \mu_i$ should be replaced with $m_j \mu_j$

- Lines 329-331: The $\log_2(\text{mutation_rate}) < -0.2$ threshold for negative selection feels remarkably lenient, especially regardless of significance. Why was this threshold selected, and how can this be discerned from Figure 2A-B? Additionally, it would be worth commenting on the possibility that this is neutral selection.

- Lines 333-341: Of the two mechanisms underlying negative selection pressures, the second one (tumours depend on oncogene function even if it is not a tumorigenic driver) feels far more logical. Have these mechanisms been reported at all in the literature, or in this case are they primarily hypothetical/speculative?

- The section entitled 'Essential genes show modest signatures of negative selection in somatic cells' does not highlight particularly exciting or strong results, and doesn't appear to add much credence to previous results. Perhaps it could be shortened. Also, it would be helpful to make it clear in the text that the data analysed here is in cell lines, as someone who is not familiar with CERES scores might be misled

to think this is still data from patient samples. Introducing more clearly the question and the motivation to look at cell lines to study essentiality would help readability.

- The authors mention that the CERES score is more correlated with their estimates of negative selection than the LOEUF score (lines 368-370). How is this determined? The correlation coefficient of CERES vs LOEUF (Fig S4D) appears greater and more significant than CERES vs pan-cancer selection in the diploid state (Fig S4A).

- The differences in CERES scores between cancer genes and random genes (Fig S4E) is incredibly marginal and this should be highlighted. However, it could be argued that this does not detract from the results presented in Fig 2, due to both the cancer type-specificity of negative selection demonstrated in Fig 2D and the differences between cell lines and patient samples.

- Lines 507-508: it may be worth specifying that, according to Fig 3A, the increased selection of OG mutations upon CNA deletion occurs only for nonsynonymous mutation in permissive hotspots. Does this indicate that these mutations are likely gain-of-function?

- The authors have combined a range of publicly available cohorts in their analyses, most of which are from primary tumours and some, like the Hartwig dataset, from metastatic samples. It would be interesting to see if the same signals of selection are observed both in primary tumours and metastases or if there are differences.

Reviewer #4 (Remarks to the Author):

Reviewer #5, expertise in cancer genomics and evolution and selection (Remarks to the Author):

This paper by Besedina et al presents a regression-based method for finding selection signals, MutMatch, which seems cleverly and carefully devised. The method is here employed, among other

things, for finding SNV selection signals that are associated with specific copy number states. Application to large scale cancer genomics data points to some interesting interactions between copy number events and selection acting on SNVs. Negative selection among "out-of-hotspot" non-synonymous mutations in oncogenes is another interesting result. In general, my main concern is some lack of clarity and conciseness in the presentation overall. The article is very long making it less accessible and appealing than it could be. Some careful editing is needed in my opinion, for clarity and to make it more to-the-point.

E.g. the introduction has a good discussion about confounders when detecting selection, including that fact that CNA is itself a mutation rate covariate. However, I general the introduction could be much shorter, while still retaining the useful information. There are also several typos. There is similar room for improvement and shortening throughout Results. E.g. a lot of text is devoted essentially to the fact that focusing on mutation hotspot regions increases power to detect driver events - this is not very surprising and the same should be true using many other methods. It would probably be good to put the more emphasis on the truly novel findings so they don't risk getting lost.

The title is not all that clear to me, and before reading the manuscript I took it to mean that the copy number events in themselves were the drivers. Perhaps saying "generate driver _mutations_" would be clearer (although see comment below about cause and effect).

In the text, there seems to be an implicit assumption (with some exception) of CNAs preceding mutations (e.g. as they "generate" common driver events, or "upon" deletion/gain). While it can sometimes in principle be possible to decipher the temporal order (e.g. above-heterozygous VAF for an SNV suggests it occurred before CNA), I don't see that this has been considered. E.g. one example is the "CNA and mutation epistasis in the same gene" effect mentioned in the introduction, e.g. where a heterozygous mutation (first hit) can precede and lead up to selection for a second hit copy number deletion event. Same for OGs and allele imbalance: "we turned to examine selection on point mutation .. upon deletions" (row 462) - what says it's not selection for deletions upon point mutations? The language may relate to how CNA is used as an independent variable in the regression, but does not necessarily reflect temporality during tumor progression.

Regarding the principle underlying the method: there are certain assumptions, e.g. "central genes are expected to be similarly affected by CNAs" and "domain-scale mutation rate is adjusted for as _typically_ the neighbour genes will be in the same 'domain'. Additional "refinements" are added to account for cases that violate these assumptions, which is reassuring. However, is a deviation of neutral mutation rate in the central gene of interest (compared to neighbour genes) necessarily detectable? Doesn't the idea of using the "neighbourhood" in the first place stem from a difficulty in estimating neutral mutation rate in the central gene alone? Generally speaking, significant signals are all either 1) actual selection or 2) exceptions when the model fails - and latter doesn't need to happen too often for these to add up in the end.

Related to the above: a linear model is used while mutations are count data, and there is a risk of over dispersion effects. It seems this is corrected for by a randomisation procedure for low counts. Is this procedure sufficiently validated, i.e. could it be worth investigating the relationship between mutation counts and P-values in the final results?

Row 172: It is not clear to me how this is a "control": wasn't the very definition of a non-cognate cancer type based on the lack of selection in a given gene? Or I suppose this refers specifically to nonsense mutations, while the cognate/non-cognate classification considers all possibilities for a given gene...? Again this is probably just be a matter of describing things more clearly.

There should be some relevant references for EGFR mutations in breast cancer, including the TCGA marker paper.

Unless I missed this, to reduce concerns about possible confounding effects (factors not considered by the model), many of the analyses would benefit from inclusion of "non-driver genes" as controls (as a group, selection should be minor).

All in all, this is at its core an interesting and comprehensive piece of work, which would have benefited a lot from having a strict word limit during writing thus forcing a more to-the-point and accessible presentation.

Minor:

Some specific examples related to text clarity:

Abstract: "Consistently, focussing on known positively selected regions identifies additional tumor types where an oncogene is relevant". By this, I assume the authors mean to say: 1) positive and negative selection occurs in different regions within oncogenes, and 2) only considering the former regions give better signals. This is not clear and requires some thinking. Missing is the information is that the two signals (positive and negative) arise in different regions (within a specific gene region...). Obvious only once you understand it.

Similarly, the logic behind "conversely" in the abstract (also relates to title) is not clear to me: there is increased selection on oncogenes both following (?) gain and deletion. Then "conversely" there is

increased selection (or at least mutations) in TS genes during gains? Is there really an opposite relationship here?

REVIEWER COMMENTS

Reviewer #1, expertise in cancer genomics and evolution, selection and statistics (Remarks to the Author):

The authors developed MutMatch, a statistical method to measure the selective advantage of mutations combined with copy number alterations, occurring in tumour suppressor genes and oncogenes in specific tumour types. The question of how different types of mutations interact to confer tumours more or less fitness is key to the development of targeted therapies in the context of precision oncology. The model is based on a linear regression that associates the number of mutation counts on a gene from WGS and WES samples, to a coefficient of selection (positive or negative) contributed by an interaction term of the mutation with the copy number state of the affected gene, plus other correction terms. In particular, the selective advantage is measured with respect to a baseline estimated from neighbouring genomic regions. The size of the aggregated cohort seems reasonably large (~18.000 samples) to provide enough statistical power to the proposed results and, in general, the paper is well written and the content exhaustive. Although the model is well thought and has the potential to provide valuable information for perspective translational studies, I have some major concerns about confounders possibly arising in its application:

We thank the reviewer for a summary of the main points from the paper, and for highlighting the strengths of the study as well as translational implications. We have performed various additional analyses to control for potential confounding factors and ruled out their influence on our principal results. Please see our answers below for specifics

The model assumes a linear relationship between the observed counts on a genomic locus and the number of allele copies thereof. While this might be a good approximation for RNA sequencing data (the more the number of copies, the higher the expression level), I am not sure about its applicability to somatic mutations, for which the mutation rate is proportional to the multiplicity of the mutation, i.e. the number of mutant alleles, rather than to the total number of copies of the gene.

The reviewer highlights an important point, which merits clarifying.

Firstly, we apologize for the possible confusion due to ambiguous wording in the Methods, which originally stated “*To describe the variability in mutation counts in a genomic locus, we model raw mutation counts Y using the following generalized linear model, regularizing by using a weakly informative prior distribution of regression coefficients (60)...*” What we use is, however, not a linear model. The model is implemented in the “generalized linear model” (glm) framework. Glm allows using various link functions in the regression to model non-normal data -- hence the qualifier “generalized” -- and these link functions often are nonlinear. In our analysis, this is indeed the case: the model uses a log link function, which is used for regression analyses on count data (Poisson, negative binomial), which is appropriate since in this work we fit models to observed mutation counts. Our model is a *bona fide* count model, employing the implementation from

Gelman et al. (Ann Appl Stat 2008); this particular implementation supports a form of regularization on the coefficients (“weakly informative prior”), which helps convergence and stability of estimates with sparse data.

Secondly, our model does not have any assumption of a linear relationship between DNA copy number and the expected mutation rate. This is due to two reasons. First, our comparisons are always between two categorical levels of the CNA indicator variable, so there is no fitting of linear or nonlinear function -- these distinctions are not meaningful with binary categorical variables. Second, critically: in MutMatch the neighboring-genes (equally affected by the CNA as the central gene) provide an empirical baseline model of neutral mutation rates. By normalizing to this baseline, we implicitly account for any effects of CNA on baseline mutation rates, be they linear or nonlinear. We elaborate on these two points below:

(i) We do agree with the reviewer that fitting a linear model between the number of copies in DNA and mutation rate would not be ideal. Therefore, our model indeed does not assume a linear relationship. We do not look at the number of allele copies but instead consider the 3 copy-number states -- losses (-1), neutral (“diploid”, 0) and gains (+1) -- encoded as categorical variables (not numeric). These are considered in two separate comparisons: neutral versus loss CNA state, and neutral versus gain CNA state. In each of these comparisons, differences in mutation rates (counts, normalized to nucleotides-at-risk) are tested. Thus we only perform comparisons between two categories, and we do not think this assumes a linear relationship. If we were comparing all 3 CNA states at once with them being ordered, which we are not doing, indeed one could speak about a linear fit or a nonlinear fit. We hope we understood the reviewer’s query correctly, and are happy to further clarify or support with analyses.

(ii) It is important to clarify that we do not directly assume a particular dosage of DNA at the CNA loss state or at the CNA gain state -- we do not normalize the mutation rate to the dosage of DNA. Instead, we normalize the mutation rate empirically, to the mutation rate in neighboring genes, which share the same CNA state as the central gene (we note, importantly: in the cases where they do not share CNA state -- which are not common as CNA events tend to span long segments -- such examples were excluded from analysis). Thus, whatever neutral relationship exists between allele copy number and mutation rates, linear or otherwise, will be mirrored also in the neighboring-genes internal control in our approach, and will therefore be accounted for by our regression.

We have edited the text at the beginning of Methods to make it clearer that we indeed do use a count model (not a linear model), that there is no assumption of linearity with DNA dosage, and that neighboring genes empirically adjust for neutral changes in mutation rates due to allele dosage. The methodology is also explained visually in the schematics in Fig. 1a and a new explanatory schematic (showing mutation rate baselines and comparisons) is now provided in Supplementary Fig. S1.

At this point, we would also mention that the MutMatch methodology has now been revised to provide much better calibrated p-values (see new Supplementary Fig S2), by drawing on a randomization test. In the original study, the p-values were quite conservative (deflation in q-q plot) which had caused false-negative calls, and this has now been fully addressed ($\lambda \approx 1.0$, see Fig. S2). We also provide additional benchmarks of the method, testing it on the ability to identify known driver genes from Cancer Gene Census; this shows competitive performance to covariate-based methods such as MutSigCV (which are, importantly, not able to test CNA interactions). We test the robustness of our method to noise from reduced dataset size; in this test it exceeds a state-of-the-art tool (Fig. S2). We hope these revisions of the methodology, and the additional benchmarking instill more confidence that MutMatch is overall a sound approach for statistical analyses of cancer genomes.

For example, consider a gene that is commonly subject to trisomy events. While the number of allele copies is always $n = 3$, the expected number of counts from that gene will depend on whether the mutation is more often previous or subsequent to the CNA. This bias may systematically affect the analyses proposed in the paper, and it clearly shows up in the analysis of VAF distribution. In Figure 4a for instance, the median adjusted VAF of missense mutations in amplified oncogenes is shown to be higher than that of random genes, but the distribution is clearly bimodal, with two seemingly equal-size populations of mutations occurring after (not amplified, adjusted VAF ~ 0.33) and before (amplified, adjusted VAF ~ 0.66) the CNA.

If we understand correctly, this point was addressed by employing neighboring genes as an empirical baseline for mutation rates. In more detail: in this hypothetical example, a locus with trisomy would indeed have different VAF distributions depending on the relative timing of the CNA gain versus the mutation at the locus (higher VAFs on average if mutation is prior to CNA). This is precisely the point that we aim to address with the MutMatch approach, which uses a neighboring-genes control matched by CNA state to the “central” gene-of-interest, to derive the expected counts for that central gene. (As a side note: we have an additional method applicable to panel-sequencing data, which segments coding regions of a single gene according to CADD pathogenicity score, and this also adjusts for mutation rate biases due to allele dosage effects with CNA).

In the example the reviewer mentions: the ordering of CNAs and mutations, under neutrality of mutations, will (averaging across many tumor genomes) similarly affect the central gene and the control neighboring genes. If there is a systematic difference between the central gene versus the CNA-matched neighboring genes, this is not a bias, but would be a signal of selection on the central gene, which our MutMatch detects. For example, positively selected mutations in the central gene happen before CNA and afterwards the mutant allele in the central gene undergoes a CNA gain (which also affects neighbor genes). In contrast, with neutral mutations in neighboring genes that happened before this same CNA gain event, this same CNA gain affects either mutant or the wild-type allele with similar chances. This difference between central gene and neighbor genes registers as a higher mutation number in the central gene (because VAFs are pushed towards higher values) and is indeed seen by MutMatch as a signature of selection on the central gene, acting upon the mutation-CNA combination.

With respect to our supporting analysis of VAFs highlighted by the reviewer (Fig. 4a), let us clarify: MutMatch itself -- method employed for almost all analyses in the paper -- does not rely on VAFs, and has empirical controls for mutation risk that implicitly account for effects of CNA/mutation ordering (see above). The supporting analysis of VAFs in Fig. 4a is meant to reinforce the evidence from the main MutMatch analysis; while this VAF analysis in Fig. 4a did not use the neighboring-genes baseline, in place of this it does show a “Random” genes group that provides estimates under neutrality. In the VAF analysis in Fig. 4a, the FDRs (written in the panel) are for comparisons against this neutral, random-genes group. We have now, as an additional control, added the data from neighboring-genes to the VAF analysis (these genes are matched to each of the gene groups of interest shown: OGS, TSGs... see additional row in Fig. 4a). These indeed show very similar VAF distributions as the random-genes control (present in VAF analysis in the original manuscript), supporting the original use of the random-genes control in the VAF analysis.

In summary, our analyses of selection on CNA/mutation interaction by MutMatch controls for CNA factors as per the empirical control described above: there is not an “expected” number of mutations, but only observed mutations in the neighboring genes, which correct for any neutral biases resulting from CNA including due to e.g. ordering of mutations. The evidence of this is that the selection effects of MutMatch are centered at ~ 0 for genes from the “Random genes” set (a control group that excluded all drivers and essential genes); and this is similarly the case for the CNA-neutral as well as in CNA-gain and in CNA-loss states which also have median ~ 0 (see Fig. 4a and Supplementary Fig 2a), thus under neutrality the CNA/ordering of mutations effect do not bias our estimates in the MutMatch framework.

To clarify our methodology further, we implemented various edits throughout the Methods text, which are marked by track changes. If the reviewer has further queries or suggestions for analyses that could further answer this point, we would be happy to implement them.

Another conclusion that the authors draw from the model, and that might be confounded by mutation multiplicity, regards LOH as a novel mechanism of double-hit activation for oncogenes. If in the deleted oncogenes the LOH is copy-neutral, mutations might still be present in two copies, yielding the same effect of an amplification. Again, this is evident from the presence of multiple VAF peaks.

We appreciate the reviewers point about how the association of oncogene mutations with CNA losses may involve either a heterozygous deletion (as we claim), or alternatively a copy-number-neutral LOH (CNN-LOH) at the locus. Indeed our analyses focussed on deletions, not CNA-LOH, so the effect we observed is due to deletions. In detail:

Our analysis draws on various data sources to assemble the large, 17,644 tumor WES/WGS cohort with CNA calls. The two quantitatively major sources of data are:

(1) The TCGA data ($\sim 10k$ samples analyzed), where we used ASCAT2 scores for calls of CNA events derived from SNP arrays. In the total copy number estimate of this software, to our understanding the CNN-LOH does not register as a CNA deletion. (we did not consider the

quantifications of minor/major allele by ASCAT). Thus our CNA loss state indeed implies deletions and not copy number neutral LOH events.

(2) The Hartwig Medical Foundation and the MMRF data (~4800 and ~700 samples, respectively): there we used calls from the PURPLE tool that draws on WGS allele dosages. We processed the gene-wise min/maxCopyNumber output variables from PURPLE, which denote the low/high boundaries for the total number of copies of a gene (as with ASCAT, we do not use the minorAlleleCopyNumber and majorAlleleCopyNumber features in our study). Thus our “CNA loss” state implies that the total number of alleles is <2 , meaning that we are not counting CNN-LOH in these datasets but actual deletions.

In summary, we did not treat the CNN-LOH cases as deletions, as we measured the overall decrease in DNA copy number. Therefore our detected positive association between selected mutations in OGs and CNA deletions in OGs did not arise from the CNN-LOH. We now make this explicit in the Methods section. There is a *bona fide* increase in selection due to deletions in OGs; these events increase the relative dosage of the mutant allele compared to the wild-type allele, which gets deleted.

Another major concern regarding the model is that it does not take into account the clonality of mutations. Especially in WGS and WES samples, most of the mutational burden is usually contributed by passenger mutations. While the authors have focused on non-synonymous variants, still I would expect the majority of them to be subclonal, and a confirmation of that is found in the VAF spectra the authors present in the paper, where adjusted VAF values < 0.5 are strongly supported. The current common interpretation of a variant under selection is that it has the power to drive cancer growth, and it is thus expected to be found at high frequency. I think the authors' estimates of gene selection coefficients might be dramatically confounded by the presence of large numbers of subclonal passenger mutations, especially because they use the number of mutations as a proxy, rather than their frequencies.

If we understand this comment well, the worry is that subclonal mutations, which are relatively numerous and also under weaker selection overall, might influence the analyses that generated our key results (the TSG*gain interaction and OG*loss interaction).

We do appreciate it is possible that these lowly-selected subclonal mutations “dilute” the signal by nearly-neutrally accumulating both in the central gene (OG or TSG) and in the neighboring genes, thereby our analysis might be biased conservatively. However, it does not seem likely that converse could be the case, and that these (presumably lowly-selected) mutations could create spurious, false-positive associations. The neighboring-genes control would adjust for effects of additional subclonal mutations, and we think confounding is unlikely.

To empirically test to what extent the subclonal mutations might be affecting the signal-to-noise in our analysis, we exclude subclonal mutations using a heuristic (purity-adjusted $\text{VAF} < 0.25$) and repeat the analyses testing for (a) TSG mutation*CNA gain interaction, (b) OG mutation*CNA loss interaction.

The new Supplementary Fig. S14a compares the distributions for effect sizes for the OGs and TSGs in different CNA states (3 columns), in the original analyses (row "Original"), versus the subclonal-filtered analyses (row "VAF filtered"). It can be appreciated that the effect sizes are fully conserved upon removal of subclonal mutations: the median effect size for the OG*loss interaction was 0.20 in the original (FDR = 6.9×10^{-4}) and 0.10 in the VAF-filtered analysis (remains significant at FDR = 9.6×10^{-3}), while for the TSG*gain interaction the median effect size was 0.12 in the original (FDR = 0.02) and 0.12 in the VAF-filtered analysis (FDR = 3.1×10^{-4}). Therefore, our principal results do not appear qualitatively affected by subclonal mutations, and we added mention of this to the Results section of the manuscript.

In order to reach adequate statistical power, the authors assembled a cohort of ~18.000 samples from multiple studies. Whereas in the regression model they take into account the possible effect of different data sources using separate terms, I think it might be useful to check a posteriori whether their results are validated separately across the various cohorts.

We thank the reviewer for highlighting the comprehensiveness of our global data set, and the adjustment for differences between the constituent data sets in the regression analysis.

Here we split the TCGA (exomes) from the representatives of other major datasets (Hartwig, PCAWG - both WGS) and analyze each separately, testing for TSG mutation*CNA gain interaction and OG mutation*CNA loss interaction. The new Supplementary Fig. S14b shows the distribution of selection effects for these different cohorts, which appear broadly consistent. In particular, the median effect size for the OG*loss interaction was 0.197 in the TCGA (FDR = 5.54×10^{-3}) and 0.195 in the Hartwig+PCAWG (FDR = 4.5×10^{-3}) dataset (in the "Original" dataset it was 0.20, FDR = 6.9×10^{-4}), while for the TSG*gain interaction the median effect size was 0.13 in the TCGA (FDR = 5.4×10^{-3}) and 0.04 in the Hartwig+PCAWG (FDR = 0.179) analysis (in the "Original" dataset it was 0.12, at FDR = 0.02). Our principal results would therefore be largely robustly observed regardless of which cohort is considered. We note that reduced sample sizes and statistical power do result from splitting up cohorts like this. This factor also does not permit us to look at smaller cohorts separately. We would also like to remind that our original analysis did validate the principal findings in the independent GENIE cohort that used panel sequencing (see Supplementary Fig. 10).

Additionally, the authors do not mention any a priori check on the quality of their samples: I expect the assembled cohort to be widely heterogeneous in terms of sequencing coverage and sample purities, contributing very differently in terms of statistical power. For example, a sample with low coverage and purity would result in the exclusion of lots of low-frequency mutations under neutral or even negative selection. I also noticed that the presented VAF spectra have, in the majority of cases, support from values greater than one. Since it is impossible that a variant is present in more than 100% of the tumour cells, there must be errors in a number of purity estimates. I think it might be useful to quality-check purity estimates and, if needed, exclude unreliable cases from the cohort.

We appreciate the reviewer's comment that the robustness of our analyses -- as with any cancer genomics study — might be affected by variable quality of samples; indeed purity is an important quality-control issue when sequencing tumor material. Low purity decreases the power to identify mutations and CNA, which affects mainly the subclonal mutations (which are, as the reviewer mentions, less important for our analyses). In the first instance, we would expect our results to be not affected by differences in purity; as noted above, this is because we employ the neighboring-genes to derive a mutation rate baseline, and these would be equally affected by the low purity, a global property of the tumor sample, as the central (query) gene.

To test this empirically, we repeated our main analysis after excluding some low-purity samples , and thus potentially problematic samples (<40% estimated purity); we also exclude those where we suspect an inaccurate purity estimate (i.e. exhibiting a certain number of mutations [>10%] with apparent VAF>100%, or with aberrant purity estimates above 100%) and we exclude samples where purity estimate was not available. This excluded ~18% of all samples from the datasets for which purity data was obtained (and this additionally excluded CPTAC3 cohort completely because we did not have purity data for it). As in the analyses above, we check the distribution of selection effects on the TSG mutation*CNA gain interactions, and on the OG mutation*CNA loss interactions. The distributions of effect sizes for the purity-filtered versus the original data are shown in the new Supplementary Fig S14a; qualitatively these results appear similar. The median effect size for the OG*loss interaction was 0.20 in the original analysis (FDR = 6.9×10^{-4}) and 0.10 in the purity-filtered analysis (FDR = 0.149), while for the TSG*gain interaction the median effect size was 0.12 originally (FDR = 0.021) and 0.12 in the purity-filtered analysis (FDR = 6.57×10^{-4}). This demonstrates that our main findings were not qualitatively affected by the inclusion of some unknown-purity or lower-purity samples. We have now mentioned this in the Results text.

On a related point, we have also revised the VAF analysis to include additional data sets, which should provide more precise results. In particular originally we had VAFs for the TCGA cohort, but now as we collected purity data also for the other cohorts (PCAWG, Hartwig, MMRF and POG570) these additional tumor samples were included in the VAF studies in Fig. 4. Number of tumor samples with VAFs increased 7,382 to 15,248 in the revised analysis.

From the authors' analysis of negative selection, it is not clear to me how excluding hotspots from genes should help in revealing negative selection acting on nearby spots. Assuming infinite size and a uniform rate of neutral mutations on the genome, I would expect positively selected regions to decrease the mutation rate of nearby regions, with the effect of masking neutrality as negative selection.

Indeed, this is important to clarify better. As the reviewer notes, removing sites harboring recurrent mutations may generate the appearance of lower mutation rates, even under neutral evolution. This will become an issue when the mutation burden is very high and/or cohorts are very large, such as to generate recurrent neutral mutations at random; in practice, this is fairly rare for the majority of cancer types.

Crucially, the main result of our analysis should not be affected by this factor, since we remove the recurrent/hotspot mutations equally in the driver genes, as in the set of random genes (non-drivers, non-essential) that are used as a mutation rate control throughout our study. (the “permissive hotspots” method we employ is fully data-driven i.e. does not rely on a predefined set of known hotspots, and can discover hotspots for any gene). Under neutrality, recurrent mutations would be removed from both driver genes and control random-genes, and thus the analysis would not register a decreased mutation rate in driver genes if there was no selection.

Consistently with these randomly-occurring neutral hotspots being quite rare in our study, we do not observe a negative selection-like signal in the random-genes control set (unlike is the case for oncogenes). The median of the distribution of effect sizes of random-genes is ~ 0 after removing the permissive hotspots, same as before removing hotspots, see Fig. 1D (in contrast to result for oncogenes).

The above suggests that our hotspot detection/removal, at mutation burdens in the current dataset, does not generate spurious signatures of negative selection in the majority of genes. Even if some genes with such spurious hotspots existed, a comparison of driver genes (under negative selection) with the random-genes (not selected), as performed in Figure 1D, would control for that.

We would also like to mention that the signal of negative selection in oncogenes is actually stronger in the noncognate oncogenes (see our response to reviewer #3 below) i.e. the ones with fewer hotspot mutations to remove, than in the cognate oncogenes, which have more hotspot mutations to remove. Thus, hotspot removal does not seem to generate the signal of negative selection.

In addition, I have some minor comments:

I think the Methods section is not detailed enough to be clear and easily followed. For instance, the authors should define the domain of every variable and coefficient in the model. The specific statistical tests used for different variables (e.g. selection coefficient, median adjusted VAF) should be reported and motivated.

We have added detail to the Methods section, with regards to definition of variables (esp. the key variable describing copy number), and which statistical tests were used. We also made some language changes to improve clarity (e.g. reducing jargon).

The authors exclude outlier genes based on a threshold of a score defined as the log ratio of the number of mutations per base. Is the choice of the threshold arbitrary? Are their results robust upon changes in its value? How many outliers are they excluding from the analysis?

The threshold for outlier removal from gene neighborhoods has been set to strike a balance between retaining enough number of nucleotides in the neighboring genes, and removing genes that do not match the central gene well. Overall this is a conservative setting, favoring a close match by requiring the $\ln(\text{OR}) < 0.2$ we removed 44% of the genes from neighborhoods. We do think that changing this threshold (thus including fewer or more genes in each neighborhood) would not materially affect the main findings, which concern contrasting different CNA states. This is because the neighborhood is the same across the CNA states being concerned and even if some biases exist (which we think do not, based e.g. on distributions of effect sizes in random genes being centered at zero) they would cancel out in these analyses.

The variable Q3 is introduced in the main text without any previous definition.

We thank the reviewer for pointing this out and have added a definition; Q3 is the upper quartile.

For the determination of cognate gene-cancer type pairs, the authors have computed selection coefficients across all copy number states. What is the reason for excluding this parameter, given that it represents a major contribution in their model of selective advantage? For example, a certain cancer type could be related to a greater number of mutations on a gene due to a greater exposure to copy number alterations in certain chromosomes, rather than to the cancer type-specific positive selection of that gene.

We thank the reviewer for bringing this up. In the initial analysis to classify gene-tissue pairs into cognate (driver) and noncognate (presumably not a driver), we indeed integrated across all CNA states. This was for two reasons. First, for consistency with methods for determining driver/non-driver genes in prior work. We do not think our approach would amount to excluding any parameter. Rather, this is simply the way driver genes are identified in various recent studies to identify driver genes from cancer genomics sequencing (i.e. interactions of mutations and CNA are not considered, effectively integrating across all CNA states like we did). Second, this CNA-naive classification into cognate and noncognate sets was in order to avoid any circularity with our subsequent analyses, which do separate the genes by CNA state. In them noncognate genes may be used as a negative control for the mutation-CNA interactions (this is particularly the case for the validation analysis in the GENIE data, which is panel sequencing so the “random genes” set is not available, hence the noncognate genes need to be used).

Regarding the specific point “*a certain cancer type could be related to a greater number of mutations on a gene due to a greater exposure to copy number alterations in certain chromosomes, rather than to the cancer type-specific positive selection of that gene*” -- we understand there is a concern that there might be an association between CNA and mutation rates at a gene, which is due to neutral increases in mutation risk instead of selection. Our MutMatch framework (whether incorporating CNA labels, or not as in the initial analysis) would account for this. A gene would not be linked (related) to a cancer type by our method because of concomitantly increased rates of mutations and CNA at a locus in a tissue. This is because the

MutMatch neighboring-genes mutation rate baseline accounts for neutral variation such as this hypothetical example (we note that the infrequent cases where neighborhoods are heterogeneous by CNA state are excluded from analyses, ensuring the neighbors match the central gene by CNA in all analyzed samples).

We have added an extensive clarification on the above matters to the manuscript in the Methods section (shown with Track changes as all other edits).

The dN/dS method is mentioned by the authors as a method for estimating selective advantage of mutations that, differently from the presented MutMatch model, does not take CNA states into account. I think a comparison between dN/dS MutMatch estimates of selection coefficients might be interesting. In particular, the authors could check whether there are differences in dN/dS values between genes harbouring mutations and CNA with respect to diploid cases (in analogy with Figure 3).

We think the reviewer may be referring to our mentioning of state-of-the-art tools for detecting somatic selection: MutSigCV and dNdScv. These methods model baseline mutation rates from covariates -- "CV" gene expression, replication time etc -- and thus would indeed be confounded by CNA when estimating baseline mutation rates. To clarify: the widely used tool dNdScv does indeed use "dS" rates (which intrinsically accounts for CNA -- however dS are very sparse and noisy) to supplement its analysis however the covariate part of the dNdScv model would be naive to CNA.

The dN/dS evolutionary biology method *sensu stricto* would control for the confounding effects of CNA on mutation rates. However this dN/dS is less useful for cancer genome analysis in practice, because the number of S (synonymous) mutations is low and analyses are underpowered. (This large difference in power between cv methods and dN/dS was discussed in the Martincorena et al. Cell 2017 study, and shown for their implementation of the dNdSloc method ["loc" = strictly dN/dS], showing results in their Fig. S2). Our MutMatch approach alleviates this bottleneck in power, because the number of neutral sites in neighboring-genes, which we use as a baseline, considerably exceeds the number of synonymous sites within the gene.

We performed a benchmarking exercise comparing methodologies, between our MutMatch and dNdScv tool, which models mutation rates from a dNdS test ("dNdSloc") and can supplement this with modeling by covariates (cv). This benchmark aims to identify known driver genes listed in Cancer Gene Census (CGC) from point mutations. We have now included this as Supplementary Fig. S2c, where we see competitive AUCs: MutMatch AUC=0.676 and dNdSloc test AUC=0.684. The dNdScv tool, which supplements dNdS with covariate modelling achieves 0.722 in this test. As an additional test for sensitivity to low number of mutations, in Fig S2e we see an analysis of consistency of results between randomly sampled, half-size subdatasets, which is higher for our MutMatch-neighbors method (R=0.79) than it is for dNdScv (R=0.68). This indicates a robustness of MutMatch to noise resulting from low mutation counts, plausibly because it uses the neighboring genes' more abundant neutral sites compared to dNdS that relies on comparatively rarer synonymous mutations.

Regarding the additional test the reviewer suggests, computing dN/dS alone in the gained (or lost) segments to compare against MutMatch. We think this is probably not feasible at current dataset sizes for reasons mentioned above -- there is too few synonymous mutations for the dN/dS to work well (see comparatively weaker performance of dNdScv on the random sampling test above), and further splitting data by CNA state makes this problem of dNdS more acute. If really needed, this might be tested for some very common CNAs like 1q but we expect hardly any significant hits from dNdS that way.

Reviewer #2 (Remarks to the Author):

Reviewer #3, expertise in cancer genomics and evolution and selection (Remarks to the Author):

Besedina & Supek conduct a rigorous investigation of patterns of selection of oncogenes and tumour suppressor genes across cancer types and varying copy number states. A statistical method called MutMatch, which estimates selection by comparing exonic mutation rate with that of surrounding mutations assumed to be under neutral selection, was applied to over 17,000 tumour samples. Indications of negative selection were often observed for somatic mutations outside of hotspot regions, suggesting certain levels of gene essentiality for oncogenes. Additionally, selection patterns in line with somatic copy number aberrations were analysed to discern additional driver events which may arise from either one- or two-hit events. Intriguingly, the authors found that two-hit events including point mutations appeared to be associated with both copy number gains and losses, suggesting alternative mechanisms of gene loss which may allude to previously overlooked therapeutic vulnerabilities.

Overall, this paper presents a very interesting rethink of established truths about genetic selection, with wide-ranging implications. The figures are broadly very clear, although it would be worth ensuring that they are properly referred to within the text and that the captions are more explicit. If the manuscript suffers from one issue, it is that a fair amount of the results appear slightly under-powered, which manifests as insignificant results presented as substantial (e.g. lines 562-564), or results with high significance which appeared marginal. However, the authors are open about these limitations in the text, and it could be argued that with some additional explanation these issues do not detract from the broader messages of the paper.

We thank the reviewer for a summary of the main contributions of the study, and for highlighting the rigor, high interest, and wide-ranging implications.

We have now made minor edits to the text to ensure the figures are correctly referenced in the text and we aimed to add more detail to the captions, as suggested. We do agree with the reviewer that additional statistical power would be helpful, particularly in terms of being able to make conclusions about CNA/mutation interactions on the level of individual genes (these findings are more robust at the level of gene groups - TSG, OG - where the distributions of effect sizes across many genes are studied, affording more statistical power). Analyses of genetic interactions such as ours may indeed require large sample sizes, and future studies with additional WGS/WES will alleviate this bottleneck.

In relation to this point: inspired by Reviewer 4/5 comments (see below), we have revised one aspect of the MutMatch methodology, such that a randomization test is now used to derive p-values for individual genes, rather than directly using p-values resulting from the regression (which were obtained via a statistical test on the coefficient and S.E.). Already in the original MutMatch method, this randomization was used to de-bias estimates of selection of individual genes (i.e. regression coefficients; as mentioned above our tests on groups of OG or TSG were operating on distributions of such estimates, and are unchanged here). In the revised study, we apply a similar procedure also to MutMatch individual gene p-values (Methods), which were very conservative originally (deflation of p-values, $\lambda \ll 1.0$), but are now adjusted (inflation factor $\lambda = 1.011$, see new panel in Supplementary Fig. 2d). This very good calibration of p-values results in several additional significant genes in the hotspot-tests in Fig. 2, and in the CNA-mutation interaction tests in Fig. 3 and Fig. 4.

(We'd mention at this point that this calibration of individual genes p-values has caused our groups of cognate vs non-cognate genes to be somewhat changed from the original study. This does slightly alter the group-comparison results throughout the study, although qualitatively the findings reported remain as earlier. We have updated the text/figures with new effect sizes and significance).

Although the insights provided by the manuscript are valuable to the cancer evolution community, some concerns need to be addressed:

Overall, the Results part was rather difficult to read due to the sheer number and complexity of the analyses performed. It would be helpful if the authors tried to make the text easier to read for a lesser specialised audience, e.g. by introducing the question that they aim to answer at the beginning of each subsection and concluding each subsection with a sentence that summarises the main messages taken from the respective analysis. Shortening the main text could also help.

We fully agree and have made many minor edits throughout the text, to improve the ease-of-reading especially for non-specialists. The edits are too numerous to list here but are shown as "track changes" in the attached revised manuscript file. We also hope that the new schematic in Supplementary Fig. S1 (showing the logic behind our methodology) helps the manuscript become more accessible.

In addition we did shorten the text by relegating some parts to a Supplementary Text, and just briefly mentioning them in the main manuscript, for instance:

1. The estimates of negative selection upon essential genes in the CEG2 set, as well as associations with LOEUF (popgen constraint) and CERES (genetic screens) scores. This negative selection on essential genes is modest, in accordance with expectations from previous work; this result supports that our method is sensitive but it is peripheral to the main points of the study (CNA-driver gene interactions) so it is moved to Supplementary Text.

2. The analysis of how focussing on known positively-selected hotspots can implicate a known driver gene in additional cancer types, discussed for examples of EGFR gene and breast cancer, and ERBB2 gene and several cancer types (kidney, prostate, HNSC).

While these findings are in our opinion interesting (so the main results for these 2 points are still shown in Fig. 2), we see that moving the lengthy accompanying text to a Supplementary file will provide a better focus on the principal findings regarding CNA interactions with OG and TSG drivers.

I would also recommend reviewing the orders of figures and labelling within the text. At points the figures are not referred to in order, making the text somewhat difficult to follow, whilst other figures (e.g. Fig. 5c) are not referenced at all (likely a mislabelling). Can you also please make sure all figure captions are sufficiently explicit to be understood without referring to the main text or the methods (e.g. what does “bin” mean in Fig 2E?).

We have made sure figures are referenced correctly, and edited the figure legends aiming to add detail and improve clarity. In Fig 2e, we renamed the “bin” to “decile”. Indeed one of the mentions of Fig. 4c was a mislabelling and is now changed to Fig. 5c.

Equations (1) and (2): For the MS96 variable, $m_i \mu_i$ should be replaced with $m_j \mu_j$

Thanks for spotting an error in our formulae. But the error is slightly different however: indeed the $m_i \mu_i$ (controlling for the 96 mutation spectrum) should remain as “i”, however the following term “z*beta” (controlling for the batch effects or confounders) had the “i” and “j” swapped, the “z_i” and “beta_i” are now changed to “z_j” and “beta_j”.

Lines 329-331: The $\log_2(\text{mutation_rate}) < -0.2$ threshold for negative selection feels remarkably lenient, especially regardless of significance. Why was this threshold selected, and how can this be discerned from Figure 2A-B? Additionally, it would be worth commenting on the possibility that this is neutral selection.

Indeed, we agree that using this threshold to declare negative selection would be extremely permissive. We did not intend to convey the genes were significant by this statistic, which might thus create confusion and so we removed it. The point was to compare the distribution of negative selection effects between different groups of oncogenes (cognate or non-cognate OG) versus the

control, random genes. We have now revised this text of the Results to contrast median versus median of the distributions of OGs and random genes, as we do with most other group-wise comparisons of effect sizes in the text.

Lines 333-341: Of the two mechanisms underlying negative selection pressures, the second one (tumours depend on oncogene function even if it is not a tumorigenic driver) feels far more logical. Have these mechanisms been reported at all in the literature, or in this case are they primarily hypothetical/speculative?

We are not aware of mentions in the literature about these two mechanisms of negative selection acting upon oncogenes -- first, preventing the “undoing” of the positively selected mutation already present on the oncogene, and second, preventing the acquisition of LoF mutations in a non-driver oncogene. We agree with the reviewer that the second mechanism, which we tentatively named “non-canonical oncogene addiction” makes more sense; in our mind this would be because, for the first mechanism to occur the OG would already need to bear one GoF mutation as a prerequisite, and because of requiring two hits on the same locus this might be rare in practice. Here, we’d note that in our revised analysis (which now has somewhat refined sets of cognate and non-cognate genes -- see response to Reviewer 5) indeed the second mechanism is more supported in the data than the first one.

To our knowledge these mechanisms are currently hypothetical. Future work will validate and quantify occurrence of these mechanisms in tumors; e.g. we see that phasing somatic mutations to paternal vs maternal alleles using long-read sequencing could ascertain whether the positively-selected and the negatively-selected mutation occur on the same allele of the oncogenes, clarifying these mechanisms. Also, experiments using allele-specific gene editing would be helpful; this is challenging since GoF mutations also need to be introduced but new technologies such as prime editing can promote this efficiently.

We now edit the text to make explicit that the mechanisms are hypothetical, that the second mechanism is more supported in our genomic data, and made minor edits to this section to make the message clearer overall.

The section entitled 'Essential genes show modest signatures of negative selection in somatic cells' does not highlight particularly exciting or strong results, and doesn't appear to add much credence to previous results. Perhaps it could be shortened. Also, it would be helpful to make it clear in the text that the data analysed here is in cell lines, as someone who is not familiar with CERES scores might be misled to think this is still data from patient samples. Introducing more clearly the question and the motivation to look at cell lines to study essentiality would help readability.

We do agree that our report of (modest) negative somatic selection on essential genes may not be the highlight of this study, since prior studies did report cancer-essential genes (see answer to the point above and studies cited), and some association of those with cell-essential genes was

noted. As mentioned above, we have now relegated this section to a Supplementary Text, leaving only a short summary in the main manuscript. We also clarify that CERES scores are from cell lines, not from patients, and clarified that these are intended to be cell-essential genes.

The authors mention that the CERES score is more correlated with their estimates of negative selection than the LOEUF score (lines 368-370). How is this determined? The correlation coefficient of CERES vs LOEUF (Fig S4D) appears greater and more significant than CERES vs pan-cancer selection in the diploid state (Fig S4A).

Indeed, CERES score (from cell line screens) is very modestly correlated with pan-cancer selection ($R=0.043$), while the correlation of LOEUF score (from population genomics) vs pan-cancer selection, although with similar magnitude is negative ($R=-0.036$). This implies that the low-ranked LOUEF genes (and thus essential in the germline) are in fact under less strong negative selection in the soma, which means that the score is anticorrelated with estimates of negative selection and correlation of CERES is higher — although, as the reviewer notes, still modest, which we clarify in the text (now moved to Supplementary Text S1).

The differences in CERES scores between cancer genes and random genes (Fig S4E) is incredibly marginal and this should be highlighted. However, it could be argued that this does not detract from the results presented in Fig 2, due to both the cancer type-specificity of negative selection demonstrated in Fig 2D and the differences between cell lines and patient samples.

Indeed we agree this difference in CERES scores is very small, and we now have written this explicitly in the text pertaining to this analysis of gene essentiality, stating that cancer genes were modestly more essential” (discussion of this has been relegated to a Supplementary Text S1, see above).

Lines 507-508: it may be worth specifying that, according to Fig 3A, the increased selection of OG mutations upon CNA deletion occurs only for nonsynonymous mutations in permissive hotspots. Does this indicate that these mutations are likely gain-of-function?

This is indeed the case, the mutations under positive selection upon CNA deletion in hotspots for OGs would be likely gain-of-function mutations (as inferred from that hotspot mutations in OGs are normally gain-of-function). We now mention this in the manuscript text in the location suggested.

The authors have combined a range of publicly available cohorts in their analyses, most of which are from primary tumours and some, like the Hartwig dataset, from metastatic samples. It would

be interesting to see if the same signals of selection are observed both in primary tumours and metastases or if there are differences.

We agree it would be interesting, and we have now performed this analysis. We urge caution in interpreting it, though, because metastatic-vs-primary status in our dataset is confounded with the cohort: two major constituent data sets in our analysis, the TCGA and the HMF (Hartwig), are almost all primary and mostly metastatic, respectively. Thus any differences observed between primaries and metastatic, might reflect other biological or technical differences between the two cohorts, and any results of this analysis are very tentative. In the new Supplementary Figure S14c, we compare the OG with CNA gain or loss associations, and TSG with CNA gain or loss, contrasting the primaries versus the metastatic cancers. Here the OG*deletion interaction has an overall similar effect in the primary with a slight increase (median of effect size 0.14, FDR = 0.069) as in the metastatic (median of effect size 0.11, FDR = 0.068). The TSG*amplification association is more strongly seen in primaries than in the mets in terms of median (although also in the mets some TSG do exceed the background distribution of random genes; Supplementary Fig. S14c).

Reviewer #4 (Remarks to the Author):

Reviewer #5, expertise in cancer genomics and evolution and selection (Remarks to the Author):

This paper by Besedina et al presents a regression-based method for finding selection signals, MutMatch, which seems cleverly and carefully devised. The method is here employed, among other things, for finding SNV selection signals that are associated with specific copy number states. Application to large scale cancer genomics data points to some interesting interactions between copy number events and selection acting on SNVs. Negative selection among "out-of-hotspot" non-synonymous mutations in oncogenes is another interesting result.

We thank the reviewer for highlighting the innovative aspect of the method, as well as the interest in the reported interactions between CNA and selection on SNV, as well as in the negative selection findings.

In general, my main concern is some lack of clarity and conciseness in the presentation overall. The article is very long making it less accessible and appealing than it could be. Some careful editing is needed in my opinion, for clarity and to make it more to-the-point.

E.g. the introduction has a good discussion about confounders when detecting selection, including that fact that CNA is itself a mutation rate covariate. However, I general the introduction could be much shorter, while still retaining the useful information. There are also several typos.

We agree and we applied various minor edits (shown with track changes) to the Introduction, aiming to shorten it, improve clarity and correct typos/improve language throughout.

There is similar room for improvement and shortening throughout Results. E.g. a lot of text is devoted essentially to the fact that focusing on mutation hotspot regions increases power to detect driver events - this is not very surprising and the same should be true using many other methods. It would probably be good to put the more emphasis on the truly novel findings so they don't risk getting lost.

We agree on the above points regarding conciseness and focus on the more striking results, and have edited the manuscript text to make it shorter. Those results where the novelty/interest is not the highest have been relegated to a new Supplementary Text file, with a brief summary remaining in the main text. This refers to two parts of the manuscript (a) negative selection on cell-essential genes (as proposed by Reviewer #3 above) and (b) the positive selection in known hotspot-regions increasing power to find additional tissues for known driver genes (as proposed by this Reviewer #5).

The title is not all that clear to me, and before reading the manuscript I took it to mean that the copy number events in themselves were the drivers. Perhaps saying "generate driver _mutations_" would be clearer (although see comment below about cause and effect).

We changed the title swapping "events" to "mutations", and it now reads "Copy number losses of oncogenes and gains of tumor suppressor genes generate common driver mutations of human cancer". This is a fair description since we measure selection on mutations in particular, and the effect of CNAs modifying this selection on mutations.

In the text, there seems to be an implicit assumption (with some exception) of CNAs preceding mutations (e.g. as they "generate" common driver events, or "upon" deletion/gain). While it can sometimes in principle be possible to decipher the temporal order (e.g. above-heterozygous VAF for an SNV suggests it occurred before CNA), I don't see that this has been considered. E.g. one example is the "CNA and mutation epistasis in the same gene" effect mentioned in the introduction, e.g. where a heterozygous mutation (first hit) can precede and lead up to selection for a second hit copy number deletion event. Same for OGs and allele imbalance: "we turned to examine selection on point mutation .. upon deletions" (row 462) - what says it's not selection for deletions upon point mutations? The language may relate to how CNA is used as an independent variable in the regression, but does not necessarily reflect temporality during tumor progression.

The reviewer is correct that our CNA-mutation association analysis does not explicitly consider the temporal order of mutations versus CNA; our current MutMatch method does not distinguish between the two cases. We have edited the text to state this explicitly (in Results and in Methods).

As the reviewer writes, by checking distributions of VAFs it is possible to infer ordering of mutation *versus* CNA -- in some cases. Because we observe a significant increase in VAFs in the OG*deletion case and in the TSG*amplification case, that means that at least in some cases of these selected mutations, the mutation must have occurred prior to the CNA (mut->CNA) rather than the converse (CNA->mut). We do not formally rule out that sometimes the CNA might precede the selected driver mutation. We write this explicitly in the Results text, in both the section "Selection change upon somatic CNAs identifies additional cancer types where a gene acts as a driver" and also the "Copy-number gain associated conditional selection on TSGs and OGs confirmed in VAF analysis" section.

Regarding wording, we take care that this (including "upon deletions", "generate" as the reviewer mentions) is consistent with the temporal order that selected mutations preferentially occur before CNA; we think this is reasonable, since that this order is common would be supported by the VAF data. We have added a statement to be explicit that this wording does suggest the ordering mutation->CNA but that we do not preclude the converse ordering in some cases (see beginning of section "Selection change upon somatic CNAs identifies...").

As a side note, we would mention that this ordering also fits with the previously reported observations that CNA deletions alone (no mutations) in OGs tend to be disfavored overall, as well as amplifications alone (no mutations) in TSGs. Thus the ordering where CNA arrives before mutation is likely to create a temporary drop in fitness. In other words, after CNA occurs but before the mutation occurs, the CNA would have a fitness penalty meaning it is plausible that this evolutionary scenario would be disfavored (although not impossible).

Regarding the principle underlying the method: there are certain assumptions, e.g. "central genes are expected to be similarly affected by CNAs" and "domain-scale mutation rate is adjusted for as typically the neighbour genes will be in the same 'domain'. Additional "refinements" are added to account for cases that violate these assumptions, which is reassuring. However, is a deviation of neutral mutation rate in the central gene of interest (compared to neighbour genes) necessarily detectable? Doesn't the idea of using the "neighbourhood" in the first place stems from a difficulty in estimating neutral mutation rate in the central gene alone? Generally speaking, significant signals are all either 1) actual selection or 2) exceptions when the model fails - and latter doesn't need to happen too often for these to add up in the end.

We agree that it would be beneficial for the study to better substantiate the choice/design of our mutation rate baseline in MutMatch, which is based on neighboring genes. As the reviewer mentions, a good baseline model means that the deviations from the neutral mutation rates are detectable, but also that such deviations are actual selection, rather than some exception where the baseline fails.

We have performed various additional analyses to test the utility of the mutation rate baseline using neighboring genes (+trinucleotide stratification, as we perform), in a regularized Poisson regression as used in MutMatch.

1.

We tested the calibration of the p-values resulting from the MutMatch analysis, to ascertain that our analysis does not suffer from false positives. We visualized qq plots, considering the case of our negative control, “Random” gene set, which excludes the known cancer genes and known essential genes. A lambda value, measuring inflation on the qq plot of p-values, close to 1 implies that the p-values are well-calibrated and that there is not a global bias towards false positive hits. Indeed there was not, but we noted there was a substantial conservative bias of the method: inflation factor lambda was $\ll 1.0$ (Supplementary Fig. 2d), meaning we would expect to have many missed hits when testing statistical significance of individual genes.

We felt this conservative bias of MutMatch in individual gene testing should be improved. To this end, we applied a randomization procedure to derive empirical p-values for genes, and this indeed drastically improved the qq-plot, resulting in a very good lambda of 1.011 (Supplementary Fig. S2d). This same randomization was in our original manuscript successfully applied to bias-correct the selection effect sizes (regression coefficients; see Supplementary Fig. S2a), but originally the p-values for individual genes were directly from the regression (Z-test on coefficients). Now they are corrected to empirical p-values using distributions from simulated random data, just as the effect sizes were corrected.

This update of the method did not change our results qualitatively: (i) this update changed p-values on individual genes. Testing of statistical significance on gene groups (OG, TSG, cognate vs noncognate), a basis for central claims in the original manuscript, did not rely on these individual gene-level p-values, but instead tested for shifts in distributions of effect sizes on OG, TSG, essential genes etc. Effect size calculations did not change, because it had the randomization-assisted debiasing already in the original study. (ii) the p-values (and consequently, FDRs) on individual genes are now not overly conservative, which results in that some additional individual genes became significant in the gene-CNA interactions tests; we think this is a welcome change. (iii) because of the different significance of individual genes, the cognate vs noncognate groups of OGs and TSGs changed somewhat. We now employ a $FDR \leq 2\%$ cutoff for declaring cognate genes (earlier, it was nominally $FDR \leq 25\%$ cutoff but in reality was more stringent), so the number of cognate genes per cancer type is broadly similar as earlier, even though individual genes in the set may not be exactly the same as earlier. Because of this, there are slight changes in the boxplots showing effect sizes of cognate vs noncognate OG and TSG throughout the study (even though effect size calculation is the same).

2.

2a. As in recent studies that developed methods to identify driver genes, we considered a benchmark using the Cancer Gene Census (CGC) as a ground-truth dataset. This has a caveat that CGC is not a complete list of driver genes, and some of CGC genes might be under no selection for SNVs or CNA (e.g. translocation/fusion drivers). We compared the broadly-used

tools MutSigCV and dNdScv, which both model mutation rates from covariates (“CV”) such as gene expression levels, DNA replication time etc, against our MutMatch, which uses the neighboring-genes baselines (with refinements as described - excluding neighboring genes that are outliers by mutation rates; excluding neighboring genes in a different CNA state than the central gene). Comparing the distributions of FDR values (for MutSigCV, dNdScv and MutMatch) and selection estimates (\log_2 fold-enrichment in mutation rates for MutMatch) in CGC genes versus random genes, MutMatch has AUC of 0.676 across all cancer types, while MutSigCV scores 0.655; the dNdScv tool that combines a covariate model with a dN/dS test scores 0.722. Thus MutMatch is comparable to both MutSigCV and dNdScv in a benchmark of identifying genes under positive selection; see new Supplementary Fig. S2c.

2b. We additionally checked if the MutMatch neighbors estimates were robust to noise introduced by small sample sizes, and compared it to the state-of-the-art method dNdScv. In particular, we checked if the two approaches yield similar selection effect sizes on various driver genes in a pan-cancer analysis thusly: the ~23k cancer exomes/genomes are randomly split into two sets of ~11.5k to determine the inter-replicate correlation for the two methods; this is $R=0.79$ for the MutMatch neighboring-genes, which exceeds the $R=0.68$ for dNdScv tool estimates (Supplementary Fig. S2e). Therefore our method is somewhat more robust to effects in small datasets as the state-of-the-art dNdScv tool. Also the concordance between the MutMatch and dNdScv estimates on individual genes is high: we obtain a $R=0.52$, close to the maximum attainable value in this test given the noise ($R=0.68$ for the dNdScv self-similarity across the two half-datasets). We note that tools MutSigCV and dNdScv were not designed with the ability to test conditional selection, that MutMatch can do, and we apply that facility in our study to studying CNA-associated conditional selection (but this may also be extended to any other condition of interest).

In summary, our mutational rate baseline drawing on neighboring-genes in MutMatch performs well in the task of identifying known driver genes, does not anymore suffer from false negatives and does not/did not suffer from false positives based on qq-plots, and is robust to noise derived from limited dataset sizes. We thank the reviewer for prompting us to test and improve our method.

Related to the above: a linear model is used while mutations are count data, and there is a risk of over dispersion effects. It seems this is corrected for by a randomisation procedure for low counts. Is this procedure sufficiently validated, i.e. could it be worth investigating the relationship between mutation counts and P-values in the final results?

Regarding the linearity: we apologize for the unclear wording in the Methods section, which originally stated “*To describe the variability in mutation counts in a genomic locus, we model raw mutation counts Y using the following generalized linear model, regularizing by using a weakly informative prior distribution of regression coefficients (60)...*” Ours is however not a linear model; it is implemented in the “generalized linear model” (glm) framework which allows implementing various link functions in the regression to model non-normal data, hence the qualifier “generalized”. In this study, our model uses a log link function, which is used for regression

analysis that models count data (Poisson, negative binomial). Our model is a *bona fide* count model. We employ the implementation of the Poisson model from reference Gelman et al. (Ann Appl Stat 2008); this implementation additionally applies a regularization to coefficients, improving stability of estimates and convergence with sparse data. (On a perhaps related topic: in our response to Reviewer 1 -- please see above -- we clarified in detail that our model makes no assumptions of linearity between the CNA state and mutation rates.)

Next, as regards overdispersion: we use Poisson regression. This by itself does not account for overdispersion (or underdispersion) when deriving p-values, however as the Reviewer correctly notes this issue is now allayed by our randomization procedure. Please see our answer to the point above.

In the analyses from the original manuscript, the randomization was employed to de-bias the effect sizes (coefficients in the regression, which are log fold enrichments); upon having subtracted the randomized effect sizes, we no longer observe these biases in effect sizes at low mutation counts (shown in Supplementary Fig S2a [was S1a in original manuscript]). The main conclusions of the study were drawn on these unbiased effect sizes; we tested significance by comparing the distributions of effect sizes on OG or TSG, cognate or noncognate versus random genes in various CNA states.

We have however implemented additionally that the significance calls/FDRs for individual genes are now based on this same type of randomization. This corrected the bias successfully (see deflation in qq plots, Supplementary Fig. S2d), meaning the p-values for individual genes are now well-calibrated, $\lambda=1.011$. Based on this we conclude that any overdispersion (or underdispersion), even if present in the regression models, does not affect our inference anymore since it would have been corrected by the randomization.

Row 172: It is not clear to me how this is a "control": wasn't the very definition of a non-cognate cancer type based on the lack of selection in a given gene? Or I suppose this refers specifically to nonsense mutations, while the cognate/non-cognate classification considers all possibilities for a given gene...? Again this is probably just be a matter of describing things more clearly.

The wording may create some confusion; we thank the reviewer for pointing this out. The understanding is correct: cognate vs non-cognate definition was indeed determined from selection on either missense, or nonsense, or on any nonsynonymous mutations (measured across all copy number states jointly). Then, observing a strong enrichment specifically in nonsense mutations for a cognate gene is not trivial, because the assignment as "cognate" may have resulted from the missense-only-test, or from the all-nonsynonymous-test. However because nonsense mutations were present in the determination cognate/noncognate, it is perhaps not fair to call this a control. We now state "as expected" instead of "as a control".

There should be some relevant references for EGFR mutations in breast cancer, including the TCGA marker paper.

In reference to the TCGA breast cancer marker paper (TCGA, 2012 Nature) and EGFR in breast cancer, we added the following sentence “*In breast cancers, EGFR is recognized to be amplified and overexpressed (TCGA, 2012, Nature), and examples of somatic mutations in EGFR were discussed in context of druggability (TCGA, 2012, Nature), however the driver status of EGFR mutations in breast cancers was not ascertained in a large-scale genomic analysis (Bailey et al 2018, Cell).*” Note that this sentence is in Supplementary Text S1, where we moved the discussion of genes with selected hotspots in various cancer types, to shorten and focus the main text as per reviewer suggestions.

Unless I missed this, to reduce concerns about possible confounding effects (factors not considered by the model), many of the analyses would benefit from inclusion of “non-driver genes” as controls (as a group, selection should be minor).

Indeed, we do have this group of non-driver genes as a control in many analysis, which is in the manuscript called “random genes”; these were defined so as to exclude driver genes (permissively defined as genes listed in MutPanning or CGC) and also cell essential genes. This was described in the text of Results, and now we further added mentions also to Fig 1, Fig 3 and Fig 4 legends (various analyses therein already used these random genes as a baseline) that the “random genes” exclude known drivers and so serve as a negative control. Fig. 1B showing the analysis that defines the cognate and noncognate genes additionally shows the random genes control set.

All in all, this is at its core an interesting and comprehensive piece of work, which would have benefited a lot from having a strict word limit during writing thus forcing a more to-the-point and accessible presentation.

Thank you again for highlighting the interest in this study and its comprehensive approach; we hope that upon the edits described above (e.g. relegating some parts to a supplementary text) the manuscript has become more focused, improving presentation.

Minor:

Some specific examples related to text clarity:

Abstract: "Consistently, focussing on known positively selected regions identifies additional tumor types where an oncogene is relevant". By this, I assume the authors mean to say: 1) positive and negative selection occurs in different regions within oncogenes, and 2) only considering the former regions give better signals. This is not clear and requires some thinking. Missing is the information is that the two signals (positive and negative) arise in different regions (within a specific gene region...). Obvious only once you understand it.

We have added to the abstract the mention that the positive and the negative selection affect different regions (segments) of oncogenes.

Similarly, the logic behind “conversely” in the abstract (also relates to title) is not clear to me: there is increased selection on oncogenes both following (?) gain and deletion. Then “conversely” there is increased selection (or at least mutations) in TS genes during gains? Is there really an opposite relationship here?

Indeed “conversely” is a confusing choice of word, we have changed this to “similarly”.

REVIEWER COMMENTS

Reviewer #1 (Remarks to the Author):

The authors have addressed all the points of concern that I reported in the previous round of revisions, and made several edits to the main text.

Whereas I am still convinced that the message delivered with this work is of interest for the scientific community and for the readership of Nature Communications, overall I find that the authors' replies to most of the issues that were raised are insufficient, a major point being that they did not report the location of the text edits they made, making a final decision very hard to take.

For example, regarding my question about purity estimates and exclusion of cases with aberrant purity, they state in their answer that "The median effect size for the OG*loss interaction was 0.20 in the original analysis (FDR = 6.9×10^{-4}) and 0.10 in the purity-filtered analysis (FDR = 0.149)" and that they "mentioned this in the Results text". However, in the interested section "Positive selection on point mutations is increased by deletions in OGs and gains in TSG" of the main text, there is no mention to this, nor could I find elsewhere any report of the specified effect sizes or FDR values. In my opinion this is a serious issue, because such a strong change in effect size (half of the original) and FDR (from <0.001 to >0.1) as due to a more restrictive selection of higher quality data indicate that results may strongly be biased by noisy data.

In addition, when addressing my point about insufficient clarity in the model description, the authors state that they "have added detail to the Methods section, with regards to definition of variables (esp. the key variable describing copy number)". I reported this as a minor comment in my previous revision because I thought that would be very easy to address. However, the extremely poor description of this section is, in my opinion, a strong deficiency of this work, since all the results reported are based on the application of the model.

In particular, it is clear that the authors used a generalized linear model to fit mutation counts Y , using selection coefficients ω and δ coupled with an indicator variable that discriminates target and baseline genes, plus other covariates. The clearer definition of the variables involved, their domain and any condition on them seems to be missing. To be more precise, I think the authors should explain in the text what identifies a mutation for which counts Y are fitted, whether Y is a scalar number, a vector or a matrix, what is the dimension of it, what kind of variables are the "mutation type" m_i , what kind of variables are the μ_i , defined as the "corresponding effects", whether these variables are categorical, integers, real numbers positive or negative, what is the range of the index i . The same holds for β and ζ . How is the number of nucleotides at risk $\log(r)$ defined? How is the positivity of the

log-counts on the left side of the equation preserved, when on the right side we have negative terms (the selection coefficient can be negative)? In the definition of the outlier score S , M is defined as the “number of mutations observed in a gene”, and L the “gene length”, but what are S_x , S_y , M_x and M_y ? In addition an example of a single fit, for instance for a gene of interest, would be very helpful to understand the model.

The level of approximation in the model description is reflected also in the poor software documentation, that is restricted to a very short README.

The illustration of the model in Figure 1a would also be much more useful to understand the methodology, if sufficiently improved: the blue rectangle in the legend called ‘baseline mutation rate’ is identical to the ones indicating the exons, which makes it of difficult interpretation. There are two windows zooming in the top curve in different regions: is the different annotation (“under positive selection” and “no selection”) due to a difference in the baseline mutation rate, in the selected gene’s rate or to the different genomic regions highlighted?. Is the same gene represented? A third window indicates the “comparison controls for trinucleotide” contexts: what is the difference with the previous two windows? Is there any relation with the fact that we are here “under negative selection”?

I also think it would be valuable to report more summary statistics concerning the pre-processing and selection of data: what is the average number of neighbouring genes per target gene? Is it comparable from one gene to another and across chromosomes? What is the effect of the reported filtering strategies (e.g. removal of cases where the same CNV is not shared between neighbouring genes, removal of outlier genes) on these statistics? This should be clearly described and reported in the text. Examples on known and relevant genes (e.g. TP53, KRAS?) would be extremely helpful.

Overall, I would invite the authors to clarify all these points so as to ensure that their results are more easily reproducible and the conclusions they draw are not affected by major biases.

Reviewer #2 (Remarks to the Author):

Reviewer #3 (Remarks to the Author):

The authors have addressed all my comments and the manuscript is much improved as a result. I would only like to point out a few typos that require correction in the text:

49: missing closed bracket

58: remove 'n'

75: below 'the' threshold of detection

95: 'an' -> 'any'

205: 'considered' -> 'we considered'

231-232: remove italics

314: remove additional full stop

325: remove additional comma

328-329: fix formatting

586-588: fix punctuation

I congratulate the authors for this excellent study.

Reviewer #3 (Remarks on code availability):

Detailed documentation is provided for the code, which is very helpful. Whilst I was able to install the `mutmatch` package, I struggled to run the example code for the `get_selection_estimates_neighbors()` function, firstly due to issues with accessing the example files within the library, and secondly, after manually loading the data, through a syntax error. However, whilst this should be checked again, I acknowledge that this may have been an issue of compatibility on my end.

Reviewer #4 (Remarks to the Author):

Reviewer #5 (Remarks to the Author):

The authors have done a serious effort to improve the general accessibility and readability of the manuscript, and the presentation is much better now in my opinion. My other concerns have been addressed in an ambitious way and the whole study has generally been improved, including a new randomization/P-value calibration procedure. I have no further concerns and wish to congratulate the authors on an impressive piece of work.

Reviewer #1 (Remarks to the Author):

The authors have addressed all the points of concern that I reported in the previous round of revisions, and made several edits to the main text.

Whereas I am still convinced that the message delivered with this work is of interest for the scientific community and for the readership of Nature Communications, overall I find that the authors' replies to most of the issues that were raised are insufficient, a major point being that they did not report the location of the text edits they made, making a final decision very hard to take.

We thank the reviewer for stressing the broad interest of our study, and that all the points of concern were addressed. In this iteration, we have made a variety of minor edits to the text to clarify the new points raised by the reviewer, as detailed below. (We do note that already in the previous submission, there was a “track-changes” document attached, where all edits were shown in a different color.)

For example, regarding my question about purity estimates and exclusion of cases with aberrant purity, they state in their answer that “The median effect size for the OG*loss interaction was 0.20 in the original analysis (FDR = 6.9×10^{-4}) and 0.10 in the purity-filtered analysis (FDR = 0.149)” and that they “mentioned this in the Results text”. However, in the interested section “Positive selection on point mutations is increased by deletions in OGs and gains in TSG” of the main text, there is no mention to this, nor could I find elsewhere any report of the specified effect sizes or FDR values. In my opinion this is a serious issue, because such a strong change in effect size (half of the original) and FDR (from <0.001 to >0.1) as due to a more restrictive selection of higher quality data indicate that results may strongly be biased by noisy data.

In the main text we had previously included a sentence “Removal of subclonal mutations, of low-purity tumor samples, and separation by cohort did not qualitatively change this TSG-CNA gain association...”, which pointed the reader to a Supplementary figure showing these results regarding purity filtering (Supplementary Fig. S14a).

Now, we additionally include the effect sizes and FDRs in the main text, which now states: “In specific, the median effect size for OG-loss interaction was 0.20 in the original analysis (FDR = 0.069%), 0.10 (FDR = 14.9%) after removing low-purity tumor samples and 0.10 (FDR = 0.96%) after removing subclonal mutations. For TSG-gain interactions the median effect size was 0.12 in the original analysis (FDR = 2%), 0.12 (FDR = 0.0657%) after removing low-purity samples and 0.12 (FDR = 0.031%) after removing subclonal mutations.” As before, we have supplied a track-change version of our manuscript that shows the edits implemented.

Regarding the reviewer mention of the change in the FDRs and effect sizes for the OG*loss interaction: we would like to mention this may well stem from the different compositions of gene-tumor type pairs in each analysis subset: the original dataset comprised 82 gene-tumor type pairs, reducing to 69 upon exclusion of low-purity samples or to 72 pairs when subclonal mutations were removed. Further, sample sizes are reduced by removal of mutations, and the statistical power thus reduced is anticipated to affect FDRs.

In addition, when addressing my point about insufficient clarity in the model description, the authors state that they “have added detail to the Methods section, with regards to definition of variables (esp. the key variable describing copy number)”. I reported this as a minor comment in my previous revision because I thought that would be very easy to address. However, the extremely poor description of this section is, in my opinion, a strong deficiency of this work, since all the results reported are based on the application of the model.

In particular, it is clear that the authors used a generalized linear model to fit mutation counts Y , using selection coefficients ω and δ coupled with an indicator variable that discriminates target and baseline genes, plus other covariates. The clearer definition of the variables involved, their domain and any condition on them seems to be missing.

We thank the reviewer for this comment, and for listing below the specific items that need descriptions in the Methods section. Please see our detailed comments below, describing every clarification implemented to the text.

To be more precise, I think the authors should explain in the text what is that identifies a mutation for which counts Y are fitted, whether Y is a scalar number, a vector or a matrix, what is the dimension of it, what kind of variables are the "mutation type" m_i , what kind of variables are the μ_i , defined as the "corresponding effects", whether these variables are categorical, integers, real numbers positive or negative, what is the range of the index i . The same holds for β and ζ .

We have now added additional information (in red font in this Response document, see below for cited text) on the variables Y , m_i , and z_j , which explains the variables more precisely. The Y variable is the vector of mutation counts (non-negative integers) across all trinucleotide contexts for a given gene or its neighboring-genes .

“To describe the variability in mutation counts in a genomic locus, we model the expected mutation counts in vector $E[Y]$ using the following count model, implemented as generalized linear model with a log link function, further regularizing the regression coefficients by using a weakly informative prior [...]. [...]. [...]. To control for the different activities of mutational processes on different oligonucleotides, the model stratifies mutation counts (non-negative integers) according to the 96-component mutation-spectrum in trinucleotide context using the categorical mutation type variables m_i and their corresponding real-valued effects μ_i , where i iterates over the possible values that the mutation type variable can assume (such as $A[C>A]A$, $A[C>G]A$, etc.). Other types of optional variables can be included to control for e.g. inter-cancer type differences in selection (in a pan-cancer analysis), confounders or batch effects, such as the study where the data was sourced. We denote these variables as z_j (can be any type of variable) and their corresponding real-valued effects as β_j . [...]. [...]. [...].

Y has a length that corresponds to the product of distinct categories for each variable on the right side of the equation. For example, considering $i = 96$ mutation types for categorical variable m , 2 possible mutation loci (inside the central gene or inside the neighboring [control] genes) for the categorical variable t , and no additional dimensions (e.g. batch effects or tissues) meaning the variable z has only 1 possible value, the length of Y would be the product $2 \times 96 \times 1 = 192$. The corresponding effects μ_i , β are the fitted coefficients

(they quantify the effect of a variable they correspond to on the mutation rate: m_i and z_j , respectively) and can take any real-numbered value.”

We made edits to another place in the text to better clarify the dimensionality of Y , for the case of analysis studying CNA effects (central to our study):

"We encode the CNA state as a categorical variable, with 3 possible levels: CNA loss, CNA neutral (“diploid”) or CNA gain. The regression is performed to compare 2 levels of this variable at a time: loss versus neutral, excluding the gain state; or gain versus neutral, excluding the loss state. For each analysis, the length of the Y vector is 384, calculated as 2 (central vs. neighboring genes) \times 96 (mutation spectrum) \times 2 (diploid state vs. copy number altered state)."

Additionally we included about the type of variable for condition variable c :

"When testing for conditional selection signals in genes (here, differentiating CNA states of the gene), we use an extended version of the regression that includes a condition variable c encoding the state of the genomic region (can be a categorical or a continuous variable, depending on the condition of interest) with respect to the condition and the interaction term of the target variable t with the condition variable c ."

How is the number of nucleotides at risk $\log(r)$ defined?

The offset $\log(r)$, where r is the number of nucleotides-at-risk, transforms the expected number (count) of mutations $E[Y]$ to obtain mutation rates (i.e. mutations per nucleotide per sample). By using the log of the number of nucleotides at risk as an offset, we normalize the expected mutation counts to account for differences in the size of genomic regions (and, therefore, the opportunities for mutation to occur).

Using the offset for this purpose is a standard way of modeling the “number of opportunities” in a count model (“events” are mutations, and “opportunities” are the nucleotides available to be mutated). A transformation of the equation illustrates how this normalization is handled by the model:

$$\begin{aligned}\log E[Y] &= \omega t + \sum_i \mu_i m_i + \sum_j \beta_j z_j + \alpha + \log(r) \\ \log E[Y] - \log(r) &= \omega t + \sum_i \mu_i m_i + \sum_j \beta_j z_j + \alpha \\ \log \frac{E[Y]}{r} &= \omega t + \sum_i \mu_i m_i + \sum_j \beta_j z_j + \alpha\end{aligned}$$

We have adapted the Methods text to include the following clarification:

"Finally, to adjust the mutation counts to the maximal number of “opportunities” for mutations to occur, we include the number of nucleotides-at-risk r as an offset in the count model ~~exposure term~~. This allows us to implicitly normalize mutation counts to mutation rate (per nucleotide per sample), ~~model mutation rates per base-pair using mutation counts i.e. it~~ accountings for variations in ~~different~~ DNA lengths across

in-genomic loci and different composition of trinucleotide contexts in the central gene versus neighboring genes."

How is the positivity of the log-counts on the left side of the equation preserved, when on the right side we have negative terms (the selection coefficient can be negative)?

The log link function is commonly used to model count data, where the response variable (Y) is expected to be always non-negative, as counts cannot be negative. The log link function transforms the expected counts $E[Y]$ to real numbers, allowing the linear predictor to include both positive and negative values without violating the requirement that the counts themselves must be positive.

The expected counts ($E[Y]$) are obtained by applying exponentiation to the linear predictor. This guarantees that $E[Y]$ is positive, satisfying the necessary condition for count data.

$$E[Y] = e^{\omega t + \sum_i \mu_i m_i + \sum_j \beta_j z_j + \alpha + \log(r)}$$

The presence of a negative selection coefficient does not compromise the model's ability to predict positive expected counts due to the exponential transformation applied to the linear predictor.

We have added to the Methods text the following clarification: "The use of the log link function ensures that the mutation counts modeled are always non-negative, regardless of the sign of the regression coefficients."

In the definition of the outlier score S , M is defined as the "number of mutations observed in a gene", and L the "gene length", but what are S_x , S_y , M_x and M_y ?

In our manuscript, the variables S_x and S_y were not utilized. We think the reviewer may be referring to terms L_x and L_y , which represent the gene length for two specific genes of interest. We have now amended the part in the Methods describing the outlier score (as before, red font denotes changes):

"In cases where neighboring genes that do not share the CNA state of the central gene in the particular tumor sample, that sample was excluded from analysis. To reduce the effect of mutation rate heterogeneity between genes within the neighborhood, which could confound the framework, we exclude genes in the neighborhood that have a different mutation profile than the central gene, quantified by the outlier score $S_{outlier}$, defined as follows

$$S_{outlier} = \ln \frac{M_x/L_x}{M_y/L_y},$$

where M is the number of mutations observed in a gene (central gene or a gene from the neighborhood), and L is the gene length. The indices x and y refer to two different genes being compared: one is the central gene (x), and the other is a single gene from its neighborhood (y)."

In addition an example of a single fit, for instance for a gene of interest, would be very helpful to understand the model.

Following the recommendation of the reviewer, we added an one illustrative example to the manuscript, describing all quantities in detail:

“These corrected p-values were then utilized in all subsequent analyses, including the calculation of the FDR.

We illustrate our model with a single fit example using the *KRAS* gene in models with CNA gain interactions in ovarian cancer. The coefficients should be interpreted as follows: The intercept of -17.20 represents the log of the baseline mutation rate when all variables are set to their baseline levels, specifically the neighboring genes and the diploid states. Exponentiating this intercept, $e^{-17.2}$, gives the baseline mutation rate per nucleotide, per tumor sample. The *isTarget1* coefficient of 4.56 indicates that the mutation rate in the *KRAS* gene, when all other variables are at their base levels, is $e^{4.56}$ times the rate of the neighboring (baseline) genes, suggesting a significantly higher mutation rate (positive selection) in the diploid state. The coefficient of 0.28 for the *CNA* variable signifies that the mutation rate in the CNA gained state is $e^{0.28}$ times the rate in the diploid state for neighboring genes. The interaction coefficient *isTarget1:CNA1* of 0.38 reflects that the mutation rate in the *KRAS* gene in samples with a *KRAS* gain is $e^{0.38}$ times (1.32-fold) higher than what would be expected from the additive effects of the *KRAS* gene being selected without the CNA gain effect, and the CNA gain effect on the baseline mutation rates. For the *Mutation* variable, the coefficients ranging from -0.03 to 5.90 show how the mutation rates (in neighboring genes in the diploid state) vary across the trinucleotide spectrum, expressed relative to the baseline mutation type (arbitrarily set at A[C>A]A), with $e^{-0.03}$ to $e^{5.9}$ times the rate of A[C>A]A. Importantly, as all effect sizes in this study are presented on a \log_2 scale, exponentiation should use base 2 to derive the corresponding mutation rates.

To obtain selection estimates on point mutations across all copy number states, we considered all samples together without stratifying into different copy number states as in (1), and not controlling for any additional factors **z**.”

The level of approximation in the model description is reflected also in the poor software documentation, that is restricted to a very short README.

We thank the reviewer for their feedback. While it's true that the model description and software documentation have been concise, much of the information about how the algorithm functions was included within the documentation of the code for specific functions. For instance, the function *get_selection_estimates_neighbors* in its description contained information on general idea of the method; function *get_gene_neighbors* included in the documentation how the neighboring genes for gene of interest are chosen; function *post_process_pvalues* contained information about the approach used to correct the p-values, and functions *fit_selection_model* and *debias_selection_estimates* explained how the debiasing of selection effect sizes is performed.

Additionally, to aid users, we have now written a detailed *vignette* tutorial (found at <https://htmlpreview.github.io/?https://github.com/ebesedina/mutmatch/blob/main/vignettes/mutmatch.html>). This tutorial covers the basic usage of the package and is designed to be helpful for users looking to test the package and see if it fits their usage scenarios.

We hope this effort has improved the software documentation.

The illustration of the model in Figure 1a would also be much more useful to understand the methodology, if sufficiently improved: the blue rectangle in the legend called ‘baseline mutation rate’ is identical to the ones indicating the exons, which makes it of difficult interpretation. There are two windows zooming in the top curve in different regions: is the different annotation (“under positive selection” and “no selection”) due to a difference in the baseline mutation rate, in the selected gene’s rate or to the different genomic regions highlighted?. Is the same gene represented? A third window indicates the “comparison controls for trinucleotide” contexts: what is the difference with the previous two windows? Is there any relation with the fact that we are here “under negative selection”?

Thanks for the insightful comments on how to improve clarity of Figure 1a, which is helpful for understanding our methodology. The revised schematic now features zoom-ins on three distinct genomic regions to illustrate the mutation rates in three genes and their neighbors (control regions). It is crucial to note that the focus is on the relative mutation rates between the gene of interest and its neighboring genes, which serve as a neutral mutation rate baseline. A higher relative mutation rate in the central gene suggests positive selection, while a lower rate compared to its neighbors indicates negative selection. A mutation rate in the central gene similar to that of the neighboring genes signifies neutral selection. These comparisons are made while controlling for trinucleotide mutation spectra differences between central and neighboring genes (by stratifying for trinucleotides as we described in the response above, and elsewhere in the Methods), ensuring a rigorous test for selection.

The blue rectangles in this scheme indicate the borders of the genes (we do not show exons but rather the whole gene for simplicity): a lighter shade is used for the neighboring genes, a darker shade for the central gene. Baseline genomic mutation rate (curve above genes) is estimated using the neighboring genes, therefore the curve has the same color as the neighboring genes. To avoid confusion, we have now changed the visual legend on the figure to add three boxes for each of the models of selection (positive, negative and neutral) explaining why they are considered to be under selection (or not). Additionally we changed the label “comparison controls for trinucleotides” to “All comparisons control for trinucleotides” and added a scheme to make it clear that trinucleotide control was not only referring to the genes under negative selection. We also added a legend of colors for both central gene and the neighboring genes (note that exon-intron structure is not shown in the figure; one box represents one whole gene).

We hope that the modifications made to Figure 1a will enhance the clarity of our methodology and address the issues the reviewer has raised. The updated version Figure 1a schematic is shown below:

The description of this figure was updated in accordance with the above comments:

“a. Schematic depiction of the MutMatch method to estimate somatic selection by using exonic regions of neighboring genes as a mutation rate baseline, adjusting for trinucleotide composition. A higher relative mutation rate in the central gene suggests positive selection, while a lower rate compared to its neighbors indicates negative selection. A mutation rate in the central gene similar to that of the neighboring genes signifies neutral selection.”

I also think it would be valuable to report more summary statistics concerning the pre-processing and selection of data: what is the average number of neighbouring genes per target gene? Is it comparable from one gene to another and across chromosomes? What is the effect of the reported filtering strategies (e.g. removal of cases where the same CNV is not shared between neighbouring genes, removal of outlier genes) on these statistics? This should be clearly described and reported in the text.

To address the query about the pre-processing/selection of gene neighborhood data, we now provide some summary statistics for neighboring genes around cancer genes. Before filtering out outlier genes (with $|S_{\text{outlier}}| > 0.2$, denoting a large difference in mutation rates between that particular neighboring gene and the central gene), the median number of neighboring genes per target (central) gene was 9 ($Q_1 = 5$, $Q_3 = 19$ genes). After this filtering, the median number of genes decreased to 5 ($Q_1 = 2$, $Q_3 = 11$).

This number was broadly consistent across chromosomes, showing some variation in the distribution of neighboring genes (with median number of neighbors 4, and $Q_1 = 3$ and $Q_3 = 6.5$).

We incorporated changes reflecting this into the manuscript text as follows (in red):

“For each gene, neighborhood genes with $|S_{\text{outlier}}| > 0.2$ were excluded if their mutation rates differ by more than approximately 22% from that of the central gene, using an outlier score $|S_{\text{outlier}}|$ of 0.2, thus allowing less than 22% difference in mutation rates between the central gene and genes in the neighborhood. The median number of neighboring genes for cancer genes

after the removal of outlier genes was reduced from 9 to 5 genes (with $Q_1 = 2$ and $Q_3 = 11$), with some variation across chromosomes (Q_1 of medians = 3, Q_3 of medians = 6.5, with the overall median 4).”

Regarding removal of cases where the same CNV is not shared between neighboring genes: To reflect how many samples are removed in the filtering strategy we changed the manuscript text accordingly (in red):

“In cases where neighboring genes that do not share the CNA state of the central gene in the particular tumor sample, that sample was excluded from analysis. For cancer genes across various cancer types, the mean fraction of excluded samples was 0.56% for the diploid state ($Q_1 = 0$, $Q_3 = 0.82\%$), 1.06% for the gained state ($Q_1 = 0$, $Q_3 = 1.91\%$), and 0.85% for the deleted state ($Q_1 = 0$, $Q_3 = 0$).”

Examples on known and relevant genes (e.g. TP53, KRAS?) would be extremely helpful.

We added the following text to the Methods section right after the text above:

“For cancer genes such as *KRAS* (chromosome 12), *TP53* (chromosome 17), *BRAF* (chromosome 7) and *PIK3CA* (chromosome 3) the final count of neighboring genes after all filters were applied was 2 (for *KRAS* and *TP53*), 4 (for *BRAF*) and 9 (for *PIK3CA*). The filtering process, which involved excluding samples where neighboring genes had a differing copy number state from the central gene, led to the removal of the following percentages of samples across the 13 major cancer types: 3.5% in the diploid state, 10.5% in the gained state, and 11.2% in the deleted state for *KRAS*; 3.2% in the diploid state, 8.8% in the gained state, and 11.5% in the deleted state for *TP53*; 1.1% in the diploid state, 1.7% in the gained state, and 4.3% in the deleted state for *BRAF* and 0.8% in the diploid state, 1.4% in the gained state, and 1.7% in the deleted state for *PIK3CA*.”

Overall, I would invite the authors to clarify all these points so as to ensure that their results are more easily reproducible and the conclusions they draw are not affected by major biases.

We hope our clarifications addressed the reviewers’ concerns and that clarity (and, via our updates to code documentation, also the reproducibility) of our study has been improved.

Based on the diverse analytical approaches employed, as well as various internal controls and external replication, we are confident that the conclusions we draw regarding genetic interactions of CNA and driver mutations are not affected by major biases.

Reviewer #2 (Remarks to the Author):

Reviewer #3 (Remarks to the Author):

The authors have addressed all my comments and the manuscript is much improved as a result. I would only like to point out a few typos that require correction in the text:

49: missing closed bracket
58: remove `n`
75: below 'the' threshold of detection
95: 'an' -> 'any'
205: 'considered' -> 'we considered'
231-232: remove italics
314: remove additional full stop
325: remove additional comma
328-329: fix formatting
586-588: fix punctuation

I congratulate the authors for this excellent study.

We have corrected the specific textual issues highlighted by Reviewer 2, including the noted typos and formatting errors. We are grateful for the feedback.

Reviewer #3 (Remarks on code availability):

Detailed documentation is provided for the code, which is very helpful. Whilst I was able to install the `mutmatch` package, I struggled to run the example code for the `get_selection_estimates_neighbors()` function, firstly due to issues with accessing the example files within the library, and secondly, after manually loading the data, through a syntax error. However, whilst this should be checked again, I acknowledge that this may have been an issue of compatibility on my end.

We thank the reviewer for detailed feedback and for trying out the *mutmatch* package. We are glad the documentation was overall helpful. Regarding the issues encountered with the `get_selection_estimates_neighbors` function, it's possible that the problems may be related to specific configurations like the operating system used, R version, or Conda environment, as these issues do not reproduce on our testing setup.

We have renewed the package such that it currently installs in the minimal *R* environment that has *devtools* (tested with Pop!_OS (Ubuntu) 22.04 LTS):

```
conda create -n testbase r-base r-devtools  
conda activate testbase
```

Within this environment, we were able to install the package from *R* and successfully run the `get_selection_estimates_neighbors` function with the default parameters:

```
# Install the mutmatch package with devtools  
devtools::install_github("ebesedina/mutmatch")
```

There might be issues with installing the package with devtools when installing from the internet, and in this case installing from locally downloaded git clone is preferred.

```
git clone https://github.com/ebesedina/mutmatch.git
cd mutmatch
```

And then in R:

```
devtools::install()
```

If the dependency issues persist, please consider preinstalling the Conda environment provided with the package from the following link: <https://github.com/ebesedina/mutmatch/blob/main/environment.yml>

```
conda env create -f environment.yml
conda activate mutmatch
```

We appreciate the reviewer bringing this to our attention, and we will continue to monitor user reports on GitHub to address any recurring errors. Should you encounter any difficulties, please feel free to post an issue on our GitHub page.

Reviewer #4 (Remarks to the Author):

Reviewer #5 (Remarks to the Author):

The authors have done a serious effort to improve the general accessibility and readability of the manuscript, and the presentation is much better now in my opinion. My other concerns have been addressed in an ambitious way and the whole study has generally been improved, including a new randomization/P-value calibration procedure. I have no further concerns and wish to congratulate the authors on an impressive piece of work.

We appreciate the reviewer's constructive feedback and the acknowledgment of our efforts to refine the study and the manuscript.

REVIEWERS' COMMENTS

Reviewer #1 (Remarks to the Author):

The authors successfully addressed all the issues I raised and brought several improvements to the text, which I find now very clear for the broad readership of Nature Communications.

I congratulate the authors for this effort and in general for this interesting work.

Reviewer #1 (Remarks on code availability):

The authors have appropriately documented the software package, providing an useful tutorial that can be followed for the guided reproduction of their analysis.

Reviewer #2 (Remarks to the Author):
